# Towards Principled Test-Time Adaptation for Time Series Forecasting

## Abstract

Test-time adaptation (TTA) has recently emerged as a promising approach for improving time series forecasting (TSF) under distribution shift. In TSF, forecasting targets become observable after predictions are made, allowing forecasters to adapt as new observations arrive. Existing TSF-TTA methods differ in when and how they reuse these revealed targets, leading to different adaptation protocols. To address this issue, we introduce a protocol using only matured ground truth, defined as targets from fully observed forecasting horizons. This reliance on delayed but complete feedback places greater emphasis on the adapter's ability to derive useful prediction corrections. We then examine the corrections produced by existing adapters and find that they often vary smoothly across frequencies and exhibit limited localized spectral structure. Driven by this observation, we propose Frequency-Aware Calibration (FAC), a lightweight calibration method that directly parameterizes prediction corrections in the frequency domain. Across diverse settings, FAC remains competitive while using substantially fewer trainable parameters than other TSF-TTA adapters.

## 1 Introduction

Time series forecasting (TSF) plays a critical role in a wide range of domains, including weather (Wu et al., 2023; Verma et al., 2024), energy (Maleki et al., 2024), traffic (Yin et al., 2022; Zhang et al., 2024), and finance (Tsay, 2010; Koa et al., 2026). Recent advances in deep learning-based architectures have substantially enhanced predictive performance across diverse time series forecasting settings (Zeng et al., 2023; Nie et al., 2023; Liu et al., 2024).

Despite this progress, deploying time series forecasters to real-world applications remains challenging due to the complex and evolving nature of temporal data. In particular, distribution shifts induced by non-stationarity pose a major obstacle to reliable forecasting (Granger, 2003; Du et al., 2021), especially in long-horizon settings. To address this issue, prior studies have mainly pursued two paradigms: introducing learnable normalization modules (Kim et al., 2022; Fan et al., 2023; Liu et al., 2023), which primarily operate during pre-training, or training forecasting models in an online manner (Pham et al., 2023; Zhang et al., 2023; Liang et al., 2024; Lau et al., 2025; Huang et al., 2026). While these approaches can partially mitigate the effects of distribution shift, they either focus largely on the training phase or require training an online forecaster from scratch. In light of these limitations, applying test-time adaptation (TTA) (Kim et al., 2025; Medeiros et al., 2025; Grover & Etemad, 2025; Im & Kwon, 2026) to a source forecaster has emerged as a promising alternative.

Existing TSF-TTA methods differ in how they utilize revealed targets, with some (Kim et al., 2025; Medeiros et al., 2025; Grover & Etemad, 2025) incorporating partially-observed ground truth (POGT) together with full ground truth, which we refer to as **matured ground truth** throughout this paper, to enable more proactive adaptation. At the mini-batch level, a past mini-batch is considered as matured when the full target horizon for every sample in that mini-batch has been completely observed. Intuitively, with respect to the current batch, matured supervision lies entirely to the left of the end of the first look-back window in the current mini-batch. In contrast, COSA (Im & Kwon, 2026) is presented under a streaming formulation based on revealed ground truth, where an output adapter is updated on the current mini-batch using targets

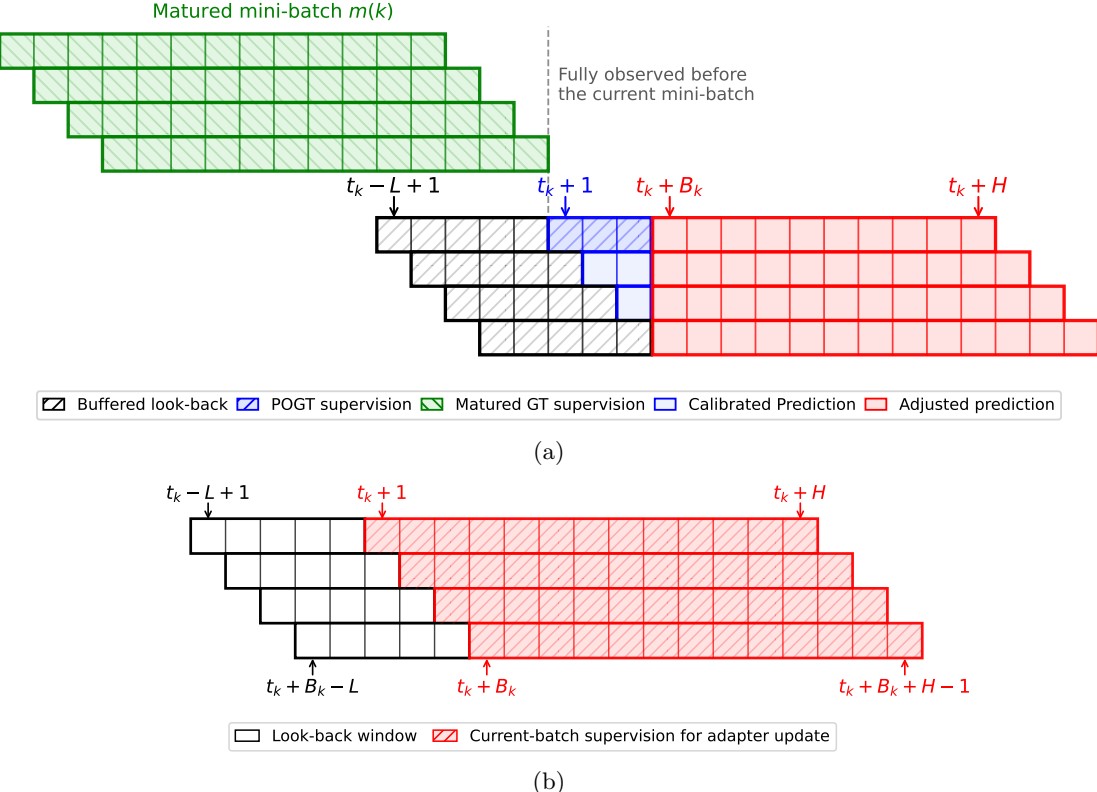

Figure 1: Comparison of adaptation supervision protocols in TSF-TTA. Here, $L$, $H$, and $B_k$ denote the look-back length, forecasting horizon, and mini-batch size, respectively, and $t_k$ denotes the time index immediately preceding the first target of mini-batch $k$. (a) Mixed supervision combines two temporally distinct sources: POGT from the current mini-batch, shown in blue and corresponding to $\mathcal{L}_k^{\text{POGT}}$ in Eq. 5, and full ground truth from the most recent matured mini-batch $m(k)$, shown in green and corresponding to $\mathcal{L}_{m(k)}^{\text{Matured}}$ in Eq. 6. Together, they form the mixed supervision objective in Eq. 4. (b) Streaming adaptation updates the adapter using current-mini-batch revealed-target supervision, indicated by the red shaded region and corresponding to $\mathcal{L}_k^{\text{stream}}$ in Eq. 11.

revealed for that mini-batch and lightweight context constructed from previously observed targets, without being explicitly framed in terms of matured ground truth. As illustrated in Figure 1, the key distinction between the two families lies in the timing and manner in which target observations are used for adaptation.

Despite these design differences, the protocol-level role of revealed targets in TSF-TTA remains insufficiently clarified. Our analysis demonstrates that existing TSF-TTA protocols lack a unified account of how revealed targets should be used for adaptation. To address this ambiguity, we formulate a protocol that restricts adaptation to matured ground truth in rolling forecasting settings. We further examine the prediction corrections of existing TSF-TTA methods in the frequency domain, finding that they often exhibit relatively smooth and weakly structured spectral patterns despite their different calibration designs. Based on this observation, we propose Frequency-Aware Calibration (FAC), which achieves competitive and consistent performance under the proposed protocol across diverse forecasting settings.

Our contributions are summarized as follows:

- We revisit how revealed targets are used in TSF-TTA and formulate an adaptation protocol using only matured ground truth.

- We provide a frequency-domain diagnosis of existing TSF-TTA methods, showing that their realized corrections often exhibit limited and weakly structured spectral modifications.

- We propose Frequency-Aware Calibration (FAC), a lightweight adapter that directly parameterizes calibration in the frequency domain and achieves competitive and consistent performance across diverse forecasting settings.

## 2 Related Work

### 2.1 Time Series Forecasting under Distribution Shift

A representative line of work addresses distribution shift in time series forecasting through learnable normalization or related distribution calibration mechanisms. RevIN (Kim et al., 2022) mitigates temporal distribution shift via reversible instance normalization and denormalization with learnable affine transformation. Dish-TS (Fan et al., 2023) characterizes both intra-space shift and inter-space shift, alleviating them through dual distribution modeling for the input and output spaces. SAN (Liu et al., 2023) further models non-stationarity by capturing statistical variation at the temporal-slice level. Although effective, these methods primarily improve robustness through training-time or architecture-level treatment of non-stationarity, rather than adapting a deployed source forecaster at test time.

Another line of work (Pham et al., 2023; Zhang et al., 2023; Liang et al., 2024; Lau et al., 2025; Huang et al., 2026) considers online time series forecasting (OTSF), where the forecasting model is continuously updated as new observations arrive. Recent studies (Liang et al., 2024; Lau et al., 2025) have pointed out that conventional OTSF protocols may suffer from evaluation leakage or unrealistic forecasting setups. In particular, Act-Now (Liang et al., 2024) and DSOF (Lau et al., 2025) revisit online forecasting protocols and advocate cleaner evaluation settings that avoid evaluating predictions on steps already used for model updates. Within this line of work, Act-Now (Liang et al., 2024) introduces a label decomposition model to tackle distribution shift, while DSOF (Lau et al., 2025) adopts a dual-stream teacher-student framework for online forecasting. ADAPT-Z (Huang et al., 2026) further studies the challenge of delayed-feedback online prediction and argues that distribution shift may be better addressed through feature-level adjustment. These approaches are related to TSF-TTA in spirit, but the primary objectives differ: OTSF focuses on training the online forecaster itself, while TSF-TTA adapts a frozen source forecaster at test time.

### 2.2 Test-Time Adaptation for Time Series Forecasting

Test-time adaptation (TTA) (Wang et al., 2021; 2022; Gong et al., 2023; Liang et al., 2025; Kim et al., 2026) has been widely studied in domains such as computer vision, where models are adapted at test time to improve robustness under distribution shift. Unlike conventional TTA methods (Wang et al., 2021; 2022; Gong et al., 2023; Kim et al., 2026), which typically rely on unlabeled test-time data, TSF-TTA can exploit forecasting targets that become sequentially revealed after prediction. As the pioneering work on TSF-TTA, TAFAS (Kim et al., 2025) incorporates both matured ground truth and partially observed ground-truth (POGT), aiming to enable earlier and more proactive adaptation. PETSA (Medeiros et al., 2025) further enhances parameter efficiency by introducing low-rank adapters together with a specialized adaptation loss. DynaTTA (Grover & Etemad, 2025) employs dynamic adaptation rates and shift-conditioned gating based on real-time estimation of distribution shift. In contrast, COSA (Im & Kwon, 2026) is presented under a streaming formulation based on revealed ground truth, where a single output adapter corrects the current mini-batch prediction using lightweight context from previously revealed targets and is updated directly against the targets of the current mini-batch. These methods therefore differ not only in adapter design, but also in when and how revealed targets are incorporated into adaptation.

## 3 Revisiting Adaptation Protocols for TSF-TTA

### 3.1 Preliminaries: Rolling TSF-TTA Formulation

Existing TSF-TTA (Kim et al., 2025; Medeiros et al., 2025; Grover & Etemad, 2025; Im & Kwon, 2026) methods adopt a rolling test-time adaptation setting, in which a pre-trained source forecaster is deployed on temporally ordered test data under distribution shift. Following the "test mini-batch" terminology used

in TAFAS (Kim et al., 2025), a mini-batch in this paper refers to a contiguous, temporally ordered block of rolling windows from a single multivariate time series, rather than a shuffled training batch. Predictions are produced for all samples within each mini-batch. Let $L$ and $H$ denote the look-back length and forecasting horizon, respectively. We index rolling mini-batches by $k$, and let $B_k$ denote the size of the $k$-th mini-batch. We further let $t_k$ denote the global time index immediately preceding the first forecasted target of the $k$-th mini-batch. Since rolling windows are grouped consecutively, the next mini-batch starts after $B_k$ shifts, yielding $t_{k+1} = t_k + B_k$. For the $j$-th sample of the $k$-th mini-batch, where $j \in \{1, \ldots, B_k\}$, the input window, the corresponding forecast, and the corresponding target vector are given by

$$\mathbf{X}_k^{[j]} = [\mathbf{x}_{t_k+j-L}, \ldots, \mathbf{x}_{t_k+j-1}], \quad \hat{\mathbf{Y}}_k^{[j]} = [\hat{\mathbf{y}}_{t_k+j}^{[j]}, \ldots, \hat{\mathbf{y}}_{t_k+j+H-1}^{[j]}], \quad \mathbf{Y}_k^{[j]} = [\mathbf{y}_{t_k+j}, \ldots, \mathbf{y}_{t_k+j+H-1}]. \quad (1)$$

where $\mathbf{x}_t \in \mathbb{R}^C$ denotes the multivariate observation at time step $t$, and $\hat{\mathbf{y}}_t^{[j]} \in \mathbb{R}^C$ denotes the prediction of the $j$-th sample at time step $t$. Note that $\hat{\mathbf{y}}_t^{[j]}$ differs for different samples in the mini-batch.

The forecasting target span of the $k$-th mini-batch can then be denoted as the union of all sample-level targets within the mini-batch:

$$\mathfrak{Y}_k = \bigcup_{j=1}^{B_k} \mathbf{Y}_k^{[j]} = \{\mathbf{y}_{t_k+1}, \ldots, \mathbf{y}_{t_k+H+B_k-1}\}. \quad (2)$$

### 3.2 A Closer Look at Existing Supervision Protocols

#### 3.2.1 Mixed Supervision with POGT and Matured Ground Truth

A representative line of TSF-TTA methods (Kim et al., 2025; Medeiros et al., 2025; Grover & Etemad, 2025) adopts a mixed supervision protocol that combines POGT from the current mini-batch with supervision from the most recent matured mini-batch, as illustrated in Figure 1a. In these methods, the POGT length $p_k$ is determined by the periodicity-aware adaptation scheduling (PAAS), with $p_k = B_k - 1$. For notational consistency, we will use $B_k - 1$ to denote the POGT length below.

Let $\mathcal{C}_{k-1}^{\text{in}}(\cdot)$ and $\mathcal{C}_{k-1}^{\text{out}}(\cdot)$ denote the input and output calibration modules available before adapting on the $k$-th mini-batch. For the $j$-th sample in that mini-batch, the calibrated prediction is

$$\hat{\mathbf{Y}}_k^{\text{cali},[j]} = \mathcal{C}_{k-1}^{\text{out}}(\mathcal{F}(\mathcal{C}_{k-1}^{\text{in}}(\mathbf{X}_k^{[j]}))) = [\hat{\mathbf{y}}_{t_k+j}^{\text{cali},[j]}, \ldots, \hat{\mathbf{y}}_{t_k+j+H-1}^{\text{cali},[j]}], \quad (3)$$

where $\mathcal{F}(\cdot)$ denotes the frozen source forecaster. Let $m(k)$ denote the index of the most recent matured mini-batch. Then the mixed supervision protocol can be abstractly written as

$$\mathcal{L}_k^{\text{mixed}} = \mathcal{L}_k^{\text{POGT}} + \mathcal{L}_{m(k)}^{\text{Matured}}, \quad (4)$$

with

$$\mathcal{L}_k^{\text{POGT}} = \text{MSE}\Big(\{\hat{\mathbf{y}}_t^{\text{cali},[1]}\}_{t=t_k+1}^{t_k+B_k-1}, \{\mathbf{y}_t\}_{t=t_k+1}^{t_k+B_k-1}\Big). \quad (5)$$

and

$$\mathcal{L}_{m(k)}^{\text{Matured}} = \text{MSE}\Big(\{\hat{\mathbf{Y}}_{m(k)}^{\text{cali},[j]}\}_{j=1}^{B_{m(k)}}, \{\mathbf{Y}_{m(k)}^{[j]}\}_{j=1}^{B_{m(k)}}\Big). \quad (6)$$

The prediction and target arguments in Eqs. 5 and 6 have dimensions $(B_k - 1) \times C$ and $B_{m(k)} \times H \times C$, respectively. Equation 4 combines two temporally distinct supervision signals: $\mathcal{L}_k^{\text{POGT}}$ enables timely adaptation using the revealed prefix of the current mini-batch, whereas $\mathcal{L}_{m(k)}^{\text{Matured}}$ provides complete-horizon supervision from the most recent matured mini-batch. In implementation, these two terms may be consumed in separate adaptation steps, but together they define the mixed supervision protocol. After adaptation, the calibration modules are updated to $\mathcal{C}_k^{\text{in}}(\cdot)$ and $\mathcal{C}_k^{\text{out}}(\cdot)$. The current mini-batch is then re-forecast using the updated calibration modules:

$$\hat{\mathbf{Y}}_k^{\text{adp},[j]} = \mathcal{C}_k^{\text{out}}\Big(\mathcal{F}\Big(\mathcal{C}_k^{\text{in}}(\mathbf{X}_k^{[j]})\Big)\Big). \quad (7)$$

The adjusted prediction keeps the already observed prefix from the original calibrated prediction and replaces the unobserved suffix with the re-forecasted one, which is given by

$$\tilde{\mathbf{y}}_{k,t}^{[j]} = \begin{cases} \hat{\mathbf{y}}_t^{\text{cali},[j]}, & t_k + j \leq t \leq t_k + B_k - 1, \\ \hat{\mathbf{y}}_t^{\text{adp},[j]}, & t_k + B_k \leq t \leq t_k + j + H - 1. \end{cases} \tag{8}$$

However, by the time POGT becomes available for adaptation, the same revealed values have already entered the look-back windows of later samples within the same mini-batch. **This raises a natural question: is it more beneficial to use these revealed values as supervision for adaptation, or simply as additional input context for later direct prediction?** To examine this, we compare the MSE on the shared overlapping horizon between the adjusted prediction of the first sample and the direct predictions of later samples within the same mini-batch.

As illustrated in Figure 2a, the endpoint comparison provides the most direct way to assess whether using the revealed values as supervision offers a meaningful advantage over simply using the same values as input context in later direct prediction. At the same time, this endpoint view is only the most intuitive illustration of a broader phenomenon within the mini-batch. To examine this more comprehensively, we compute the mean overlapping-region MSE at each sample position over the mini-batches with $B_k = 97$, since this is the most common mini-batch size induced by PAAS in our experiments.

Let $\mathcal{I}_{\{B_k=B\}}$ denote the set of mini-batches satisfying $B_k = B$. For the direct prediction of the $j$-th sample, we define the mean overlapping-region MSE as

$$\hat{M}_{\mathcal{I}_{\{B_k=B\}}}^{[j]} = \frac{1}{\left|\mathcal{I}_{\{B_k=B\}}\right|} \sum_{k \in \mathcal{I}_{\{B_k=B\}}} \text{MSE}\left([\hat{\mathbf{y}}_{t_k+B}^{[j]}, \ldots, \hat{\mathbf{y}}_{t_k+H}^{[j]}], [\mathbf{y}_{t_k+B}, \ldots, \mathbf{y}_{t_k+H}]\right). \tag{9}$$

The corresponding metric for the adjusted prediction, $\tilde{M}_{\mathcal{I}_{\{B_k=B\}}}^{[j]}$, is defined analogously by replacing $\hat{\mathbf{y}}^{[j]}$ with $\tilde{\mathbf{y}}^{[j]}$. Using the metrics computed for each sample $j$ within the mini-batch, Figure 2b shows that the endpoint comparison reflects a broader pattern. Across multiple datasets, forecasting horizons and frozen source forecasters, the adjusted prediction of the first sample is generally inferior to the direct prediction of the last sample. In fact, the direct predictions from many intermediate samples already achieve comparable or lower overlapping-region MSE than the adjusted prediction of the first sample. These results indicate that using revealed values as supervision is not consistently more beneficial than simply using the same values as input context for later direct prediction.

### 3.2.2 Streaming Adaptation with Revealed Ground Truth

COSA (Im & Kwon, 2026) adopts a streaming adaptation protocol based on revealed ground truth, as illustrated in Figure 1b. Under this protocol, the current mini-batch is first predicted by the frozen source forecaster, after which an output adapter produces the final adapter-corrected prediction using a lightweight context vector constructed from previously revealed targets. Unlike the mixed-supervision protocol above, the output adapter is then updated directly by minimizing the MSE loss between these adapter-corrected predictions of the current mini-batch and the revealed ground truth of the current mini-batch.

Let $\mathcal{A}_{k-1}(\cdot, \cdot)$ denote the output adapter available before adapting on the $k$-th mini-batch, and let $\mathbf{c}_{k-1}$ denote the context vector constructed from previously revealed targets. For the $j$-th sample in the $k$-th mini-batch, the final streaming prediction is given by

$$\hat{\mathbf{Y}}_k^{\text{stream},[j]} = \mathcal{A}_{k-1}(\mathcal{F}(\mathbf{X}_k^{[j]}), \mathbf{c}_{k-1}). \tag{10}$$

After the targets of the current mini-batch are revealed, their summary statistic is added to the context memory, updating $\mathbf{c}_{k-1}$ to $\mathbf{c}_k$. The adapter is then optimized on the same mini-batch using the updated context; further implementation details are provided in Appendix B.2. Accordingly, the streaming adaptation objective can be written as

$$\mathcal{L}_k^{\text{stream}} = \frac{1}{B_k} \sum_{j=1}^{B_k} \text{MSE}(\hat{\mathbf{Y}}_k^{\text{stream},[j]}, \mathbf{Y}_k^{[j]}). \tag{11}$$

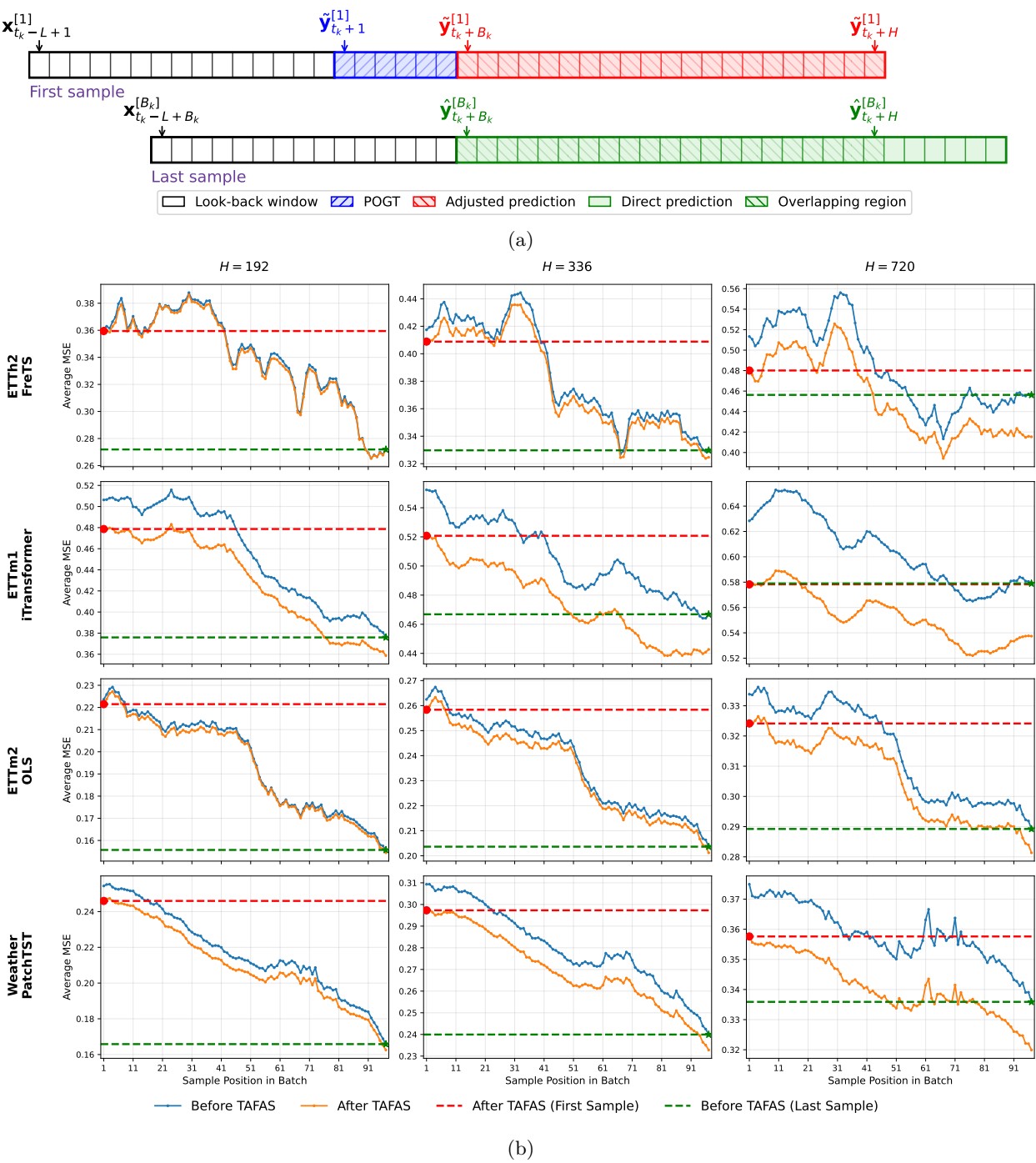

Figure 2: Comparison between the adjusted prediction of the first sample and later direct predictions within the same mini-batch. (a) Schematic illustration of the overlapping horizon shared by the first sample's adjusted prediction and the last sample's direct prediction. (b) Mean overlapping-region MSE at each sample position, averaged over mini-batches with $B_k = 97$ across representative datasets, horizons, and source forecasters. The blue and orange curves denote direct and adjusted predictions, respectively; the red dashed line marks the adjusted prediction of the first sample, and the green dashed line marks the direct prediction of the last sample. Additional results for $B_k = 25$ are provided in Appendix A.3.

Although the protocol is presented in terms of revealed ground truth, such supervision is not necessarily matured relative to subsequent rolling predictions. Recall that the target span of the $k$-th mini-batch is $\mathfrak{Y}_k = \{\mathbf{y}_{t_k+1}, \ldots, \mathbf{y}_{t_k+H+B_k-1}\}$. For the $(k+1)$-th mini-batch, since $t_{k+1} = t_k + B_k$, the prediction span of the $j$-th sample is

$$\hat{\mathbf{Y}}_{k+1}^{\text{stream},[j]} = [\hat{\mathbf{y}}_{t_k+B_k+j}^{\text{stream},[j]}, \ldots, \hat{\mathbf{y}}_{t_k+B_k+j+H-1}^{\text{stream},[j]}]. \tag{12}$$

When the forecasting horizon is sufficiently larger than the mini-batch size, the targets used to update on batch $k$ may still overlap with the future prediction span of later mini-batches. For example, if $H \geq B_k + B_{k+1}$, then

$$\{t_k+1, \ldots, t_k+H+B_k-1\} \cap \{t_k+B_k+j, \ldots, t_k+B_k+j+H-1\} = \{t_k+B_k+j, \ldots, t_k+H+B_k-1\} \neq \emptyset. \tag{13}$$

This shows that the supervision used to update on batch $k$ is not necessarily matured relative to subsequent rolling prediction spans.

### 3.3 A Protocol Based Solely on Matured Ground Truth

In the mixed-supervision setting, the use of POGT naturally induces a comparison between using revealed values as supervision and using the same values as additional input context for later direct prediction. In the streaming setting, supervision based on revealed ground truth is not necessarily matured relative to subsequent rolling predictions. These observations motivate us to formulate an alternative protocol that uses only matured ground truth.

The proposed protocol is defined by the temporal status of the supervision it permits, rather than by a unique implementation rule. For example, one may update based on any subset of available matured mini-batches. Formally, denote the index set of matured mini-batches available at mini-batch $k$ as

$$\mathcal{M}_k := \{m < k : t_m + B_m + H - 1 \leq t_k\}. \tag{14}$$

Then a general adaptation objective based on matured ground truth may be written as

$$\mathcal{L}_k^{\text{mat}} = \sum_{m \in \mathcal{S}_k} \omega_{k,m} \mathcal{L}_m^{\text{matured}}, \qquad \mathcal{S}_k \subseteq \mathcal{M}_k, \tag{15}$$

where $\mathcal{S}_k$ denotes the subset of matured mini-batches selected for adaptation at mini-batch $k$, $\omega_{k,m} \geq 0$ are combination weights, and $\mathcal{L}_m^{\text{matured}}$ is defined by Eq. 6, with $m$ replacing $m(k)$.

Under our proposed protocol based solely on matured ground truth, predictions may still be organized in mini-batches, but the role of the mini-batch becomes different. Conceptually, the protocol no longer depends on supervision drawn from the current mini-batch; instead, adaptation is triggered only by previously matured targets. In this sense, the mini-batch mainly serves as an implementation-level grouping for prediction and evaluation. Moreover, although predictions within a mini-batch still overlap due to the rolling construction, the comparison between earlier and later samples in the overlapped region is no longer central to the protocol itself, since adaptation for each sample no longer relies on POGT from the current mini-batch. Rather, adaptation can already be triggered using matured past supervision before later samples in the current mini-batch are generated.

## 4 Frequency-Aware Calibration

Having established a protocol based solely on matured ground truth, we now turn to the design of the adaptation module. Under this protocol, adaptation no longer uses the timely feedback provided by POGT within the current mini-batch and instead relies on delayed supervision from matured mini-batches. This places greater emphasis on the adapter's ability to derive useful prediction corrections from the delayed supervision that remains available. Existing TSF-TTA adapters differ primarily in how they parameterize calibration. TAFAS (Kim et al., 2025) calibrates predictions through temporal-domain transformations, whereas PETSA (Medeiros et al., 2025) introduces spectral information through a frequency-domain loss applied to a low-rank temporal calibration module. Although temporal-domain transformations can also

realize global and frequency-selective corrections, these effects are represented indirectly through their temporal parameterization. Empirically, we observe that the resulting corrections often vary smoothly across frequencies and exhibit limited localized spectral structure. This observation motivates Frequency-Aware Calibration (FAC), which directly parameterizes corrections in the frequency domain.

The key intuition behind FAC is that each Fourier basis function extends across the full temporal window, so modifying its coefficient induces coordinated changes across multiple temporal positions after the inverse Fourier transform. FAC parameterizes calibration through complex coefficients associated with individual Fourier components, making the amplitude and phase of each component directly adjustable. In this way, FAC can represent structured corrections through frequency-wise affine transformations without requiring a general dense transformation across temporal positions.

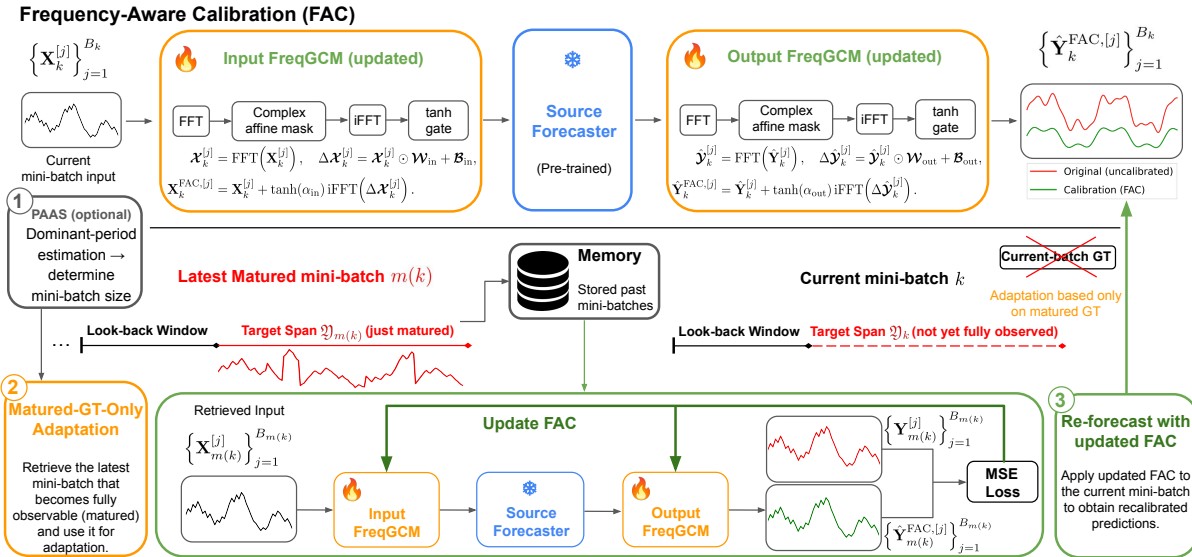

Figure 3: Overview of Frequency-Aware Calibration (FAC) under the proposed protocol. FAC uses input and output frequency-domain gated calibration modules (FreqGCMs), each applying a complex affine mask in the frequency domain followed by inverse FFT and gated residual correction. For simplicity, our implementation retrieves the most recent matured mini-batch for each update, after which the updated adapter is applied to re-forecast the current mini-batch. PAAS can optionally be used to determine the mini-batch size.

FAC adopts a minimalist design, since a central requirement for test-time adaptation in TSF is to keep the adapter lightweight. Figure 3 provides an overview. Similar to TAFAS (Kim et al., 2025) and PETSA (Medeiros et al., 2025), FAC places calibration modules before and after the frozen source forecaster. Each module performs a gated residual correction to the signal being calibrated, rather than directly replacing it. Specifically, FAC consists of an input and an output FreqGCM. Each FreqGCM first transforms the signal being calibrated into the frequency domain, applies a learnable complex affine mask, maps the frequency-domain correction back to the temporal domain through inverse FFT, and then controls the correction magnitude through a hyperbolic-tangent gate. On the input side, the input FreqGCM for the $j$-th sample in the $k$-th mini-batch is given by

$$
\begin{aligned}
\boldsymbol{\mathcal{X}}_k^{[j]} &= \mathrm{FFT}\Big(\mathbf{X}_k^{[j]}\Big), \\
\Delta\boldsymbol{\mathcal{X}}_k^{[j]} &= \boldsymbol{\mathcal{X}}_k^{[j]} \odot \boldsymbol{\mathcal{W}}_{\mathrm{in}} + \boldsymbol{\mathcal{B}}_{\mathrm{in}}, \\
\mathbf{X}_k^{\mathrm{FAC},[j]} &= \mathbf{X}_k^{[j]} + \tanh(\alpha_{\mathrm{in}})\,\mathrm{iFFT}\Big(\Delta\boldsymbol{\mathcal{X}}_k^{[j]}\Big).
\end{aligned}
\tag{16}
$$

The calibrated input is then passed to the frozen source forecaster as $\hat{\mathbf{Y}}_k^{[j]} = \mathcal{F}(\mathbf{X}_k^{\mathrm{FAC},[j]})$. The output FreqGCM follows the same form on $\hat{\mathbf{Y}}_k^{[j]}$ with output-side parameters $(\boldsymbol{\mathcal{W}}_{\mathrm{out}}, \boldsymbol{\mathcal{B}}_{\mathrm{out}}, \alpha_{\mathrm{out}})$, yielding the final calibrated forecast $\hat{\mathbf{Y}}_k^{\mathrm{FAC},[j]}$. Let $F_{\mathrm{in}} = \lfloor L/2 \rfloor + 1$ and $F_{\mathrm{out}} = \lfloor H/2 \rfloor + 1$ denote the numbers of frequency

components. The complex affine parameters satisfy $\boldsymbol{\mathcal{W}}_{\text{in}}, \boldsymbol{\mathcal{B}}_{\text{in}} \in \mathbb{C}^{F_{\text{in}} \times C}$ and $\boldsymbol{\mathcal{W}}_{\text{out}}, \boldsymbol{\mathcal{B}}_{\text{out}} \in \mathbb{C}^{F_{\text{out}} \times C}$, while $\alpha_{\text{in}}, \alpha_{\text{out}} \in \mathbb{R}^C$ are per-channel gates. We write FFT/iFFT for notational simplicity, while using rFFT/irFFT in implementation for real-valued time series. The element-wise affine form keeps FAC lightweight and frequency-wise interpretable: the multiplicative term rescales and phase-shifts Fourier coefficients, while the additive term can modify weak or near-zero components that purely multiplicative scaling may miss. The residual form preserves the original signal when the learned correction is small, and the tanh gate controls the overall correction strength.

Our protocol allows adaptation on any subset of matured ground truth; for simplicity, FAC uses the most recent matured mini-batch:

$$\mathcal{L}_{m(k)}^{\text{FAC}} = \text{MSE}\left( \{\hat{\mathbf{Y}}_{m(k)}^{\text{FAC},[j]}\}_{j=1}^{B_{m(k)}}, \{\mathbf{Y}_{m(k)}^{[j]}\}_{j=1}^{B_{m(k)}} \right). \tag{17}$$

Although additional regularization terms could be added to this objective, we keep the adaptation loss as a simple MSE on matured ground truth. This isolates the effect of frequency-domain parameterization and avoids conflating it with additional regularization choices. After the FAC modules are updated, the current mini-batch is re-forecast with the updated input and output FreqGCMs to obtain recalibrated predictions.

## 5 Experiments

### 5.1 Experimental Settings

We conduct our primary evaluation on six multivariate TSF benchmark datasets: ETTh1, ETTh2, ETTm1, ETTm2, Weather, and Exchange. We use chronological train/validation/test splits for all datasets: the four ETT datasets use a ratio of $(0.6, 0.2, 0.2)$, while Weather and Exchange use $(0.7, 0.1, 0.2)$. The lookback length is set to $L = 96$, and the prediction horizon is chosen from $H \in \{96, 192, 336, 720\}$, except for Exchange, where we use $H \in \{96, 192, 336\}$ because the full target horizon for $H = 720$ does not mature before the end of the test sequence. Consistent with existing TSF-TTA studies, we evaluate five representative source forecasters on these primary datasets: iTransformer (Liu et al., 2024), PatchTST (Nie et al., 2023), DLinear (Zeng et al., 2023), OLS (Toner & Darlow, 2024), and FreTS (Yi et al., 2023) [1].

Recent forecasting benchmarks, such as GIFT-Eval, emphasize coverage across application domains and varying numbers of variates (Aksu et al., 2024). Motivated by these considerations, we additionally evaluate FAC on Electricity and Traffic (Zhou et al., 2021), two multivariate benchmarks from the energy and transportation domains. GIFT-Eval includes Electricity in its evaluation component and Traffic Hourly in its pretraining component. In our multivariate LTSF formulation, these datasets contain 321 and 862 variables, respectively, broadening the range of variable counts and application domains covered by our evaluation. We evaluate DLinear, FreTS, and iTransformer as three representative source forecasters spanning linear, MLP-based, and Transformer-based architectures, respectively. Each forecaster is evaluated on both datasets across $H \in \{96, 192, 336, 720\}$, yielding 24 model–horizon settings in total.

We compare FAC with the frozen source forecaster without TTA, TAFAS (Kim et al., 2025), and PETSA (Medeiros et al., 2025) under the proposed protocol based solely on matured ground truth [2]. For fairness and simplicity, only the most recent matured mini-batch is used for adaptation across methods. Since the current mini-batch does not provide supervision under this protocol, the POGT-based partial prediction adjustment in the original TAFAS protocol is no longer used. Instead, after the adapter is updated on the most recent matured mini-batch, the current mini-batch is re-forecast and the full prediction horizon is evaluated using the updated adapter. In particular, for the $j$-th sample in the $k$-th mini-batch, we use

$$\tilde{\mathbf{Y}}_k^{[j]} = \hat{\mathbf{Y}}_k^{\text{adp},[j]}, \quad j = 1, \ldots, B_k, \tag{18}$$

---

[1] We omit MICN (Wang et al., 2023) from the main experiments because its architecture categorization is ambiguous in prior TSF-TTA comparisons. Although some studies group it with MLP-based forecasters, MICN is more accurately viewed as a multi-scale convolution-based model.

[2] We omit DynaTTA due to reproducibility issues with its officially released implementation; similar issues were also noted in COSA (Im & Kwon, 2026). We do not include a matured-only COSA variant because, unlike TAFAS and PETSA, its target-derived context and immediate streaming update do not admit a unique direct conversion to our protocol; see Appendix B for a detailed comparison.

rather than a partial stitching rule between pre-update and post-update predictions. Hyperparameters are taken from the officially released code whenever available. All experiments were conducted on an NVIDIA GeForce RTX 4070 Super 12GB and an NVIDIA GeForce RTX 4090 24GB.

## 5.2 Main Results

### 5.2.1 Forecasting Performance

Table 1: Forecasting performance under the proposed protocol using only matured ground truth. Lower MSE is better. The best result within each backbone and horizon is highlighted in red bold, and the second-best result is underlined.

| Dataset | H | DLinear | | | | FreTS | | | | iTransformer | | | | OLS | | | | PatchTST | | | |
|---|---|---|---|---|---|---|---|---|---|---|---|---|---|---|---|---|---|---|---|---|---|
| | | Base | TAFAS | PETSA | **FAC** | Base | TAFAS | PETSA | **FAC** | Base | TAFAS | PETSA | **FAC** | Base | TAFAS | PETSA | **FAC** | Base | TAFAS† | PETSA | **FAC** |
| **ETTh1** | 96 | 0.4695 | 0.4617 | _0.4613_ | **0.4554** | 0.4461 | 0.4391 | _0.4383_ | **0.4357** | 0.4482 | 0.4421 | **0.4394** | _0.4400_ | 0.4510 | _0.4428_ | 0.4449 | **0.4382** | 0.4324 | 0.4270 | _0.4255_ | **0.4241** |
| | 192 | 0.5213 | _0.5112_ | 0.5123 | **0.5089** | 0.5022 | _0.4943_ | 0.4960 | **0.4934** | 0.5091 | 0.5020 | _0.5014_ | **0.5010** | 0.5045 | _0.4929_ | 0.5007 | **0.4921** | 0.4900 | _0.4818_ | 0.4818 | **0.4797** |
| | 336 | 0.5659 | _0.5595_ | 0.5626 | **0.5582** | 0.5544 | **0.5483** | 0.5530 | _0.5498_ | 0.5662 | _0.5624_ | 0.5641 | **0.5587** | 0.5510 | _0.5427_ | 0.5488 | **0.5422** | 0.5608 | **0.5506** | 0.5570 | _0.5525_ |
| | 720 | 0.7117 | _0.6926_ | 0.6938 | **0.6891** | 0.7181 | **0.6857** | _0.6908_ | 0.7067 | 0.7007 | _0.6755_ | 0.6763 | **0.6687** | 0.6996 | **0.6704** | 0.6805 | _0.6780_ | 0.7112 | **0.6807** | _0.7019_ | 0.7096 |
| **ETTh2** | 96 | 0.2323 | _0.2308_ | 0.2312 | **0.2288** | 0.2385 | _0.2363_ | 0.2366 | **0.2338** | 0.2642 | 0.2620 | _0.2602_ | **0.2561** | 0.2306 | _0.2289_ | 0.2293 | **0.2274** | 0.2385 | _0.2380_ | 0.2383 | **0.2374** |
| | 192 | 0.2862 | 0.2830 | _0.2820_ | **0.2819** | 0.2867 | _0.2830_ | 0.2844 | **0.2814** | 0.3081 | _0.3004_ | 0.3005 | **0.2970** | 0.2838 | _0.2821_ | 0.2829 | **0.2796** | 0.2840 | _0.2752_ | 0.2810 | **0.2747** |
| | 336 | 0.3252 | _0.3175_ | 0.3190 | **0.3144** | 0.3318 | _0.3195_ | 0.3250 | **0.3172** | 0.3402 | _0.3290_ | 0.3312 | **0.3284** | 0.3257 | _0.3189_ | 0.3240 | **0.3137** | 0.3233 | _0.3108_ | 0.3190 | **0.3079** |
| | 720 | 0.4087 | **0.3825** | 0.3884 | _0.3825_ | 0.4118 | _0.3814_ | 0.3968 | **0.3788** | 0.4261 | **0.3959** | 0.4049 | _0.3982_ | 0.4161 | _0.3859_ | 0.3927 | **0.3854** | 0.4292 | _0.4065_ | 0.4203 | **0.4002** |
| **ETTm1** | 96 | 0.3715 | _0.3504_ | 0.3541 | **0.3465** | 0.3676 | _0.3559_ | 0.3572 | **0.3517** | 0.3823 | _0.3680_ | 0.3700 | **0.3459** | 0.3709 | _0.3542_ | 0.3591 | **0.3462** | 0.4041 | 0.3826 | _0.3817_ | **0.3530** |
| | 192 | 0.4438 | _0.4165_ | 0.4192 | **0.4139** | 0.4325 | 0.4191 | _0.4189_ | **0.4183** | 0.4407 | _0.4189_ | 0.4189 | **0.4087** | 0.4437 | _0.4184_ | 0.4252 | **0.4140** | 0.4515 | _0.4370_ | 0.4405 | **0.4213** |
| | 336 | 0.5183 | _0.4801_ | 0.4831 | **0.4791** | 0.5006 | _0.4810_ | **0.4793** | 0.4836 | 0.5077 | 0.4800 | _0.4788_ | **0.4781** | 0.5179 | _0.4811_ | 0.4896 | **0.4794** | 0.5044 | _0.4864_ | 0.4970 | **0.4787** |
| | 720 | 0.5929 | _0.5521_ | 0.5600 | **0.5494** | 0.5703 | **0.5472** | _0.5523_ | 0.5533 | 0.6021 | _0.5579_ | 0.5615 | **0.5460** | 0.5923 | _0.5515_ | 0.5594 | **0.5497** | 0.5608 | _0.5413_ | 0.5479 | **0.5399** |
| **ETTm2** | 96 | 0.1598 | _0.1581_ | 0.1583 | **0.1560** | 0.1582 | _0.1566_ | 0.1569 | **0.1563** | 0.1647 | _0.1640_ | 0.1640 | **0.1613** | 0.1602 | _0.1594_ | 0.1595 | **0.1563** | 0.1581 | 0.1569 | _0.1569_ | **0.1558** |
| | 192 | 0.1930 | _0.1917_ | 0.1920 | **0.1894** | 0.1923 | **0.1901** | _0.1908_ | 0.1901 | 0.2189 | 0.2164 | _0.2157_ | **0.2067** | 0.1935 | _0.1924_ | 0.1925 | **0.1897** | 0.2097 | _0.2058_ | 0.2079 | **0.1976** |
| | 336 | 0.2324 | _0.2299_ | 0.2303 | **0.2286** | 0.2320 | **0.2291** | 0.2312 | _0.2294_ | 0.2808 | 0.2735 | _0.2653_ | **0.2623** | 0.2331 | _0.2308_ | 0.2316 | **0.2289** | 0.2482 | 0.2482 | _0.2473_ | **0.2422** |
| | 720 | 0.3062 | _0.2993_ | **0.2984** | 0.2984 | 0.3013 | **0.2924** | 0.2983 | _0.2955_ | 0.3457 | 0.3303 | **0.3263** | _0.3290_ | 0.3066 | _0.2996_ | 0.3003 | **0.2973** | 0.3283 | _0.3218_ | 0.3240 | **0.3192** |
| **Weather** | 96 | 0.1954 | _0.1803_ | 0.1812 | **0.1797** | 0.1855 | _0.1757_ | 0.1795 | **0.1678** | 0.1724 | _0.1662_ | 0.1662 | **0.1605** | 0.1957 | _0.1825_ | 0.1890 | **0.1728** | 0.1746 | 0.1700 | _0.1676_ | **0.1635** |
| | 192 | 0.2403 | _0.2265_ | 0.2308 | **0.2231** | 0.2309 | _0.2167_ | 0.2229 | **0.2114** | 0.2239 | _0.2129_ | 0.2165 | **0.2103** | 0.2406 | _0.2238_ | 0.2293 | **0.2171** | 0.2188 | 0.2119 | _0.2120_ | **0.2050** |
| | 336 | 0.2918 | _0.2726_ | 0.2776 | **0.2717** | 0.2842 | _0.2661_ | 0.2720 | **0.2621** | 0.2806 | _0.2629_ | 0.2672 | **0.2618** | 0.2921 | _0.2729_ | 0.2785 | **0.2665** | 0.2771 | _0.2659_ | 0.2703 | **0.2570** |
| | 720 | 0.3643 | _0.3509_ | 0.3557 | **0.3426** | 0.3599 | _0.3423_ | 0.3550 | **0.3376** | 0.3563 | _0.3418_ | 0.3470 | **0.3357** | 0.3645 | _0.3467_ | 0.3598 | **0.3401** | 0.3555 | _0.3356_ | 0.3529 | **0.3341** |
| **Exchange** | 96 | 0.0913 | **0.0875** | 0.0898 | _0.0875_ | 0.0828 | **0.0796** | 0.0825 | _0.0810_ | 0.0882 | _0.0873_ | 0.0881 | **0.0867** | 0.0814 | _0.0799_ | 0.0813 | **0.0793** | 0.0854 | _0.0816_ | 0.0850 | **0.0813** |
| | 192 | 0.1827 | **0.1732** | 0.1787 | _0.1758_ | 0.1734 | **0.1654** | 0.1729 | _0.1667_ | 0.1841 | _0.1800_ | 0.1820 | **0.1773** | 0.1727 | _0.1650_ | 0.1719 | **0.1648** | 0.1774 | **0.1666** | 0.1765 | _0.1692_ |
| | 336 | 0.3277 | _0.3052_ | 0.3154 | **0.2985** | 0.3241 | _0.3180_ | 0.3223 | **0.3166** | 0.3157 | _0.3014_ | 0.3105 | **0.2983** | 0.3225 | **0.2846** | 0.3058 | _0.2924_ | 0.3382 | _0.3048_ | 0.3188 | **0.3011** |

*Note.* FAC denotes our proposed method. For PatchTST, TAFAS† follows the official PatchTST-specific implementation, using single-channel output calibration and bypassing input calibration. FAC instead uses the same input and output FreqGCM design as for the other source forecasters. Thus, the two methods are not architecturally identical, although both are evaluated under the same matured-only supervision protocol. PETSA uses multivariate calibration on PatchTST to match its frequency-domain and distributional adaptation objectives over multivariate forecasts.

Table 1 reports the MSE results from one experimental run for each forecasting setting. To assess robustness to adaptation randomness, we repeat each matched setting with five adaptation seeds while fixing the source-forecaster checkpoint, with the resulting means and standard deviations reported in Appendix A.1.

Overall, FAC achieves the best or second-best performance in most settings and often outperforms TAFAS and PETSA under the same protocol using only matured ground truth. In addition, all three adaptation methods generally improve over the frozen source forecaster, indicating that matured ground truth can serve as an effective adaptation signal for TSF-TTA. Consistent with these observations, dataset-wise MSE mean±standard deviation summaries show that FAC achieves the lowest average MSE on five of the six datasets, while paired Wilcoxon tests indicate statistically significant improvements over the frozen source forecaster and PETSA on all six datasets and over TAFAS on four; full results are provided in Appendix A.1.

We further examine whether FAC remains effective on datasets with larger numbers of variables. Table 2 summarizes the additional results on Electricity and Traffic.

Table 2: Summary of additional results on Electricity and Traffic over the common comparison set, comprising all 12 Electricity settings and the nine Traffic settings with $H \in \{96, 192, 336\}$. The MSE columns report averages of the setting-level mean MSEs across the included model–horizon settings of each dataset. "Wins (S/T/P)" denotes the number of included settings in which FAC achieves lower MSE than the frozen source forecaster, TAFAS, and PETSA, respectively. Lower MSE is better.

| Dataset | # Settings | Source | TAFAS | PETSA | FAC | Wins (S/T/P) |
|---|---|---|---|---|---|---|
| Electricity | 12 | 0.2137 | 0.2104 | 0.2122 | **0.2078** | 10/12, 10/12, 12/12 |
| Traffic | 9 | 0.5196 | 0.5160 | 0.5158 | **0.5111** | 9/9, 9/9, 9/9 |

Across the 21 settings in Table 2, FAC achieves lower MSE than the frozen source forecaster and TAFAS in 19 settings each, and lower MSE than PETSA in all 21 settings. FAC also outperforms all three alternatives in all nine summarized Traffic settings. The three Traffic settings at $H = 720$ are reported separately to maintain a common comparison set across source forecasters, since TAFAS exceeds the available GPU memory for FreTS and iTransformer. At $H = 720$, FAC achieves the lowest available MSE for FreTS and iTransformer and ties PETSA for DLinear at the reported precision. Overall, these results show that FAC remains effective on datasets with larger numbers of variables and across the additional energy and transportation domains. All available setting-level results, including means and standard deviations over five adaptation seeds, are provided in Appendix A.2.

### 5.2.2 Parameter and Runtime Efficiency

Table 3: Trainable parameter counts of different test-time adapters. For TAFAS, we report both the standard count and the PatchTST-specific count, since TAFAS uses channel dimension 1 for PatchTST but the number of variables $C$ for other backbones. The smallest count in each row is highlighted in red bold, and the second smallest is underlined.

| Dataset | $H$ | TAFAS | TAFAS (PatchTST) | PETSA | FAC (Ours) |
|---|---|---|---|---|---|
| **ETT (h1/2, m1/2)** | 96 | 130,382 | 18,626 | 25,934 | **2,758** |
| | 192 | 324,590 | 46,370 | 38,894 | **4,102** |
| | 336 | 857,822 | 122,546 | 58,334 | **6,118** |
| | 720 | 3,699,038 | 528,434 | 110,174 | **11,494** |
| **Weather** | 96 | 391,146 | 18,626 | 71,658 | **8,274** |
| | 192 | 973,770 | 46,370 | 107,466 | **12,306** |
| | 336 | 2,573,466 | 122,546 | 161,178 | **18,354** |
| | 720 | 11,097,114 | 528,434 | 304,410 | **34,482** |
| **Exchange** | 96 | 149,008 | 18,626 | 29,200 | **3,152** |
| | 192 | 370,960 | 46,370 | 43,792 | **4,688** |
| | 336 | 980,368 | 122,546 | 65,680 | **6,992** |

Besides forecasting error, parameter efficiency is a central consideration when designing a test-time adapter. This consideration is particularly relevant in larger-scale forecasting settings, where memory constraints may limit the applicability of some adaptation methods, as observed in the Traffic experiments reported in Appendix A.2. A direct way to measure parameter efficiency is to compare the number of trainable parameters in the calibration modules. Table 3 reports the trainable parameter counts of different adapters under our experimental settings. FAC consistently requires the fewest trainable parameters across datasets and horizons.

Table 4 summarizes how the adapter size scales with the look-back length $L$, prediction horizon $H$, number of variables $C$, and PETSA rank $r$. TAFAS scales quadratically with the sequence lengths due to its dense temporal calibration, whereas PETSA achieves rank-dependent linear scaling through low-rank adapters. FAC instead uses an element-wise frequency-domain affine parameterization, yielding linear scaling in $C$, $L$, and $H$ without the rank factor.

Table 4: Parameter scaling of different test-time adapters.

| Adapter | Parameter scaling |
|---------|-------------------|
| **TAFAS** | $O\big(C(L^2 + H^2)\big)$ |
| **TAFAS (PatchTST)** | $O\big(L^2 + H^2\big)$ |
| **PETSA** | $O(Cr(L + H))$ |
| **FAC (Ours)** | $O(C(L + H))$ |

Trainable parameter count and wall-clock efficiency should be distinguished. Although FAC requires substantially fewer trainable parameters, this parameter efficiency does not always translate into lower wall-clock runtime under our current implementation. As detailed in Appendix A.6, FAC is not uniformly faster than TAFAS. The additional rFFT/iFFT operations and tensor transformations required by the current implementation likely contribute to the observed runtime overhead. This difference is particularly pronounced for PatchTST, where TAFAS uses its PatchTST-specific implementation with single-channel output calibration and bypasses input calibration, whereas FAC retains both input and output FreqGCMs.

### 5.3 Analysis

We further analyze two aspects of our design: whether POGT is necessary for effective adaptation, and how different adapters modify predictions in the frequency domain. The first analysis examines the role of supervision timing, while the second characterizes the structure of adaptation-induced corrections. Detailed ablations of the multiplicative and additive terms, complex phase modification, residual gate, and input- and output-side calibration are provided in Appendix A.5.

#### 5.3.1 Is POGT Necessary for Effective Adaptation?

We first compare the original mixed-supervision TAFAS protocol with a matured-only variant to assess the empirical contribution of POGT. Table 5 presents a representative subset of this comparison on ETTh1, while complete method-native comparisons for both TAFAS and PETSA across all 115 primary settings are provided in Appendix A.4. Across the complete comparison, original TAFAS achieves lower MSE in 77 settings, whereas matured-only TAFAS achieves lower MSE in 38. Similarly, original PETSA achieves lower MSE in 74 settings, whereas matured-only PETSA achieves lower MSE in 40, with one tie. These results indicate that adding timely POGT supervision to matured-ground-truth supervision is beneficial more often, but not universally; the matured-only variants remain competitive in many settings without this additional signal. Accordingly, matured-only supervision provides a temporally conservative reference condition for comparing adapter architectures without the additional POGT signal.

Table 5: Representative comparison between original TAFAS and its matured-only variant on ETTh1. Lower MSE is better; the better result in each pair is highlighted in bold red.

| $H$ | DLinear | | FreTS | | iTransformer | | OLS | | PatchTST | |
|-----|---------|---------|-------|---------|--------------|---------|-----|---------|----------|---------|
| | Orig. | Matured | Orig. | Matured | Orig. | Matured | Orig. | Matured | Orig. | Matured |
| 96 | 0.4618 | **0.4617** | 0.4394 | **0.4391** | **0.4398** | 0.4421 | **0.4419** | 0.4428 | **0.4261** | 0.4270 |
| 192 | 0.5117 | **0.5112** | **0.4940** | 0.4943 | **0.5017** | 0.5020 | **0.4924** | 0.4929 | **0.4809** | 0.4818 |
| 336 | 0.5604 | **0.5595** | **0.5475** | 0.5483 | 0.5643 | **0.5624** | **0.5415** | 0.5427 | **0.5501** | 0.5506 |
| 720 | **0.6820** | 0.6926 | 0.6872 | **0.6857** | **0.6591** | 0.6755 | **0.6654** | 0.6704 | 0.6815 | **0.6807** |

#### 5.3.2 Frequency-Domain Behavior of Prediction Corrections

We further examine how different adapters modify the forecast in the frequency domain. We first characterize the frequency-domain structure of the induced prediction corrections, without directly measuring forecasting error. Specifically, we compute the difference between predictions before and after adaptation, apply rFFT along the forecasting horizon, and average the correction magnitude over samples and channels.

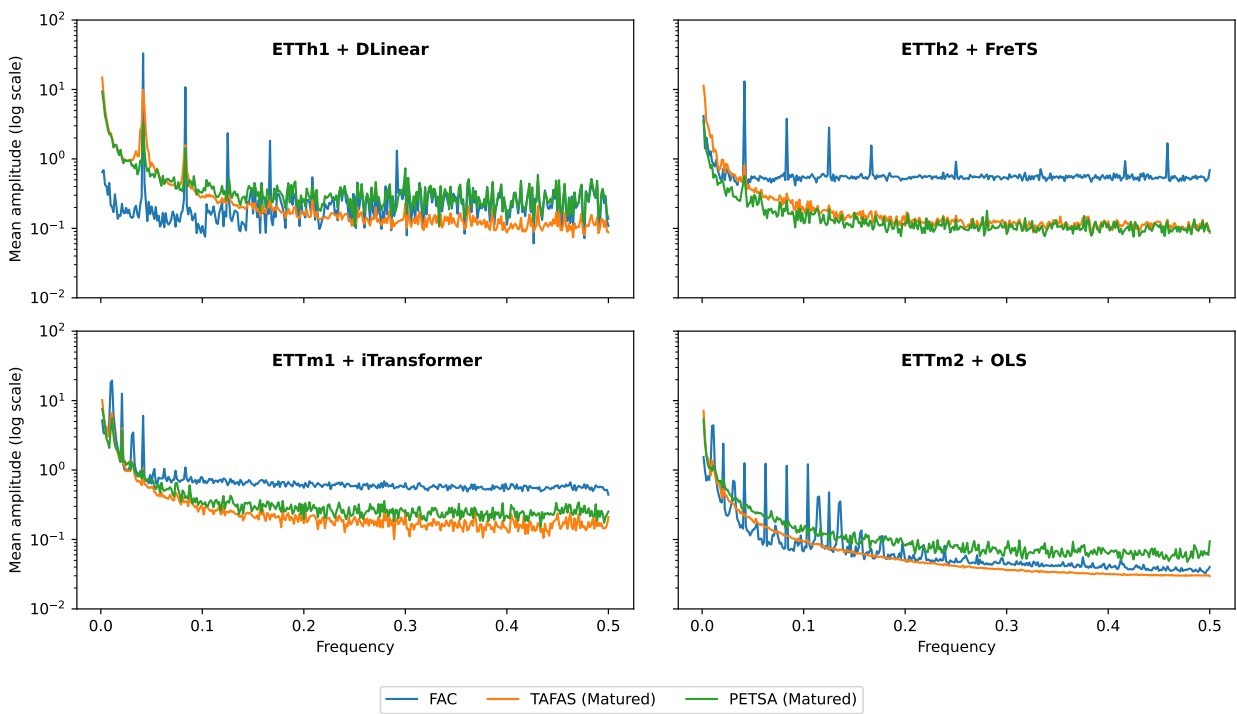

Figure 4: Frequency-domain magnitude of prediction corrections under the matured-only protocol for forecasting horizon $H = 720$. The plotted spectra are computed from the differences between predictions after and before adaptation. The x-axis denotes frequency, while the y-axis shows the correction magnitude on a logarithmic scale. The figure characterizes adaptation-induced corrections rather than forecasting error; a larger magnitude does not necessarily indicate a more accurate forecast.

Figure 4 illustrates the resulting correction spectra for representative dataset–forecaster pairs at forecasting horizon $H = 720$, with more comprehensive results across horizons deferred to Appendix A.7. As shown in the figure, TAFAS and PETSA often produce relatively smooth frequency-domain corrections, with magnitudes changing gradually across frequencies. In contrast, FAC exhibits more localized peaks across the frequency range. This behavior is consistent with the motivation of FAC: by directly applying an element-wise complex affine mask in the frequency domain, FAC allows individual Fourier components to be adjusted separately, which may naturally produce more frequency-localized corrections. Accordingly, these spectra provide a descriptive view of how the corrections are distributed across frequencies. The correction magnitude itself should not be interpreted as a direct measure of forecasting accuracy; a larger correction magnitude does not necessarily imply a more accurate forecast.

To examine whether these localized correction patterns coincide with forecasting improvements, we further measure residual-error reduction within low-, middle-, and high-frequency bands. We partition the one-sided rFFT frequency range into three contiguous equal-width bands over normalized frequency: $[0, 1/6)$, $[1/6, 1/3)$, and $[1/3, 1/2]$, respectively. Table 6 reports the resulting within-band reduction rates.

The DC component is retained in the low-frequency band so that the three bands cover the complete residual spectral energy. For a prediction $\hat{\mathbf{Y}}$, let $E_b(\hat{\mathbf{Y}})$ denote its residual spectral energy in frequency band $b$, computed from the difference between the prediction and the corresponding forecasting target. We define the within-band error-reduction rate as

$$R_b = 100 \times \frac{E_b(\hat{\mathbf{Y}}) - E_b(\hat{\mathbf{Y}}^{\mathrm{adp}})}{E_b(\hat{\mathbf{Y}})}. \tag{19}$$

Positive values indicate residual-error reduction within band $b$.

As shown in Table 6, FAC achieves the largest total residual-error reduction in all four representative settings. It also yields substantially larger middle- and high-frequency reduction rates in the first three settings,

Table 6: Within-band residual-error reduction rates (%) for representative settings at $H = 720$. The Total column reports the reduction in residual energy over the complete frequency range. Higher values indicate larger error reduction. Within each setting, the best result in each column is highlighted in bold.

| Setting | Method | Low | Middle | High | Total |
|---|---|---|---|---|---|
| ETTh1 + DLinear | TAFAS | 2.930 | 0.110 | 0.170 | 2.692 |
| | PETSA | 2.730 | 0.220 | 0.400 | 2.524 |
| | FAC | **3.300** | **2.570** | **0.780** | **3.183** |
| ETTh2 + FreTS | TAFAS | 7.890 | 0.120 | 0.070 | 7.392 |
| | PETSA | 3.900 | 0.080 | 0.050 | 3.657 |
| | FAC | **8.260** | **3.640** | **5.330** | **8.015** |
| ETTm1 + iTransformer | TAFAS | 7.620 | 0.460 | 0.610 | 7.350 |
| | PETSA | 6.970 | 0.280 | 0.380 | 6.709 |
| | FAC | **9.590** | **2.930** | **4.770** | **9.360** |
| ETTm2 + OLS | TAFAS | 2.370 | **0.045** | **0.037** | 2.290 |
| | PETSA | 2.130 | 0.027 | 0.005 | 2.058 |
| | FAC | **3.140** | 0.017 | 0.017 | **3.031** |

consistent with the localized correction spectra in Figure 4. For ETTm2 with OLS, however, the middle- and high-frequency reductions remain small for all methods, and the overall improvement is driven primarily by the low-frequency band. Across these cases, FAC's localized correction patterns coincide with residual-error reduction in the corresponding frequency bands, although the relationship depends on the spectral structure of each forecasting setting. The corresponding additive decomposition of total residual-error reduction across frequency bands is provided in Appendix A.8.

To further isolate the effect of frequency-domain parameterization, we compare FAC with a comparably sized temporal-domain counterpart that preserves the same element-wise affine and gated residual structure. The temporal counterpart improves over the source forecaster in 113 of the 115 primary settings, while FAC achieves lower MSE in all 115 matched comparisons; complete results are provided in Appendix A.9.

## 6 Conclusions, Limitations, and Future Work

**Conclusions.** We re-examine the adaptation protocols employed by existing TSF-TTA methods. Specifically, we revisit the mixed-supervision protocol that incorporates both POGT and matured ground truth, as well as the streaming adaptation protocol based on revealed targets, and discuss their potential protocol-level limitations. To make the supervision criterion explicit, we propose an adaptation protocol that uses only matured ground truth. We further diagnose existing adapters in the frequency domain and propose FAC, a lightweight adapter that directly parameterizes prediction corrections in the frequency domain. Experimental results demonstrate that FAC achieves competitive and consistent forecasting performance across diverse settings while requiring substantially fewer trainable parameters.

**Limitations.** As discussed in Section 5.2.2 and detailed in Appendix A.6, FAC is not uniformly faster than TAFAS under our current implementation. The additional rFFT/iFFT operations and tensor transformations required by FAC likely contribute to the observed runtime overhead in some settings.

Although the additional experiments on Electricity and Traffic broaden the coverage of variable counts and application domains, our evaluation remains within the standard multivariate LTSF setting and does not cover the broader range of domains, frequencies, and forecasting tasks represented in benchmarks such as GIFT-Eval (Aksu et al., 2024). Moreover, it does not systematically characterize performance across controlled shift types and severities. Such controlled analyses remain important directions for future work.

**Future Work.** Future work could explore more flexible matured-supervision implementations that do not require waiting for the entire mini-batch to mature before using individual samples whose full target horizons have already matured. Another direction is to use multiple available matured mini-batches, as allowed by Eq. 15, instead of only the most recent one. When multiple matured mini-batches are used, cached frequency-domain representations of historical matured mini-batches may help reduce repeated FFT computations and enhance the runtime efficiency of frequency-aware adaptation. More generally, reducing redundant tensor transformations and other implementation overhead may further improve the wall-clock efficiency of FAC.

Beyond these protocol and efficiency directions, the proposed protocol and FAC could be evaluated across a broader collection of datasets, forecasting settings, and adaptation objectives.

The present work instantiates FAC for deterministic point forecasting with an MSE adaptation loss; future work could explore other point-forecasting losses such as MAE or stabilized MAPE, as well as extensions to probabilistic or quantile forecasting, which would require additional design for distributional or multi-quantile outputs. Another useful direction is to characterize more systematically how FAC's effectiveness and learned corrections vary with the temporal and spectral structure of the data. Together, these directions could make adaptation based on matured ground truth more broadly applicable, timely, and efficient while retaining the matured-only supervision condition.

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

# A Additional Experimental Results

## A.1 Statistical Significance Analysis

We evaluate the consistency of the results in Table 1 using matched forecasting settings. For every dataset, source forecaster, forecasting horizon, and adaptation method, we repeat test-time adaptation with five random seeds while keeping the source-forecaster checkpoint fixed. Each setting is represented by its five-seed mean for statistical testing, and all comparisons are computed using full-precision values before rounding.

Tables A.1–A.3 report the complete setting-level results. Source gives the fixed source-forecaster MSE, while TAFAS, PETSA, and FAC are reported as MSE mean±standard deviation over five adaptation seeds. The lowest five-seed mean MSE in each setting is highlighted in bold.

Table A.1: Setting-level MSE results for DLinear and FreTS.

| | | | **DLinear** | | |
|---|---|---|---|---|---|
| Dataset | $H$ | Source | TAFAS | PETSA | FAC |
| ETTh1 | 96 | 0.4695 | 0.4617 ± 0.0000 | 0.4613 ± 0.0002 | **0.4554 ± 0.0000** |
| | 192 | 0.5213 | 0.5112 ± 0.0000 | 0.5121 ± 0.0003 | **0.5089 ± 0.0000** |
| | 336 | 0.5659 | 0.5595 ± 0.0000 | 0.5620 ± 0.0004 | **0.5582 ± 0.0000** |
| | 720 | 0.7117 | 0.6926 ± 0.0000 | 0.6932 ± 0.0005 | **0.6891 ± 0.0000** |
| ETTh2 | 96 | 0.2323 | 0.2308 ± 0.0000 | 0.2312 ± 0.0001 | **0.2288 ± 0.0000** |
| | 192 | 0.2862 | 0.2830 ± 0.0000 | 0.2820 ± 0.0001 | **0.2819 ± 0.0000** |
| | 336 | 0.3252 | 0.3175 ± 0.0000 | 0.3190 ± 0.0000 | **0.3144 ± 0.0000** |
| | 720 | 0.4087 | **0.3825 ± 0.0000** | 0.3883 ± 0.0002 | **0.3825 ± 0.0000** |
| ETTm1 | 96 | 0.3715 | 0.3504 ± 0.0000 | 0.3544 ± 0.0003 | **0.3465 ± 0.0000** |
| | 192 | 0.4438 | 0.4165 ± 0.0000 | 0.4197 ± 0.0003 | **0.4139 ± 0.0000** |
| | 336 | 0.5183 | 0.4801 ± 0.0000 | 0.4831 ± 0.0000 | **0.4791 ± 0.0000** |
| | 720 | 0.5929 | 0.5521 ± 0.0000 | 0.5598 ± 0.0005 | **0.5494 ± 0.0000** |
| ETTm2 | 96 | 0.1598 | 0.1581 ± 0.0000 | 0.1583 ± 0.0001 | **0.1560 ± 0.0000** |
| | 192 | 0.1930 | 0.1917 ± 0.0000 | 0.1919 ± 0.0001 | **0.1894 ± 0.0000** |
| | 336 | 0.2324 | 0.2299 ± 0.0000 | 0.2304 ± 0.0001 | **0.2286 ± 0.0000** |
| | 720 | 0.3062 | 0.2993 ± 0.0000 | 0.2989 ± 0.0005 | **0.2984 ± 0.0000** |
| Weather | 96 | 0.1954 | 0.1803 ± 0.0000 | 0.1814 ± 0.0005 | **0.1797 ± 0.0000** |
| | 192 | 0.2403 | 0.2265 ± 0.0000 | 0.2308 ± 0.0003 | **0.2231 ± 0.0000** |
| | 336 | 0.2918 | 0.2726 ± 0.0000 | 0.2776 ± 0.0001 | **0.2717 ± 0.0000** |
| | 720 | 0.3643 | 0.3509 ± 0.0000 | 0.3559 ± 0.0004 | **0.3426 ± 0.0000** |
| Exchange | 96 | 0.0913 | **0.0875 ± 0.0000** | 0.0899 ± 0.0001 | **0.0875 ± 0.0000** |
| | 192 | 0.1827 | **0.1732 ± 0.0000** | 0.1786 ± 0.0003 | 0.1758 ± 0.0000 |
| | 336 | 0.3277 | 0.3052 ± 0.0000 | 0.3148 ± 0.0005 | **0.2985 ± 0.0000** |

| | | | **FreTS** | | |
|---|---|---|---|---|---|
| Dataset | $H$ | Source | TAFAS | PETSA | FAC |
| ETTh1 | 96 | 0.4461 | 0.4391 ± 0.0000 | 0.4380 ± 0.0004 | **0.4357 ± 0.0000** |
| | 192 | 0.5022 | 0.4943 ± 0.0000 | 0.4964 ± 0.0003 | **0.4934 ± 0.0000** |
| | 336 | 0.5544 | **0.5483 ± 0.0000** | 0.5529 ± 0.0002 | 0.5498 ± 0.0000 |
| | 720 | 0.7181 | **0.6857 ± 0.0000** | 0.6906 ± 0.0010 | 0.7067 ± 0.0000 |
| ETTh2 | 96 | 0.2385 | 0.2363 ± 0.0000 | 0.2365 ± 0.0001 | **0.2338 ± 0.0000** |
| | 192 | 0.2867 | 0.2830 ± 0.0000 | 0.2845 ± 0.0001 | **0.2814 ± 0.0000** |
| | 336 | 0.3318 | 0.3195 ± 0.0000 | 0.3249 ± 0.0002 | **0.3172 ± 0.0000** |
| | 720 | 0.4118 | 0.3814 ± 0.0000 | 0.3967 ± 0.0006 | **0.3788 ± 0.0000** |
| ETTm1 | 96 | 0.3676 | 0.3559 ± 0.0000 | 0.3566 ± 0.0005 | **0.3517 ± 0.0000** |
| | 192 | 0.4325 | 0.4191 ± 0.0000 | 0.4192 ± 0.0003 | **0.4183 ± 0.0000** |
| | 336 | 0.5006 | 0.4810 ± 0.0000 | **0.4794 ± 0.0002** | 0.4836 ± 0.0000 |
| | 720 | 0.5703 | **0.5472 ± 0.0000** | 0.5523 ± 0.0002 | 0.5533 ± 0.0000 |
| ETTm2 | 96 | 0.1582 | 0.1566 ± 0.0000 | 0.1568 ± 0.0001 | **0.1563 ± 0.0000** |
| | 192 | 0.1923 | **0.1901 ± 0.0000** | 0.1908 ± 0.0001 | **0.1901 ± 0.0000** |
| | 336 | 0.2320 | **0.2291 ± 0.0000** | 0.2312 ± 0.0001 | 0.2294 ± 0.0000 |
| | 720 | 0.3013 | **0.2924 ± 0.0000** | 0.2983 ± 0.0002 | 0.2955 ± 0.0000 |
| Weather | 96 | 0.1855 | 0.1757 ± 0.0000 | 0.1791 ± 0.0002 | **0.1678 ± 0.0000** |
| | 192 | 0.2309 | 0.2167 ± 0.0000 | 0.2232 ± 0.0003 | **0.2114 ± 0.0000** |
| | 336 | 0.2842 | 0.2661 ± 0.0000 | 0.2720 ± 0.0002 | **0.2621 ± 0.0000** |
| | 720 | 0.3599 | 0.3423 ± 0.0000 | 0.3550 ± 0.0003 | **0.3376 ± 0.0000** |
| Exchange | 96 | 0.0828 | **0.0796 ± 0.0000** | 0.0819 ± 0.0001 | 0.0810 ± 0.0000 |
| | 192 | 0.1734 | **0.1654 ± 0.0000** | 0.1716 ± 0.0001 | 0.1667 ± 0.0000 |
| | 336 | 0.3241 | 0.3180 ± 0.0000 | 0.3171 ± 0.0007 | **0.3166 ± 0.0000** |

Table A.2: Setting-level MSE results for iTransformer and OLS.

| iTransformer | | | | | |
|---|---|---|---|---|---|
| Dataset | $H$ | Source | TAFAS | PETSA | FAC |
| ETTh1 | 96 | 0.4482 | $0.4423 \pm 0.0001$ | $\mathbf{0.4397 \pm 0.0004}$ | $0.4401 \pm 0.0002$ |
| | 192 | 0.5091 | $0.5020 \pm 0.0001$ | $0.5011 \pm 0.0003$ | $\mathbf{0.5010 \pm 0.0002}$ |
| | 336 | 0.5662 | $0.5625 \pm 0.0001$ | $0.5642 \pm 0.0003$ | $\mathbf{0.5589 \pm 0.0002}$ |
| | 720 | 0.7007 | $0.6753 \pm 0.0003$ | $0.6768 \pm 0.0008$ | $\mathbf{0.6684 \pm 0.0004}$ |
| ETTh2 | 96 | 0.2642 | $0.2620 \pm 0.0001$ | $0.2599 \pm 0.0003$ | $\mathbf{0.2557 \pm 0.0006}$ |
| | 192 | 0.3081 | $0.3006 \pm 0.0002$ | $0.3006 \pm 0.0001$ | $\mathbf{0.2972 \pm 0.0003}$ |
| | 336 | 0.3402 | $\mathbf{0.3289 \pm 0.0001}$ | $0.3313 \pm 0.0001$ | $0.3290 \pm 0.0012$ |
| | 720 | 0.4261 | $\mathbf{0.3959 \pm 0.0001}$ | $0.4051 \pm 0.0004$ | $0.3999 \pm 0.0017$ |
| ETTm1 | 96 | 0.3823 | $0.3681 \pm 0.0001$ | $0.3697 \pm 0.0004$ | $\mathbf{0.3456 \pm 0.0004}$ |
| | 192 | 0.4407 | $0.4191 \pm 0.0001$ | $0.4189 \pm 0.0003$ | $\mathbf{0.4089 \pm 0.0001}$ |
| | 336 | 0.5077 | $0.4798 \pm 0.0002$ | $0.4789 \pm 0.0004$ | $\mathbf{0.4780 \pm 0.0002}$ |
| | 720 | 0.6021 | $0.5579 \pm 0.0001$ | $0.5615 \pm 0.0005$ | $\mathbf{0.5459 \pm 0.0001}$ |
| ETTm2 | 96 | 0.1647 | $0.1640 \pm 0.0000$ | $0.1640 \pm 0.0001$ | $\mathbf{0.1608 \pm 0.0003}$ |
| | 192 | 0.2189 | $0.2164 \pm 0.0001$ | $0.2155 \pm 0.0003$ | $\mathbf{0.2065 \pm 0.0003}$ |
| | 336 | 0.2808 | $0.2734 \pm 0.0002$ | $0.2669 \pm 0.0017$ | $\mathbf{0.2619 \pm 0.0010}$ |
| | 720 | 0.3457 | $0.3300 \pm 0.0001$ | $\mathbf{0.3256 \pm 0.0004}$ | $0.3291 \pm 0.0010$ |
| Weather | 96 | 0.1724 | $0.1662 \pm 0.0001$ | $0.1661 \pm 0.0001$ | $\mathbf{0.1607 \pm 0.0002}$ |
| | 192 | 0.2239 | $0.2130 \pm 0.0001$ | $0.2163 \pm 0.0004$ | $\mathbf{0.2102 \pm 0.0001}$ |
| | 336 | 0.2806 | $0.2627 \pm 0.0001$ | $0.2668 \pm 0.0002$ | $\mathbf{0.2617 \pm 0.0002}$ |
| | 720 | 0.3563 | $0.3419 \pm 0.0001$ | $0.3472 \pm 0.0002$ | $\mathbf{0.3357 \pm 0.0003}$ |
| Exchange | 96 | 0.0882 | $0.0873 \pm 0.0001$ | $0.0881 \pm 0.0001$ | $\mathbf{0.0869 \pm 0.0004}$ |
| | 192 | 0.1841 | $0.1801 \pm 0.0001$ | $0.1820 \pm 0.0001$ | $\mathbf{0.1776 \pm 0.0001}$ |
| | 336 | 0.3157 | $0.3009 \pm 0.0002$ | $0.3108 \pm 0.0003$ | $\mathbf{0.2965 \pm 0.0029}$ |

| OLS | | | | | |
|---|---|---|---|---|---|
| Dataset | $H$ | Source | TAFAS | PETSA | FAC |
| ETTh1 | 96 | 0.4510 | $0.4428 \pm 0.0000$ | $0.4452 \pm 0.0003$ | $\mathbf{0.4382 \pm 0.0000}$ |
| | 192 | 0.5045 | $0.4929 \pm 0.0000$ | $0.5005 \pm 0.0003$ | $\mathbf{0.4921 \pm 0.0000}$ |
| | 336 | 0.5510 | $0.5427 \pm 0.0000$ | $0.5488 \pm 0.0002$ | $\mathbf{0.5422 \pm 0.0000}$ |
| | 720 | 0.6996 | $\mathbf{0.6704 \pm 0.0000}$ | $0.6805 \pm 0.0002$ | $0.6780 \pm 0.0000$ |
| ETTh2 | 96 | 0.2306 | $0.2289 \pm 0.0000$ | $0.2294 \pm 0.0001$ | $\mathbf{0.2274 \pm 0.0000}$ |
| | 192 | 0.2838 | $0.2821 \pm 0.0000$ | $0.2829 \pm 0.0000$ | $\mathbf{0.2796 \pm 0.0000}$ |
| | 336 | 0.3257 | $0.3189 \pm 0.0000$ | $0.3240 \pm 0.0001$ | $\mathbf{0.3137 \pm 0.0000}$ |
| | 720 | 0.4161 | $0.3859 \pm 0.0000$ | $0.3928 \pm 0.0002$ | $\mathbf{0.3854 \pm 0.0000}$ |
| ETTm1 | 96 | 0.3709 | $0.3542 \pm 0.0000$ | $0.3599 \pm 0.0008$ | $\mathbf{0.3462 \pm 0.0000}$ |
| | 192 | 0.4437 | $0.4184 \pm 0.0000$ | $0.4255 \pm 0.0006$ | $\mathbf{0.4140 \pm 0.0000}$ |
| | 336 | 0.5179 | $0.4811 \pm 0.0000$ | $0.4893 \pm 0.0002$ | $\mathbf{0.4794 \pm 0.0000}$ |
| | 720 | 0.5923 | $0.5515 \pm 0.0000$ | $0.5595 \pm 0.0001$ | $\mathbf{0.5497 \pm 0.0000}$ |
| ETTm2 | 96 | 0.1602 | $0.1594 \pm 0.0000$ | $0.1595 \pm 0.0000$ | $\mathbf{0.1563 \pm 0.0000}$ |
| | 192 | 0.1935 | $0.1924 \pm 0.0000$ | $0.1925 \pm 0.0001$ | $\mathbf{0.1897 \pm 0.0000}$ |
| | 336 | 0.2331 | $0.2308 \pm 0.0000$ | $0.2316 \pm 0.0001$ | $\mathbf{0.2289 \pm 0.0000}$ |
| | 720 | 0.3066 | $0.2996 \pm 0.0000$ | $0.3002 \pm 0.0001$ | $\mathbf{0.2973 \pm 0.0000}$ |
| Weather | 96 | 0.1957 | $0.1825 \pm 0.0000$ | $0.1894 \pm 0.0004$ | $\mathbf{0.1728 \pm 0.0000}$ |
| | 192 | 0.2406 | $0.2238 \pm 0.0000$ | $0.2296 \pm 0.0002$ | $\mathbf{0.2171 \pm 0.0000}$ |
| | 336 | 0.2921 | $0.2729 \pm 0.0000$ | $0.2785 \pm 0.0001$ | $\mathbf{0.2665 \pm 0.0000}$ |
| | 720 | 0.3645 | $0.3467 \pm 0.0000$ | $0.3597 \pm 0.0002$ | $\mathbf{0.3401 \pm 0.0000}$ |
| Exchange | 96 | 0.0814 | $0.0799 \pm 0.0000$ | $0.0811 \pm 0.0001$ | $\mathbf{0.0793 \pm 0.0000}$ |
| | 192 | 0.1727 | $0.1650 \pm 0.0000$ | $0.1708 \pm 0.0001$ | $\mathbf{0.1648 \pm 0.0000}$ |
| | 336 | 0.3225 | $\mathbf{0.2846 \pm 0.0000}$ | $0.2967 \pm 0.0006$ | $0.2924 \pm 0.0000$ |

Table A.3: Setting-level MSE results for PatchTST.

| Dataset | $H$ | Source | TAFAS | PETSA | FAC |
|---------|-----|--------|-------|-------|-----|
| | | | **PatchTST** | | |
| ETTh1 | 96 | 0.4324 | $0.4270 \pm 0.0000$ | $0.4253 \pm 0.0002$ | $\mathbf{0.4242 \pm 0.0000}$ |
| | 192 | 0.4900 | $0.4818 \pm 0.0000$ | $0.4815 \pm 0.0001$ | $\mathbf{0.4797 \pm 0.0001}$ |
| | 336 | 0.5608 | $\mathbf{0.5506 \pm 0.0000}$ | $0.5569 \pm 0.0004$ | $0.5525 \pm 0.0001$ |
| | 720 | 0.7112 | $\mathbf{0.6808 \pm 0.0001}$ | $0.7017 \pm 0.0013$ | $0.7092 \pm 0.0003$ |
| ETTh2 | 96 | 0.2385 | $0.2380 \pm 0.0000$ | $0.2383 \pm 0.0000$ | $\mathbf{0.2364 \pm 0.0001}$ |
| | 192 | 0.2840 | $0.2752 \pm 0.0000$ | $0.2807 \pm 0.0002$ | $\mathbf{0.2747 \pm 0.0001}$ |
| | 336 | 0.3233 | $0.3107 \pm 0.0001$ | $0.3188 \pm 0.0004$ | $\mathbf{0.3080 \pm 0.0001}$ |
| | 720 | 0.4292 | $0.4065 \pm 0.0000$ | $0.4202 \pm 0.0004$ | $\mathbf{0.4004 \pm 0.0002}$ |
| ETTm1 | 96 | 0.4041 | $0.3825 \pm 0.0001$ | $0.3819 \pm 0.0010$ | $\mathbf{0.3528 \pm 0.0001}$ |
| | 192 | 0.4515 | $0.4371 \pm 0.0001$ | $0.4408 \pm 0.0005$ | $\mathbf{0.4214 \pm 0.0001}$ |
| | 336 | 0.5044 | $0.4863 \pm 0.0000$ | $0.4970 \pm 0.0006$ | $\mathbf{0.4786 \pm 0.0001}$ |
| | 720 | 0.5608 | $0.5414 \pm 0.0001$ | $0.5478 \pm 0.0010$ | $\mathbf{0.5399 \pm 0.0001}$ |
| ETTm2 | 96 | 0.1581 | $0.1569 \pm 0.0000$ | $0.1568 \pm 0.0001$ | $\mathbf{0.1558 \pm 0.0001}$ |
| | 192 | 0.2097 | $0.2058 \pm 0.0001$ | $0.2077 \pm 0.0001$ | $\mathbf{0.1977 \pm 0.0001}$ |
| | 336 | 0.2482 | $0.2482 \pm 0.0000$ | $0.2473 \pm 0.0000$ | $\mathbf{0.2423 \pm 0.0001}$ |
| | 720 | 0.3283 | $0.3218 \pm 0.0000$ | $0.3240 \pm 0.0003$ | $\mathbf{0.3193 \pm 0.0004}$ |
| Weather | 96 | 0.1746 | $0.1701 \pm 0.0002$ | $0.1676 \pm 0.0004$ | $\mathbf{0.1635 \pm 0.0000}$ |
| | 192 | 0.2188 | $0.2118 \pm 0.0001$ | $0.2116 \pm 0.0002$ | $\mathbf{0.2051 \pm 0.0001}$ |
| | 336 | 0.2771 | $0.2659 \pm 0.0001$ | $0.2704 \pm 0.0005$ | $\mathbf{0.2571 \pm 0.0001}$ |
| | 720 | 0.3555 | $0.3354 \pm 0.0005$ | $0.3523 \pm 0.0015$ | $\mathbf{0.3338 \pm 0.0002}$ |
| Exchange | 96 | 0.0854 | $0.0816 \pm 0.0000$ | $0.0841 \pm 0.0001$ | $\mathbf{0.0813 \pm 0.0001}$ |
| | 192 | 0.1774 | $\mathbf{0.1666 \pm 0.0000}$ | $0.1739 \pm 0.0003$ | $0.1694 \pm 0.0001$ |
| | 336 | 0.3382 | $0.3052 \pm 0.0004$ | $0.3071 \pm 0.0008$ | $\mathbf{0.3025 \pm 0.0036}$ |

Across the five adaptation seeds, the run-to-run variation is small in most settings, supporting the stability of the comparisons across adaptation initializations. We next summarize performance within each dataset and evaluate paired significance across matched settings.

Table A.4: Dataset-wise MSE mean±standard deviation across matched source-forecaster and forecasting-horizon settings. Each ETT dataset and Weather contain 20 settings, while Exchange contains 15. The standard deviations characterize variation across forecasting settings. The lowest mean MSE in each row is highlighted in bold.

| Dataset | # Settings | Source | TAFAS | PETSA | FAC |
|---------|-----------|--------|-------|-------|-----|
| ETTh1 | 20 | $0.5557 \pm 0.0993$ | $\mathbf{0.5432 \pm 0.0914}$ | $0.5464 \pm 0.0944$ | $0.5441 \pm 0.0966$ |
| ETTh2 | 20 | $0.3196 \pm 0.0674$ | $0.3084 \pm 0.0574$ | $0.3124 \pm 0.0613$ | $\mathbf{0.3063 \pm 0.0579}$ |
| ETTm1 | 20 | $0.4788 \pm 0.0790$ | $0.4540 \pm 0.0719$ | $0.4578 \pm 0.0734$ | $\mathbf{0.4478 \pm 0.0760}$ |
| ETTm2 | 20 | $0.2312 \pm 0.0615$ | $0.2273 \pm 0.0583$ | $0.2274 \pm 0.0581$ | $\mathbf{0.2245 \pm 0.0585}$ |
| Weather | 20 | $0.2652 \pm 0.0674$ | $0.2512 \pm 0.0645$ | $0.2565 \pm 0.0678$ | $\mathbf{0.2460 \pm 0.0647}$ |
| Exchange | 15 | $0.1965 \pm 0.1024$ | $0.1853 \pm 0.0938$ | $0.1899 \pm 0.0955$ | $\mathbf{0.1851 \pm 0.0930}$ |

FAC achieves the lowest average MSE on five of the six datasets and remains close to the best result on ETTh1.

At the dataset level, paired Wilcoxon tests indicate that FAC yields statistically significant reductions in MSE relative to the frozen source forecaster and PETSA on all six primary datasets. Relative to TAFAS, the reduction is statistically significant on ETTh2, ETTm1, ETTm2, and Weather, but not on ETTh1 or Exchange.

Table A.5: Dataset-wise one-sided paired Wilcoxon signed-rank tests over matched forecasting settings. Each entry reports wins/ties/losses from the perspective of FAC, followed by the corresponding $p$-value. Statistically significant results at $p < 0.05$ are highlighted in bold.

| Dataset | FAC vs. Source | FAC vs. TAFAS | FAC vs. PETSA |
|---|---|---|---|
| ETTh1 | **20/0/0, $p = 9.54{\times}10^{-7}$** | 15/0/5, $p = 1.01{\times}10^{-1}$ | **17/0/3, $p = 6.04{\times}10^{-3}$** |
| ETTh2 | **20/0/0, $p = 9.54{\times}10^{-7}$** | **17/1/2, $p = 8.47{\times}10^{-4}$** | **20/0/0, $p = 9.54{\times}10^{-7}$** |
| ETTm1 | **20/0/0, $p = 9.54{\times}10^{-7}$** | **18/0/2, $p = 3.54{\times}10^{-4}$** | **18/0/2, $p = 2.38{\times}10^{-5}$** |
| ETTm2 | **20/0/0, $p = 9.54{\times}10^{-7}$** | **17/1/2, $p = 6.41{\times}10^{-4}$** | **19/0/1, $p = 1.31{\times}10^{-4}$** |
| Weather | **20/0/0, $p = 9.54{\times}10^{-7}$** | **20/0/0, $p = 9.54{\times}10^{-7}$** | **20/0/0, $p = 9.54{\times}10^{-7}$** |
| Exchange | **15/0/0, $p = 3.05{\times}10^{-5}$** | 9/1/5, $p = 3.19{\times}10^{-1}$ | **15/0/0, $p = 3.05{\times}10^{-5}$** |

Table A.6: Aggregate one-sided paired Wilcoxon signed-rank tests across all 115 matched forecasting settings. Wins/ties/losses are reported from the perspective of FAC. This analysis provides complementary aggregate evidence to the dataset-wise tests in Table A.5.

| Comparison | Wins/Ties/Losses | Wilcoxon $p$-value |
|---|---|---|
| FAC vs. Source | 115/0/0 | $6.56{\times}10^{-21}$ |
| FAC vs. TAFAS | 96/3/16 | $3.50{\times}10^{-11}$ |
| FAC vs. PETSA | 109/0/6 | $9.43{\times}10^{-18}$ |

The aggregate results lead to the same conclusion as the dataset-wise analysis. FAC improves over the frozen source forecaster in all 115 matched settings and outperforms PETSA in 109 of 115 settings. Against TAFAS, FAC records 96 wins, 3 ties, and 16 losses. All three aggregate paired Wilcoxon tests are statistically significant, providing strong evidence that the overall improvements are systematic rather than being driven by a small number of forecasting settings.

## A.2 Additional Results on Datasets with Larger Numbers of Variables

We report setting-level results on Electricity and Traffic, which broaden the range of variable counts and application domains covered by our primary evaluation. We evaluate DLinear, FreTS, and iTransformer with $H \in \{96, 192, 336, 720\}$ on both datasets. The source-forecaster checkpoint is fixed for each setting, while each completed adaptation method is repeated with five adaptation seeds.

Table A.7: Setting-level MSE results on Electricity and Traffic. Source denotes the fixed source-forecaster checkpoint. When available, TAFAS, PETSA, and FAC are reported as mean ± standard deviation over five adaptation seeds. Lower MSE is better. The best available result in each row is highlighted in bold, and the second-best available result is underlined. Ties at the reported precision are jointly highlighted in bold.

| (a) Electricity | | | | | |
|---|---|---|---|---|---|
| Model | $H$ | Source | TAFAS | PETSA | FAC |
| DLinear | 96 | 0.2235 | 0.2171±0.0000 | 0.2188±0.0001 | **0.2096±0.0000** |
| | 192 | 0.2242 | 0.2183±0.0000 | 0.2201±0.0001 | **0.2104±0.0000** |
| | 336 | 0.2383 | 0.2329±0.0000 | 0.2357±0.0001 | **0.2260±0.0000** |
| | 720 | 0.2792 | 0.2795±0.0000 | 0.2843±0.0001 | **0.2746±0.0000** |
| FreTS | 96 | 0.1806 | **0.1738±0.0000** | 0.1763±0.0001 | 0.1758±0.0000 |
| | 192 | 0.1880 | 0.1820±0.0000 | 0.1844±0.0001 | **0.1819±0.0000** |
| | 336 | 0.2056 | 0.1999±0.0000 | 0.2028±0.0001 | **0.1992±0.0000** |
| | 720 | **0.2482** | 0.2513±0.0000 | 0.2513±0.0002 | 0.2491±0.0000 |
| iTransformer | 96 | 0.1662 | 0.1630±0.0001 | 0.1640±0.0001 | **0.1608±0.0000** |
| | 192 | 0.1791 | **0.1760±0.0000** | 0.1771±0.0000 | 0.1763±0.0000 |
| | 336 | 0.1960 | 0.1936±0.0000 | 0.1948±0.0001 | **0.1932±0.0001** |
| | 720 | **0.2352** | 0.2376±0.0000 | 0.2372±0.0002 | 0.2366±0.0001 |

| (b) Traffic | | | | | |
|---|---|---|---|---|---|
| Model | $H$ | Source | TAFAS | PETSA | FAC |
| DLinear | 96 | 0.6467 | 0.6423±0.0000 | 0.6402±0.0001 | **0.6289±0.0000** |
| | 192 | 0.5996 | 0.5957±0.0000 | 0.5921±0.0001 | **0.5871±0.0000** |
| | 336 | 0.6071 | 0.6032±0.0000 | 0.5985±0.0003 | **0.5983±0.0000** |
| | 720 | 0.6457 | 0.6438±0.0000 | **0.6433±0.0001** | **0.6433±0.0000** |
| FreTS | 96 | 0.4854 | 0.4797±0.0000 | 0.4823±0.0001 | **0.4764±0.0000** |
| | 192 | 0.5065 | 0.5015±0.0000 | 0.5033±0.0002 | **0.4979±0.0000** |
| | 336 | 0.5206 | 0.5162±0.0000 | 0.5187±0.0002 | **0.5160±0.0000** |
| | 720 | 0.5575 | – | 0.5606±0.0003 | **0.5557±0.0000** |
| iTransformer | 96 | 0.4164 | 0.4150±0.0002 | 0.4145±0.0001 | **0.4100±0.0001** |
| | 192 | 0.4392 | 0.4372±0.0000 | 0.4381±0.0003 | **0.4344±0.0001** |
| | 336 | 0.4549 | 0.4534±0.0001 | 0.4541±0.0003 | **0.4514±0.0000** |
| | 720 | 0.4965 | – | 0.4921±0.0006 | **0.4862±0.0000** |

*Note.* For Traffic at $H = 720$, TAFAS exceeded the available 24 GB GPU memory for FreTS and iTransformer under the same experimental configuration. The corresponding entries are therefore marked with dashes.

Because two TAFAS entries at Traffic $H = 720$ are unavailable, the aggregate summary in Table 2 and the corresponding win counts in the main text are computed over the 21 settings. Among these settings, FAC achieves the lowest MSE in 17 cases, while its absolute gap from the best result is at most 0.0020 in the remaining four. In the additional Traffic settings at $H = 720$, FAC achieves the lowest available MSE for FreTS and iTransformer and ties PETSA at the reported precision for DLinear.

### A.3 Additional Early-versus-Late Comparison

Figure A.1 provides an additional early-versus-late comparison for $B_k = 25$, which is the second most common mini-batch size induced by PAAS in our experiments after $B_k = 97$. Compared with the $B_k = 97$ setting, this smaller mini-batch size gives later samples less additional revealed context, making the comparison relatively more favorable to earlier adjusted predictions. Nevertheless, later direct predictions still generally achieve comparable or lower MSE than the adjusted predictions of earlier samples. This further supports our argument that using revealed values as POGT supervision is not consistently more beneficial than using the same values as input context in later rolling windows.

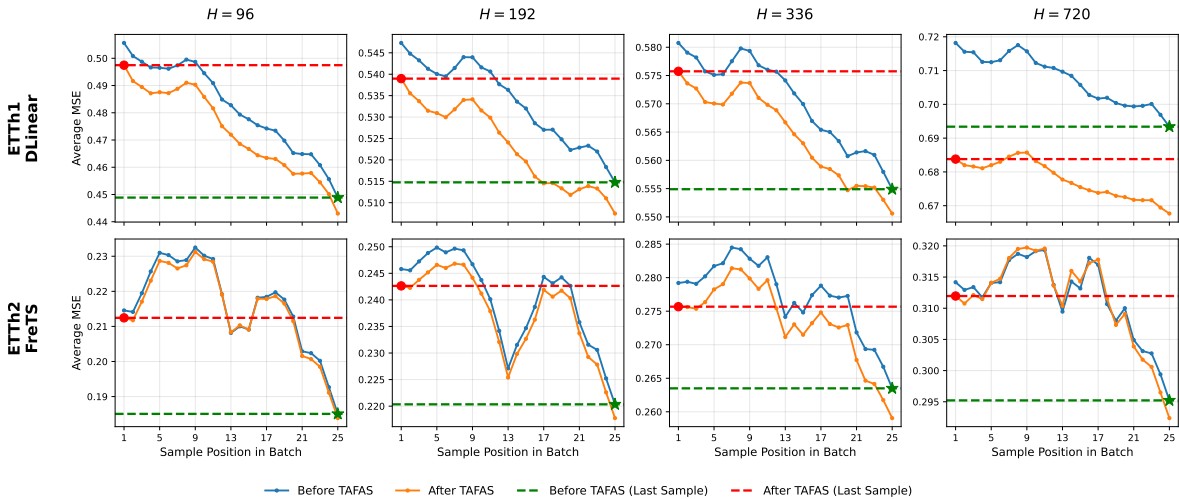

Figure A.1: Additional early-versus-late comparison with mini-batch size $B_k = 25$ on ETTh1 with DLinear and ETTh2 with FreTS. The curves show the average overlapping-region MSE across sample positions within a mini-batch. The dashed horizontal lines mark the adjusted prediction of the first sample and the direct prediction of the last sample as endpoint references.

### A.4 Effect of Using Only Matured Ground Truth

Tables A.8 and A.9 compare TAFAS and PETSA under their original mixed-supervision protocols with the corresponding matured-only variants used in our protocol-controlled experiments. At the reported four-decimal precision, original TAFAS achieves lower MSE in 77 of the 115 matched settings, compared with 38 wins for its matured-only variant. Similarly, original PETSA achieves lower MSE in 74 settings, compared with 40 wins for its matured-only variant and one tie. These results indicate that timely POGT supervision is beneficial more often, although its advantage is not universal across datasets, forecasting horizons, and source forecasters, and the matured-only variants remain competitive. FAC records 79 wins, two ties, and 34 losses against original TAFAS, and 102 wins, one tie, and 12 losses against original PETSA, showing that it remains competitive when the baselines are evaluated under their original supervision protocols.

Table A.8: Original TAFAS versus its variant using only matured ground truth across datasets and source forecasters. Lower MSE is highlighted within each original/matured pair.

| Dataset | Horizon | DLinear | | FreTS | | iTransformer | | OLS | | PatchTST | |
|---------|---------|---------|---------|---------|---------|---------|---------|---------|---------|---------|---------|
| | | Orig. | Matured | Orig. | Matured | Orig. | Matured | Orig. | Matured | Orig. | Matured |
| ETTh1 | 96 | 0.4618 | **0.4617** | 0.4394 | **0.4391** | **0.4398** | 0.4421 | **0.4419** | 0.4428 | **0.4261** | 0.4270 |
| | 192 | 0.5117 | **0.5112** | **0.4940** | 0.4943 | **0.5017** | 0.5020 | **0.4924** | 0.4929 | **0.4809** | 0.4818 |
| | 336 | 0.5604 | **0.5595** | **0.5475** | 0.5483 | 0.5643 | **0.5624** | **0.5415** | 0.5427 | **0.5501** | 0.5506 |
| | 720 | **0.6820** | 0.6926 | 0.6872 | **0.6857** | **0.6591** | 0.6755 | **0.6654** | 0.6704 | 0.6815 | **0.6807** |
| ETTh2 | 96 | **0.2303** | 0.2308 | **0.2362** | 0.2363 | **0.2613** | 0.2620 | **0.2284** | 0.2289 | **0.2378** | 0.2380 |
| | 192 | 0.2842 | **0.2830** | **0.2828** | 0.2830 | **0.2984** | 0.3004 | **0.2817** | 0.2821 | **0.2743** | 0.2752 |
| | 336 | 0.3185 | **0.3175** | 0.3204 | **0.3195** | **0.3286** | 0.3290 | **0.3186** | 0.3189 | 0.3119 | **0.3108** |

| Dataset | Horizon | DLinear | | FreTS | | iTransformer | | OLS | | PatchTST | |
|---|---|---|---|---|---|---|---|---|---|---|---|
| | | Orig. | Matured | Orig. | Matured | Orig. | Matured | Orig. | Matured | Orig. | Matured |
| | 720 | 0.3873 | **0.3825** | 0.3831 | **0.3814** | 0.3981 | **0.3959** | 0.3906 | **0.3859** | **0.4023** | 0.4065 |
| ETTm1 | 96 | **0.3497** | 0.3504 | 0.3574 | **0.3559** | **0.3658** | 0.3680 | **0.3521** | 0.3542 | 0.3912 | **0.3826** |
| | 192 | 0.4166 | **0.4165** | 0.4205 | **0.4191** | **0.4182** | 0.4189 | **0.4164** | 0.4184 | **0.4366** | 0.4370 |
| | 336 | **0.4799** | 0.4801 | 0.4816 | **0.4810** | **0.4780** | 0.4800 | **0.4786** | 0.4811 | **0.4863** | 0.4864 |
| | 720 | **0.5488** | 0.5521 | 0.5474 | **0.5472** | **0.5529** | 0.5579 | **0.5479** | 0.5515 | 0.5414 | **0.5413** |
| ETTm2 | 96 | 0.1584 | **0.1581** | 0.1571 | **0.1566** | **0.1639** | 0.1640 | **0.1592** | 0.1594 | 0.1575 | **0.1569** |
| | 192 | **0.1913** | 0.1917 | 0.1907 | **0.1901** | **0.2159** | 0.2164 | **0.1921** | 0.1924 | 0.2063 | **0.2058** |
| | 336 | **0.2289** | 0.2299 | 0.2292 | **0.2291** | 0.2751 | **0.2735** | **0.2299** | 0.2308 | 0.2487 | **0.2482** |
| | 720 | **0.2968** | 0.2993 | **0.2918** | 0.2924 | **0.3278** | 0.3303 | **0.2982** | 0.2996 | 0.3223 | **0.3218** |
| Weather | 96 | **0.1796** | 0.1803 | **0.1744** | 0.1757 | **0.1647** | 0.1662 | **0.1810** | 0.1825 | 0.1715 | **0.1700** |
| | 192 | **0.2244** | 0.2265 | **0.2151** | 0.2167 | **0.2119** | 0.2129 | **0.2221** | 0.2238 | 0.2130 | **0.2119** |
| | 336 | **0.2709** | 0.2726 | **0.2640** | 0.2661 | **0.2615** | 0.2629 | **0.2707** | 0.2729 | 0.2660 | **0.2659** |
| | 720 | **0.3500** | 0.3509 | **0.3400** | 0.3423 | **0.3410** | 0.3418 | **0.3443** | 0.3467 | 0.3365 | **0.3356** |
| Exchange | 96 | 0.0885 | **0.0875** | **0.0790** | 0.0796 | **0.0867** | 0.0873 | **0.0795** | 0.0799 | **0.0813** | 0.0816 |
| | 192 | 0.1760 | **0.1732** | **0.1639** | 0.1654 | **0.1760** | 0.1800 | **0.1643** | 0.1650 | 0.1673 | **0.1666** |
| | 336 | **0.2941** | 0.3052 | **0.2945** | 0.3180 | **0.2937** | 0.3014 | **0.2835** | 0.2846 | **0.2807** | 0.3048 |

Table A.9: Original PETSA versus its matured-only variant across datasets and source forecasters. Lower MSE is highlighted within each original/matured pair.

| Dataset | Horizon | DLinear | | FreTS | | iTransformer | | OLS | | PatchTST | |
|---|---|---|---|---|---|---|---|---|---|---|---|
| | | Orig. | Matured | Orig. | Matured | Orig. | Matured | Orig. | Matured | Orig. | Matured |
| ETTh1 | 96 | **0.4593** | 0.4613 | **0.4379** | 0.4383 | 0.4553 | **0.4394** | **0.4421** | 0.4449 | 0.4900 | **0.4255** |
| | 192 | **0.5117** | 0.5123 | **0.4946** | 0.4960 | 0.5175 | **0.5014** | **0.4996** | 0.5007 | 0.5458 | **0.4818** |
| | 336 | **0.5618** | 0.5626 | **0.5523** | 0.5530 | 0.5826 | **0.5641** | **0.5477** | 0.5488 | 0.6055 | **0.5570** |
| | 720 | **0.6773** | 0.6938 | 0.6928 | **0.6908** | **0.6748** | 0.6763 | **0.6650** | 0.6805 | 0.7456 | **0.7019** |
| ETTh2 | 96 | **0.2305** | 0.2312 | **0.2361** | 0.2366 | **0.2561** | 0.2602 | **0.2287** | 0.2293 | 0.2476 | **0.2383** |
| | 192 | 0.2874 | **0.2820** | **0.2833** | 0.2844 | **0.2963** | 0.3005 | **0.2824** | 0.2829 | 0.2859 | **0.2810** |
| | 336 | **0.3181** | 0.3190 | **0.3222** | 0.3250 | **0.3291** | 0.3312 | **0.3234** | 0.3240 | 0.3226 | **0.3190** |
| | 720 | **0.3847** | 0.3884 | **0.3879** | 0.3968 | **0.4007** | 0.4049 | **0.3878** | 0.3927 | **0.4131** | 0.4203 |
| ETTm1 | 96 | **0.3524** | 0.3541 | 0.3576 | **0.3572** | 0.3787 | **0.3700** | **0.3560** | 0.3591 | 0.4430 | **0.3817** |
| | 192 | **0.4178** | 0.4192 | 0.4190 | **0.4189** | 0.4289 | **0.4189** | **0.4213** | 0.4252 | 0.4883 | **0.4405** |
| | 336 | **0.4801** | 0.4831 | **0.4781** | 0.4793 | 0.4881 | **0.4788** | **0.4836** | 0.4896 | 0.5395 | **0.4970** |
| | 720 | **0.5534** | 0.5600 | **0.5476** | 0.5523 | 0.5631 | **0.5615** | **0.5523** | 0.5594 | 0.5951 | **0.5479** |
| ETTm2 | 96 | 0.1584 | **0.1583** | 0.1570 | **0.1569** | **0.1639** | 0.1640 | **0.1592** | 0.1595 | 0.1638 | **0.1569** |
| | 192 | **0.1913** | 0.1920 | **0.1907** | 0.1908 | **0.2124** | 0.2157 | **0.1919** | 0.1925 | 0.2126 | **0.2079** |
| | 336 | **0.2291** | 0.2303 | **0.2308** | 0.2312 | **0.2578** | 0.2653 | **0.2302** | 0.2316 | 0.2531 | **0.2473** |
| | 720 | **0.2963** | 0.2984 | **0.2961** | 0.2983 | 0.3265 | **0.3263** | **0.2978** | 0.3003 | 0.3306 | **0.3240** |
| Weather | 96 | 0.1825 | **0.1812** | **0.1763** | 0.1795 | 0.1666 | **0.1662** | **0.1806** | 0.1890 | 0.1794 | **0.1676** |
| | 192 | **0.2254** | 0.2308 | **0.2211** | 0.2229 | **0.2151** | 0.2165 | **0.2253** | 0.2293 | 0.2217 | **0.2120** |
| | 336 | **0.2741** | 0.2776 | **0.2693** | 0.2720 | **0.2669** | 0.2672 | **0.2754** | 0.2785 | 0.2771 | **0.2703** |
| | 720 | **0.3497** | 0.3557 | **0.3527** | 0.3550 | 0.3471 | **0.3470** | **0.3576** | 0.3598 | 0.3402 | **0.3529** |
| Exchange | 96 | **0.0879** | 0.0898 | **0.0818** | 0.0825 | 0.0887 | **0.0881** | **0.0808** | 0.0813 | 0.0911 | **0.0850** |
| | 192 | **0.1734** | 0.1787 | **0.1718** | 0.1729 | **0.1820** | 0.1820 | **0.1705** | 0.1719 | 0.1827 | **0.1765** |
| | 336 | **0.2955** | 0.3154 | **0.3191** | 0.3223 | 0.3179 | **0.3105** | **0.2920** | 0.3058 | 0.3348 | **0.3188** |

### A.5 Ablation Studies on Calibration Design

We conduct ablation studies to examine the contributions of the main components of FAC. We first evaluate the multiplicative term, additive term, complex phase modification, and residual gate in each FreqGCM. We then study the respective roles of calibration on the input and output. All component ablations are evaluated on the same 115 experimental settings used in the primary experiments.

#### A.5.1 Multiplicative Term

We first remove the frequency-wise multiplicative term while retaining the additive term, complex-valued parameterization, and residual gate. As shown in Table A.10, full FAC achieves lower MSE in all 115 settings. This substantial and consistent degradation indicates that the multiplicative term is central to FAC.

Table A.10: Full FAC versus its variant without the multiplicative term across datasets and source forecasters. "No Mult." removes the frequency-wise multiplicative transformation while retaining the other components. Lower MSE is highlighted within each pair.

| Dataset | Horizon | DLinear | | FreTS | | iTransformer | | OLS | | PatchTST | |
|---------|---------|---------|----------|---------|----------|--------------|----------|---------|----------|----------|----------|
| | | FAC | No Mult. | FAC | No Mult. | FAC | No Mult. | FAC | No Mult. | FAC | No Mult. |
| ETTh1 | 96 | **0.4554** | 0.4695 | **0.4357** | 0.4461 | **0.4400** | 0.4478 | **0.4382** | 0.4512 | **0.4241** | 0.4317 |
| | 192 | **0.5089** | 0.5214 | **0.4934** | 0.5022 | **0.5010** | 0.5091 | **0.4921** | 0.5047 | **0.4797** | 0.4893 |
| | 336 | **0.5582** | 0.5660 | **0.5498** | 0.5544 | **0.5587** | 0.5663 | **0.5422** | 0.5512 | **0.5525** | 0.5603 |
| | 720 | **0.6891** | 0.7112 | **0.7067** | 0.7179 | **0.6687** | 0.7009 | **0.6780** | 0.6995 | **0.7096** | 0.7110 |
| ETTh2 | 96 | **0.2288** | 0.2322 | **0.2338** | 0.2381 | **0.2561** | 0.2641 | **0.2274** | 0.2307 | **0.2374** | 0.2385 |
| | 192 | **0.2819** | 0.2859 | **0.2814** | 0.2860 | **0.2970** | 0.3070 | **0.2796** | 0.2836 | **0.2747** | 0.2827 |
| | 336 | **0.3144** | 0.3241 | **0.3172** | 0.3278 | **0.3284** | 0.3385 | **0.3137** | 0.3240 | **0.3079** | 0.3219 |
| | 720 | **0.3825** | 0.4058 | **0.3788** | 0.4012 | **0.3982** | 0.4206 | **0.3854** | 0.4109 | **0.4002** | 0.4270 |
| ETTm1 | 96 | **0.3465** | 0.3715 | **0.3517** | 0.3656 | **0.3459** | 0.3743 | **0.3462** | 0.3709 | **0.3530** | 0.3925 |
| | 192 | **0.4139** | 0.4439 | **0.4183** | 0.4311 | **0.4087** | 0.4368 | **0.4140** | 0.4438 | **0.4213** | 0.4502 |
| | 336 | **0.4791** | 0.5184 | **0.4836** | 0.4991 | **0.4781** | 0.5047 | **0.4794** | 0.5180 | **0.4787** | 0.5041 |
| | 720 | **0.5494** | 0.5931 | **0.5533** | 0.5698 | **0.5460** | 0.5997 | **0.5497** | 0.5925 | **0.5399** | 0.5606 |
| ETTm2 | 96 | **0.1560** | 0.1599 | **0.1563** | 0.1580 | **0.1613** | 0.1642 | **0.1563** | 0.1602 | **0.1558** | 0.1577 |
| | 192 | **0.1894** | 0.1931 | **0.1901** | 0.1922 | **0.2067** | 0.2174 | **0.1897** | 0.1936 | **0.1976** | 0.2083 |
| | 336 | **0.2286** | 0.2325 | **0.2294** | 0.2320 | **0.2623** | 0.2704 | **0.2289** | 0.2332 | **0.2422** | 0.2479 |
| | 720 | **0.2984** | 0.3061 | **0.2955** | 0.3013 | **0.3290** | 0.3451 | **0.2973** | 0.3066 | **0.3192** | 0.3279 |
| Weather | 96 | **0.1797** | 0.1954 | **0.1678** | 0.1846 | **0.1605** | 0.1707 | **0.1728** | 0.1958 | **0.1635** | 0.1744 |
| | 192 | **0.2231** | 0.2403 | **0.2114** | 0.2309 | **0.2103** | 0.2240 | **0.2171** | 0.2407 | **0.2050** | 0.2186 |
| | 336 | **0.2717** | 0.2919 | **0.2621** | 0.2842 | **0.2618** | 0.2802 | **0.2665** | 0.2922 | **0.2570** | 0.2772 |
| | 720 | **0.3426** | 0.3644 | **0.3376** | 0.3600 | **0.3357** | 0.3566 | **0.3401** | 0.3646 | **0.3341** | 0.3560 |
| Exchange | 96 | **0.0875** | 0.0910 | **0.0810** | 0.0827 | **0.0867** | 0.0877 | **0.0793** | 0.0808 | **0.0813** | 0.0837 |
| | 192 | **0.1758** | 0.1818 | **0.1667** | 0.1718 | **0.1773** | 0.1792 | **0.1648** | 0.1710 | **0.1692** | 0.1760 |
| | 336 | **0.2985** | 0.3262 | **0.3166** | 0.3181 | **0.2983** | 0.3112 | **0.2924** | 0.3142 | **0.3011** | 0.3299 |

#### A.5.2 Additive Term

We next remove the frequency-wise additive term while retaining the other FAC components. As shown in Table A.11, full FAC achieves lower MSE in 92 settings, the ablated variant performs better in 20 settings, and three settings are tied at the reported precision. The additive term therefore provides a broadly beneficial, though not universal, complement to multiplicative modulation.

Table A.11: Full FAC versus its variant without the additive term across datasets and source forecasters. "No Add." removes the frequency-wise additive term while retaining the other components. Lower MSE is highlighted within each pair.

| Dataset | Horizon | DLinear | | FreTS | | iTransformer | | OLS | | PatchTST | |
|---|---|---|---|---|---|---|---|---|---|---|---|
| | | FAC | No Add. | FAC | No Add. | FAC | No Add. | FAC | No Add. | FAC | No Add. |
| ETTh1 | 96 | **0.4554** | 0.4569 | **0.4357** | 0.4375 | **0.4400** | 0.4402 | **0.4382** | 0.4398 | **0.4241** | 0.4250 |
| | 192 | **0.5089** | 0.5095 | 0.4934 | **0.4932** | **0.5010** | 0.5011 | **0.4921** | 0.4925 | **0.4797** | 0.4801 |
| | 336 | 0.5582 | **0.5577** | **0.5498** | 0.5502 | 0.5587 | **0.5582** | 0.5422 | **0.5417** | **0.5525** | 0.5527 |
| | 720 | **0.6891** | 0.6901 | **0.7067** | 0.7081 | **0.6687** | 0.6688 | **0.6780** | 0.6787 | **0.7096** | 0.7097 |
| ETTh2 | 96 | **0.2288** | 0.2290 | **0.2338** | 0.2339 | 0.2561 | **0.2548** | **0.2274** | 0.2276 | 0.2374 | **0.2365** |
| | 192 | **0.2819** | 0.2823 | **0.2814** | 0.2816 | **0.2970** | 0.2972 | **0.2796** | 0.2802 | **0.2747** | 0.2752 |
| | 336 | **0.3144** | 0.3152 | **0.3172** | 0.3179 | **0.3284** | 0.3300 | **0.3137** | 0.3148 | **0.3079** | 0.3084 |
| | 720 | **0.3825** | 0.3836 | **0.3788** | 0.3814 | **0.3982** | 0.3986 | **0.3854** | 0.3872 | **0.4002** | 0.4012 |
| ETTm1 | 96 | **0.3465** | 0.3482 | **0.3517** | 0.3539 | **0.3459** | 0.3488 | **0.3462** | 0.3478 | **0.3530** | 0.3569 |
| | 192 | **0.4139** | 0.4158 | **0.4183** | 0.4197 | **0.4087** | 0.4102 | **0.4140** | 0.4160 | **0.4213** | 0.4240 |
| | 336 | **0.4791** | 0.4814 | **0.4836** | 0.4850 | **0.4781** | 0.4798 | **0.4794** | 0.4819 | **0.4787** | 0.4801 |
| | 720 | **0.5494** | 0.5509 | **0.5533** | 0.5538 | 0.5460 | **0.5458** | **0.5497** | 0.5516 | **0.5399** | 0.5403 |
| ETTm2 | 96 | **0.1560** | 0.1563 | **0.1563** | 0.1569 | 0.1613 | **0.1611** | **0.1563** | 0.1566 | **0.1558** | 0.1563 |
| | 192 | **0.1894** | 0.1896 | **0.1901** | 0.1906 | **0.2067** | 0.2080 | **0.1897** | 0.1900 | **0.1976** | 0.1984 |
| | 336 | **0.2286** | 0.2290 | **0.2294** | 0.2314 | **0.2623** | 0.2711 | **0.2289** | 0.2292 | **0.2422** | 0.2429 |
| | 720 | **0.2984** | 0.2990 | **0.2955** | 0.3152 | **0.3290** | 0.3292 | **0.2973** | 0.2977 | **0.3192** | 0.3196 |
| Weather | 96 | **0.1797** | **0.1797** | **0.1678** | 0.1687 | **0.1605** | 0.1610 | 0.1728 | **0.1726** | **0.1635** | **0.1635** |
| | 192 | 0.2231 | **0.2230** | 0.2114 | **0.2113** | 0.2103 | **0.2092** | 0.2171 | **0.2169** | **0.2050** | 0.2055 |
| | 336 | **0.2717** | **0.2717** | 0.2621 | **0.2619** | **0.2618** | 0.2635 | 0.2665 | **0.2664** | 0.2570 | **0.2566** |
| | 720 | **0.3426** | 0.3428 | **0.3376** | 0.3390 | **0.3357** | 0.3361 | **0.3401** | 0.3407 | 0.3341 | **0.3329** |
| Exchange | 96 | **0.0875** | 0.0876 | **0.0810** | 0.0811 | **0.0867** | 0.0870 | 0.0793 | **0.0790** | **0.0813** | 0.0815 |
| | 192 | 0.1758 | **0.1755** | **0.1667** | 0.1678 | 0.1773 | **0.1746** | **0.1648** | 0.1651 | **0.1692** | 0.1695 |
| | 336 | **0.2985** | 0.3001 | **0.3166** | 0.3222 | **0.2983** | 0.3031 | **0.2924** | 0.2950 | **0.3011** | 0.3108 |

### A.5.3 Complex Phase Modification

To examine the role of phase adjustment, we restrict the frequency-domain transformation so that it cannot introduce complex phase modification. Table A.12 shows that full FAC achieves lower MSE in 84 settings, compared with 29 wins for the ablated variant and two ties. These results indicate that direct phase adjustment is useful in a majority of settings, while its effect remains dependent on the dataset and source forecaster.

Table A.12: Full FAC versus its variant without complex phase modification across datasets and source forecasters. "No Cplx." removes the complex phase modification while retaining the other components. Lower MSE is highlighted within each pair.

| Dataset | Horizon | DLinear | | FreTS | | iTransformer | | OLS | | PatchTST | |
|---|---|---|---|---|---|---|---|---|---|---|---|
| | | FAC | No Cplx. | FAC | No Cplx. | FAC | No Cplx. | FAC | No Cplx. | FAC | No Cplx. |
| ETTh1 | 96 | 0.4554 | **0.4535** | 0.4357 | **0.4356** | **0.4400** | 0.4402 | **0.4382** | 0.4383 | **0.4241** | 0.4242 |
| | 192 | 0.5089 | **0.5053** | 0.4934 | **0.4922** | **0.5010** | 0.5015 | 0.4921 | **0.4904** | 0.4797 | **0.4777** |
| | 336 | 0.5582 | **0.5535** | 0.5498 | **0.5473** | 0.5587 | **0.5561** | 0.5422 | **0.5379** | **0.5525** | 0.5530 |
| | 720 | 0.6891 | **0.6842** | 0.7067 | **0.7064** | 0.6687 | **0.6642** | 0.6780 | **0.6751** | 0.7096 | **0.7053** |
| ETTh2 | 96 | **0.2288** | **0.2288** | 0.2338 | **0.2331** | 0.2561 | **0.2550** | **0.2274** | 0.2276 | 0.2374 | **0.2363** |
| | 192 | **0.2819** | **0.2819** | 0.2814 | **0.2807** | **0.2970** | 0.2976 | **0.2796** | 0.2800 | 0.2747 | **0.2744** |
| | 336 | **0.3144** | 0.3148 | **0.3172** | 0.3282 | **0.3284** | 0.3309 | **0.3137** | 0.3145 | **0.3079** | 0.3080 |
| | 720 | **0.3825** | 0.3830 | **0.3788** | 0.4063 | 0.3982 | **0.3967** | **0.3854** | 0.3867 | **0.4002** | 0.4011 |
| ETTm1 | 96 | **0.3465** | 0.3473 | **0.3517** | 0.3529 | **0.3459** | 0.3582 | **0.3462** | 0.3470 | **0.3530** | 0.3630 |
| | 192 | **0.4139** | 0.4149 | **0.4183** | 0.4196 | **0.4087** | 0.4177 | **0.4140** | 0.4153 | **0.4213** | 0.4253 |
| | 336 | **0.4791** | 0.4806 | **0.4836** | 0.4844 | **0.4781** | 0.4869 | **0.4794** | 0.4812 | **0.4787** | 0.4828 |
| | 720 | 0.5494 | **0.5488** | **0.5533** | 0.5537 | **0.5460** | 0.5528 | **0.5497** | 0.5500 | **0.5399** | 0.5404 |
| ETTm2 | 96 | **0.1560** | 0.1562 | **0.1563** | 0.1566 | **0.1613** | 0.1628 | **0.1563** | 0.1564 | **0.1558** | 0.1567 |

| Dataset | Horizon | DLinear | | FreTS | | iTransformer | | OLS | | PatchTST | |
|---|---|---|---|---|---|---|---|---|---|---|---|
| | | FAC | No Cplx. | FAC | No Cplx. | FAC | No Cplx. | FAC | No Cplx. | FAC | No Cplx. |
| | 192 | **0.1894** | 0.1895 | **0.1901** | 0.1904 | **0.2067** | 0.2098 | **0.1897** | 0.1899 | **0.1976** | 0.1996 |
| | 336 | **0.2286** | 0.2289 | **0.2294** | 0.2309 | **0.2623** | 0.2665 | **0.2289** | 0.2290 | **0.2422** | 0.2432 |
| | 720 | **0.2984** | 0.2989 | **0.2955** | 0.3109 | **0.3290** | 0.3319 | **0.2973** | 0.2976 | **0.3192** | 0.3205 |
| Weather | 96 | **0.1797** | 0.1827 | **0.1678** | 0.1688 | **0.1605** | 0.1615 | **0.1728** | 0.1772 | 0.1635 | **0.1634** |
| | 192 | **0.2231** | 0.2244 | **0.2114** | 0.2117 | **0.2103** | 0.2112 | **0.2171** | 0.2192 | **0.2050** | 0.2051 |
| | 336 | **0.2717** | 0.2723 | **0.2621** | 0.2625 | **0.2618** | 0.2645 | **0.2665** | 0.2676 | 0.2570 | **0.2563** |
| | 720 | **0.3426** | 0.3428 | **0.3376** | 0.3390 | **0.3357** | 0.3368 | **0.3401** | 0.3407 | 0.3341 | **0.3333** |
| Exchange | 96 | **0.0875** | 0.0876 | 0.0810 | **0.0807** | **0.0867** | 0.0870 | 0.0793 | **0.0790** | **0.0813** | 0.0814 |
| | 192 | 0.1758 | **0.1756** | **0.1667** | 0.1678 | 0.1773 | **0.1738** | **0.1648** | 0.1651 | **0.1692** | 0.1693 |
| | 336 | **0.2985** | 0.2995 | **0.3166** | 0.3215 | **0.2983** | 0.3031 | **0.2924** | 0.2941 | **0.3011** | 0.3115 |

### A.5.4  Residual Gate

We further remove the residual gate while retaining the complete frequency-domain affine transformation. As shown in Table A.13, full FAC records 60 wins, compared with 48 wins for the ungated variant and seven ties. The gate therefore provides a modest overall benefit, although its effect is more setting-dependent than those of the multiplicative and additive components.

Table A.13:  Full FAC versus its variant without the residual gate across datasets and source forecasters. "No Gate" removes the residual gate while retaining the other components. Lower MSE is highlighted within each pair.

| Dataset | Horizon | DLinear | | FreTS | | iTransformer | | OLS | | PatchTST | |
|---|---|---|---|---|---|---|---|---|---|---|---|
| | | FAC | No Gate | FAC | No Gate | FAC | No Gate | FAC | No Gate | FAC | No Gate |
| ETTh1 | 96 | **0.4554** | 0.4558 | **0.4357** | 0.4442 | **0.4400** | 0.4429 | **0.4382** | 0.4404 | **0.4241** | 0.4277 |
| | 192 | **0.5089** | 0.5102 | **0.4934** | 0.4977 | **0.5010** | 0.5078 | **0.4921** | 0.4921 | **0.4797** | 0.4800 |
| | 336 | **0.5582** | 0.5643 | **0.5498** | 0.5587 | **0.5587** | 0.5590 | **0.5422** | 0.5422 | **0.5525** | 0.5530 |
| | 720 | **0.6891** | 0.6908 | **0.7067** | 0.7104 | 0.6687 | **0.6679** | **0.6780** | 0.6780 | 0.7096 | **0.7095** |
| ETTh2 | 96 | 0.2288 | **0.2286** | 0.2338 | **0.2329** | 0.2561 | **0.2541** | **0.2274** | 0.2288 | 0.2374 | **0.2368** |
| | 192 | 0.2819 | **0.2818** | 0.2814 | **0.2807** | **0.2970** | 0.2980 | **0.2796** | 0.2826 | **0.2747** | 0.2764 |
| | 336 | **0.3144** | 0.3145 | **0.3172** | 0.3178 | **0.3284** | 0.3350 | **0.3137** | 0.3140 | **0.3079** | 0.3083 |
| | 720 | **0.3825** | 0.3888 | **0.3788** | 0.4010 | **0.3982** | 0.4103 | **0.3854** | 0.3918 | 0.4002 | **0.3978** |
| ETTm1 | 96 | **0.3465** | 0.3467 | 0.3517 | **0.3489** | 0.3459 | **0.3442** | 0.3462 | **0.3460** | 0.3530 | **0.3518** |
| | 192 | **0.4139** | 0.4144 | 0.4183 | **0.4176** | 0.4087 | **0.4086** | **0.4140** | 0.4140 | 0.4213 | **0.4216** |
| | 336 | **0.4791** | 0.4795 | **0.4836** | 0.4843 | 0.4781 | **0.4777** | 0.4794 | **0.4792** | 0.4787 | **0.4792** |
| | 720 | **0.5494** | 0.5502 | **0.5533** | 0.5551 | 0.5460 | **0.5455** | **0.5497** | 0.5505 | 0.5399 | **0.5396** |
| ETTm2 | 96 | **0.1560** | 0.1561 | **0.1563** | 0.1563 | 0.1613 | **0.1610** | 0.1563 | **0.1560** | **0.1558** | 0.1562 |
| | 192 | **0.1894** | 0.1907 | 0.1901 | **0.1896** | 0.2067 | **0.2061** | **0.1897** | 0.1907 | **0.1976** | 0.1981 |
| | 336 | **0.2286** | 0.2314 | **0.2294** | 0.2294 | **0.2623** | 0.2637 | **0.2289** | 0.2306 | **0.2422** | 0.2422 |
| | 720 | **0.2984** | 0.3024 | 0.2955 | **0.2946** | **0.3290** | 0.3306 | 0.2973 | **0.2964** | 0.3192 | **0.3180** |
| Weather | 96 | 0.1797 | **0.1781** | 0.1678 | **0.1671** | **0.1605** | 0.1608 | 0.1728 | **0.1707** | **0.1635** | 0.1641 |
| | 192 | 0.2231 | **0.2215** | 0.2114 | **0.2107** | 0.2103 | **0.2097** | 0.2171 | **0.2155** | **0.2050** | 0.2054 |
| | 336 | 0.2717 | **0.2702** | 0.2621 | **0.2614** | **0.2618** | 0.2625 | 0.2665 | **0.2653** | 0.2570 | **0.2567** |
| | 720 | 0.3426 | **0.3411** | 0.3376 | **0.3375** | 0.3357 | **0.3348** | 0.3401 | **0.3395** | 0.3341 | **0.3337** |
| Exchange | 96 | 0.0875 | **0.0874** | **0.0810** | 0.0814 | **0.0867** | 0.0873 | **0.0793** | 0.0800 | **0.0813** | 0.0817 |
| | 192 | **0.1758** | 0.1820 | **0.1667** | 0.1658 | **0.1773** | 0.1858 | **0.1648** | 0.1647 | **0.1692** | 0.1693 |
| | 336 | 0.2985 | **0.2925** | 0.3166 | **0.3150** | 0.2983 | **0.2949** | **0.2924** | 0.2941 | **0.3011** | 0.3062 |

### A.5.5  Input- and Output-Side Calibration

Finally, we examine whether both input- and output-side FreqGCMs are needed. We first remove the output-side FreqGCM, producing an input-only variant. As shown in Table A.14, full FAC achieves lower MSE in 111 of the 115 settings, while the input-only variant performs better in four settings. This result demonstrates that output-side calibration is important for the complete FAC design.

Table A.14: Full FAC versus its input-only variant across datasets and source forecasters. "Input Only" removes the output-side FreqGCM while retaining the input-side FreqGCM. Lower MSE is highlighted within each pair.

| Dataset | Horizon | DLinear | | FreTS | | iTransformer | | OLS | | PatchTST | |
|---|---|---|---|---|---|---|---|---|---|---|---|
| | | FAC | Input Only | FAC | Input Only | FAC | Input Only | FAC | Input Only | FAC | Input Only |
| ETTh1 | 96 | **0.4554** | 0.4581 | **0.4357** | 0.4467 | **0.4400** | 0.4492 | **0.4382** | 0.4410 | **0.4241** | 0.4314 |
| | 192 | 0.5089 | **0.5088** | **0.4934** | 0.5030 | **0.5010** | 0.5097 | **0.4921** | 0.4942 | **0.4797** | 0.4873 |
| | 336 | **0.5582** | 0.5583 | **0.5498** | 0.5557 | **0.5587** | 0.5667 | **0.5422** | 0.5424 | **0.5525** | 0.5613 |
| | 720 | **0.6891** | 0.6927 | **0.7067** | 0.7194 | **0.6687** | 0.6871 | **0.6780** | 0.6842 | **0.7096** | 0.7147 |
| ETTh2 | 96 | **0.2288** | 0.2298 | **0.2338** | 0.2395 | **0.2561** | 0.2635 | **0.2274** | 0.2281 | **0.2374** | 0.2383 |
| | 192 | **0.2819** | 0.2830 | **0.2814** | 0.2891 | **0.2970** | 0.3015 | **0.2796** | 0.2800 | **0.2747** | 0.2759 |
| | 336 | **0.3144** | 0.3160 | **0.3172** | 0.3429 | **0.3284** | 0.3450 | **0.3137** | 0.3147 | **0.3079** | 0.3105 |
| | 720 | **0.3825** | 0.3858 | **0.3788** | 0.4311 | **0.3982** | 0.4203 | 0.3854 | **0.3881** | **0.4002** | 0.4209 |
| ETTm1 | 96 | **0.3465** | 0.3506 | **0.3517** | 0.3686 | **0.3459** | 0.3638 | **0.3462** | 0.3504 | **0.3530** | 0.3617 |
| | 192 | **0.4139** | 0.4178 | **0.4183** | 0.4368 | **0.4087** | 0.4310 | **0.4140** | 0.4185 | **0.4213** | 0.4310 |
| | 336 | **0.4791** | 0.4832 | **0.4836** | 0.5072 | **0.4781** | 0.5014 | **0.4794** | 0.4843 | **0.4787** | 0.4863 |
| | 720 | **0.5494** | 0.5527 | **0.5533** | 0.5802 | **0.5460** | 0.5703 | **0.5497** | 0.5543 | **0.5399** | 0.5479 |
| ETTm2 | 96 | **0.1560** | 0.1573 | **0.1563** | 0.1583 | **0.1613** | 0.1648 | **0.1563** | 0.1577 | **0.1558** | 0.1559 |
| | 192 | **0.1894** | 0.1903 | **0.1901** | 0.1924 | **0.2067** | 0.2160 | **0.1897** | 0.1907 | 0.1976 | **0.1972** |
| | 336 | **0.2286** | 0.2291 | **0.2294** | 0.2325 | **0.2623** | 0.2770 | **0.2289** | 0.2304 | **0.2422** | 0.2441 |
| | 720 | **0.2984** | 0.2987 | **0.2955** | 0.3035 | **0.3290** | 0.3452 | **0.2973** | 0.3036 | **0.3192** | 0.3291 |
| Weather | 96 | **0.1797** | 0.1838 | **0.1678** | 0.1720 | **0.1605** | 0.1708 | **0.1728** | 0.1785 | **0.1635** | 0.1697 |
| | 192 | **0.2231** | 0.2274 | **0.2114** | 0.2167 | **0.2103** | 0.2201 | **0.2171** | 0.2218 | **0.2050** | 0.2097 |
| | 336 | **0.2717** | 0.2766 | **0.2621** | 0.2678 | **0.2618** | 0.2791 | **0.2665** | 0.2708 | **0.2570** | 0.2642 |
| | 720 | **0.3426** | 0.3484 | **0.3376** | 0.3439 | **0.3357** | 0.3551 | **0.3401** | 0.3428 | **0.3341** | 0.3505 |
| Exchange | 96 | **0.0875** | 0.0885 | **0.0810** | 0.0841 | **0.0867** | 0.0880 | **0.0793** | 0.0796 | **0.0813** | 0.0826 |
| | 192 | 0.1758 | **0.1734** | **0.1667** | 0.1754 | 0.1773 | **0.1724** | **0.1648** | 0.1669 | **0.1692** | 0.1718 |
| | 336 | **0.2985** | 0.3107 | **0.3166** | 0.3278 | **0.2983** | 0.3143 | **0.2924** | 0.3028 | **0.3011** | 0.3201 |

We next remove input calibration and retain only the output adapter. Table A.15 compares the standard input-output design with this output-only variant across datasets, forecasting horizons, and source forecasters. Input-output calibration generally performs better across TAFAS, PETSA, and FAC, indicating that calibrating the look-back input provides additional benefits beyond output calibration alone. Together with the input-only results, these comparisons show that input and output calibration play complementary roles.

Table A.15: Input-output versus output-only calibration across source forecasters. Lower MSE is highlighted within each input-output/output-only pair.

| Dataset | Horizon | DLinear | | | | | | FreTS | | | | | |
|---|---|---|---|---|---|---|---|---|---|---|---|---|---|
| | | TAFAS | | PETSA | | FAC | | TAFAS | | PETSA | | FAC | |
| | | In+Out | Out | In+Out | Out | In+Out | Out | In+Out | Out | In+Out | Out | In+Out | Out |
| ETTh1 | 96 | **0.4617** | 0.4626 | **0.4613** | 0.4616 | **0.4554** | 0.4558 | **0.4391** | **0.4391** | **0.4383** | **0.4383** | **0.4357** | 0.4358 |
| | 192 | **0.5112** | 0.5115 | **0.5123** | 0.5128 | 0.5089 | **0.5071** | **0.4943** | 0.4945 | **0.4960** | 0.4961 | 0.4934 | **0.4922** |
| | 336 | **0.5595** | 0.5600 | 0.5626 | **0.5622** | 0.5582 | **0.5555** | **0.5483** | 0.5485 | **0.5530** | **0.5530** | 0.5498 | **0.5468** |
| | 720 | **0.6926** | 0.6970 | **0.6938** | 0.7003 | **0.6891** | 0.6918 | 0.6857 | **0.6855** | **0.6908** | 0.6937 | 0.7067 | **0.7028** |
| ETTh2 | 96 | **0.2308** | 0.2315 | **0.2312** | 0.2316 | **0.2288** | 0.2302 | **0.2363** | 0.2366 | **0.2366** | 0.2370 | **0.2338** | 0.2341 |
| | 192 | 0.2830 | **0.2818** | **0.2820** | 0.2834 | **0.2819** | 0.2832 | **0.2830** | 0.2831 | **0.2844** | 0.2846 | 0.2814 | **0.2810** |
| | 336 | **0.3175** | 0.3180 | **0.3190** | 0.3202 | **0.3144** | 0.3165 | **0.3195** | **0.3195** | **0.3250** | 0.3252 | **0.3172** | 0.3180 |
| | 720 | **0.3825** | 0.3836 | **0.3884** | 0.3909 | **0.3825** | 0.3868 | **0.3814** | **0.3814** | **0.3968** | 0.3971 | **0.3788** | 0.3862 |
| ETTm1 | 96 | **0.3504** | 0.3528 | **0.3541** | 0.3580 | **0.3465** | 0.3483 | **0.3559** | 0.3591 | **0.3572** | 0.3593 | **0.3517** | 0.3557 |
| | 192 | **0.4165** | 0.4172 | **0.4192** | 0.4226 | **0.4139** | 0.4160 | **0.4191** | 0.4200 | **0.4189** | 0.4201 | **0.4183** | 0.4187 |
| | 336 | **0.4801** | 0.4805 | **0.4831** | 0.4863 | **0.4791** | 0.4832 | **0.4810** | 0.4816 | **0.4793** | 0.4802 | 0.4836 | **0.4826** |
| | 720 | **0.5521** | 0.5526 | **0.5600** | 0.5618 | **0.5494** | 0.5517 | **0.5472** | 0.5475 | **0.5523** | 0.5526 | 0.5533 | **0.5496** |
| ETTm2 | 96 | **0.1581** | 0.1586 | **0.1583** | 0.1587 | **0.1560** | 0.1569 | **0.1566** | 0.1571 | **0.1569** | 0.1575 | **0.1563** | 0.1565 |
| | 192 | **0.1917** | 0.1919 | **0.1920** | 0.1922 | **0.1894** | 0.1901 | **0.1901** | 0.1904 | **0.1908** | 0.1912 | **0.1901** | 0.1903 |

| Dataset | Horizon | DLinear | | | | | | FreTS | | | | | |
|---|---|---|---|---|---|---|---|---|---|---|---|---|---|
| | | TAFAS | | PETSA | | FAC | | TAFAS | | PETSA | | FAC | |
| | | In+Out | Out | In+Out | Out | In+Out | Out | In+Out | Out | In+Out | Out | In+Out | Out |
| | 336 | **0.2299** | 0.2301 | **0.2303** | 0.2304 | 0.2286 | **0.2285** | **0.2291** | 0.2293 | **0.2312** | 0.2314 | 0.2294 | **0.2287** |
| | 720 | 0.2993 | **0.2987** | 0.2984 | **0.2968** | 0.2984 | **0.2980** | **0.2924** | 0.2926 | **0.2983** | 0.2984 | 0.2955 | **0.2934** |
| Weather | 96 | **0.1803** | 0.1839 | **0.1812** | 0.1876 | **0.1797** | 0.1881 | **0.1757** | 0.1787 | **0.1795** | 0.1814 | **0.1678** | 0.1764 |
| | 192 | **0.2265** | 0.2282 | **0.2308** | 0.2313 | **0.2231** | 0.2298 | **0.2167** | 0.2186 | **0.2229** | 0.2240 | **0.2114** | 0.2175 |
| | 336 | **0.2726** | 0.2737 | **0.2776** | 0.2778 | **0.2717** | 0.2777 | **0.2661** | 0.2669 | **0.2720** | 0.2720 | **0.2621** | 0.2661 |
| | 720 | **0.3509** | 0.3542 | **0.3557** | 0.3565 | **0.3426** | 0.3482 | **0.3423** | 0.3426 | **0.3550** | 0.3554 | **0.3376** | 0.3392 |
| Exchange | 96 | **0.0875** | 0.0892 | **0.0898** | 0.0904 | **0.0875** | 0.0887 | **0.0796** | 0.0809 | **0.0825** | 0.0827 | **0.0810** | 0.0828 |
| | 192 | **0.1732** | 0.1758 | **0.1787** | 0.1802 | 0.1758 | **0.1740** | **0.1654** | 0.1673 | **0.1729** | 0.1731 | 0.1667 | **0.1635** |
| | 336 | **0.3052** | 0.3083 | **0.3154** | 0.3188 | **0.2985** | 0.3125 | **0.3180** | 0.3197 | **0.3223** | 0.3230 | 0.3166 | **0.3017** |

| Dataset | Horizon | iTransformer | | | | | | OLS | | | | | | PatchTST | | | | | |
|---|---|---|---|---|---|---|---|---|---|---|---|---|---|---|---|---|---|---|---|
| | | TAFAS | | PETSA | | FAC | | TAFAS | | PETSA | | FAC | | TAFAS (PatchTST) | | PETSA | | FAC | |
| | | In+Out | Out | In+Out | Out | In+Out | Out | In+Out | Out | In+Out | Out | In+Out | Out | In+Out | Out | In+Out | Out | In+Out | Out |
| ETTh1 | 96 | 0.4421 | **0.4420** | **0.4394** | 0.4398 | 0.4400 | **0.4378** | **0.4428** | 0.4454 | **0.4449** | 0.4459 | **0.4382** | 0.4390 | 0.4270 | **0.4260** | 0.4255 | **0.4253** | 0.4241 | **0.4238** |
| | 192 | 0.5020 | **0.5016** | **0.5014** | 0.5017 | 0.5010 | **0.4987** | **0.4929** | 0.4951 | **0.5007** | 0.5016 | **0.4921** | 0.4926 | 0.4818 | **0.4811** | 0.4818 | **0.4818** | 0.4797 | **0.4790** |
| | 336 | **0.5624** | 0.5625 | 0.5641 | **0.5639** | 0.5587 | **0.5561** | **0.5427** | 0.5443 | **0.5488** | 0.5492 | 0.5422 | **0.5405** | 0.5506 | **0.5505** | 0.5570 | 0.5572 | 0.5525 | **0.5495** |
| | 720 | **0.6755** | 0.6772 | **0.6763** | 0.6783 | **0.6687** | 0.6718 | **0.6704** | 0.6747 | **0.6805** | 0.6835 | **0.6780** | 0.6832 | 0.6807 | **0.6802** | 0.7019 | **0.7012** | 0.7096 | **0.6964** |
| ETTh2 | 96 | **0.2620** | 0.2623 | 0.2602 | **0.2593** | 0.2561 | **0.2561** | **0.2289** | 0.2296 | **0.2293** | 0.2299 | **0.2274** | 0.2280 | 0.2380 | **0.2378** | **0.2383** | 0.2383 | **0.2374** | 0.2374 |
| | 192 | **0.3004** | 0.3011 | **0.3005** | 0.3008 | 0.2970 | **0.2955** | **0.2821** | 0.2826 | **0.2829** | 0.2833 | **0.2796** | 0.2804 | **0.2752** | 0.2753 | **0.2810** | 0.2817 | 0.2747 | **0.2737** |
| | 336 | 0.3290 | **0.3286** | 0.3312 | **0.3310** | 0.3284 | **0.3266** | **0.3189** | 0.3201 | **0.3240** | 0.3245 | **0.3137** | 0.3163 | **0.3108** | 0.3108 | **0.3190** | 0.3200 | **0.3079** | 0.3103 |
| | 720 | **0.3959** | 0.3961 | **0.4049** | 0.4059 | 0.3982 | **0.3952** | **0.3859** | 0.3870 | **0.3927** | 0.3955 | **0.3854** | 0.3901 | **0.4065** | 0.4073 | **0.4203** | 0.4215 | **0.4002** | 0.4043 |
| ETTm1 | 96 | **0.3680** | 0.3753 | **0.3700** | 0.3750 | **0.3459** | 0.3619 | **0.3542** | 0.3565 | **0.3591** | 0.3601 | **0.3462** | 0.3489 | **0.3826** | 0.3904 | **0.3817** | 0.3872 | **0.3530** | 0.3804 |
| | 192 | **0.4189** | 0.4238 | **0.4189** | 0.4238 | **0.4087** | 0.4159 | **0.4184** | 0.4203 | **0.4252** | 0.4266 | **0.4140** | 0.4171 | 0.4370 | **0.4367** | 0.4405 | **0.4403** | **0.4213** | 0.4269 |
| | 336 | **0.4800** | 0.4833 | **0.4788** | 0.4822 | **0.4781** | 0.4820 | **0.4811** | 0.4827 | **0.4896** | 0.4909 | **0.4794** | 0.4847 | **0.4864** | 0.4867 | **0.4970** | 0.4975 | **0.4787** | 0.4805 |
| | 720 | **0.5579** | 0.5595 | **0.5615** | 0.5622 | **0.5460** | 0.5506 | **0.5515** | 0.5524 | **0.5594** | 0.5603 | **0.5497** | 0.5535 | 0.5413 | **0.5402** | **0.5479** | 0.5485 | **0.5399** | 0.5420 |
| ETTm2 | 96 | **0.1640** | 0.1642 | **0.1640** | 0.1642 | 0.1613 | **0.1606** | **0.1594** | 0.1595 | **0.1595** | 0.1595 | **0.1563** | 0.1573 | 0.1569 | 0.1571 | **0.1569** | 0.1573 | **0.1558** | 0.1562 |
| | 192 | 0.2164 | **0.2163** | **0.2157** | 0.2164 | **0.2067** | 0.2094 | **0.1924** | 0.1925 | **0.1925** | 0.1927 | **0.1897** | 0.1907 | **0.2058** | 0.2063 | **0.2079** | 0.2084 | **0.1976** | 0.2018 |
| | 336 | **0.2735** | 0.2758 | **0.2653** | 0.2748 | **0.2623** | 0.2700 | **0.2308** | 0.2309 | **0.2316** | 0.2318 | **0.2289** | 0.2293 | **0.2482** | 0.2482 | **0.2473** | 0.2476 | **0.2422** | 0.2458 |
| | 720 | 0.3303 | **0.3295** | **0.3263** | 0.3264 | 0.3290 | **0.3251** | **0.2996** | 0.2996 | **0.3003** | 0.3003 | **0.2973** | 0.2985 | **0.3218** | 0.3223 | **0.3240** | 0.3242 | **0.3192** | 0.3196 |
| Weather | 96 | **0.1662** | 0.1674 | **0.1662** | 0.1686 | **0.1605** | 0.1635 | **0.1825** | 0.1855 | **0.1890** | 0.1914 | **0.1728** | 0.1843 | **0.1700** | 0.1713 | **0.1676** | 0.1721 | **0.1635** | 0.1682 |
| | 192 | **0.2129** | 0.2136 | **0.2165** | 0.2171 | **0.2103** | 0.2113 | **0.2238** | 0.2255 | **0.2293** | 0.2304 | **0.2171** | 0.2249 | **0.2119** | 0.2130 | **0.2120** | 0.2137 | **0.2050** | 0.2088 |
| | 336 | **0.2629** | 0.2632 | **0.2672** | 0.2675 | 0.2618 | **0.2599** | **0.2729** | 0.2740 | **0.2785** | 0.2785 | **0.2665** | 0.2721 | **0.2659** | 0.2667 | **0.2703** | 0.2716 | **0.2570** | 0.2614 |
| | 720 | **0.3418** | 0.3422 | **0.3470** | 0.3474 | **0.3357** | 0.3365 | **0.3467** | 0.3475 | **0.3598** | 0.3604 | **0.3401** | 0.3424 | 0.3356 | **0.3348** | **0.3529** | 0.3543 | **0.3341** | 0.3357 |
| Exchange | 96 | **0.0873** | 0.0877 | **0.0881** | 0.0881 | 0.0867 | **0.0863** | **0.0799** | 0.0805 | **0.0813** | 0.0813 | **0.0793** | 0.0794 | **0.0816** | 0.0825 | **0.0850** | 0.0853 | **0.0813** | 0.0821 |
| | 192 | **0.1800** | 0.1820 | **0.1820** | 0.1836 | 0.1773 | **0.1756** | **0.1650** | 0.1669 | **0.1719** | 0.1722 | **0.1648** | 0.1677 | 0.1666 | **0.1659** | **0.1765** | 0.1769 | **0.1692** | 0.1718 |
| | 336 | **0.3014** | 0.3023 | **0.3105** | 0.3110 | **0.2983** | 0.3026 | 0.2846 | **0.2823** | **0.3058** | 0.3093 | **0.2924** | 0.3009 | 0.3048 | **0.2763** | **0.3188** | 0.3261 | **0.3011** | 0.3149 |

Table A.16: Trainable parameter counts for input-output and output-only calibration. For each method, "In+Out" denotes the standard input-output variant and "Out" denotes the output-only variant. The ETT datasets share the same parameter counts. For TAFAS, the PatchTST-specific implementation is listed separately.

| Dataset | H | TAFAS | | TAFAS (PatchTST) | | PETSA | | FAC (Ours) | |
|---|---|---|---|---|---|---|---|---|---|
| | | In+Out | Out | In+Out | Out | In+Out | Out | In+Out | Out |
| **ETT (h1/2, m1/2)** | 96 | 130,382 | 65,191 | 18,626 | 9,313 | 25,934 | 12,967 | **2,758** | **1,379** |
| | 192 | 324,590 | 259,399 | 46,370 | 37,057 | 38,894 | 25,927 | **4,102** | **2,723** |
| | 336 | 857,822 | 792,631 | 122,546 | 113,233 | 58,334 | 45,367 | **6,118** | **4,739** |
| | 720 | 3,699,038 | 3,633,847 | 528,434 | 519,121 | 110,174 | 97,207 | **11,494** | **10,115** |
| **Weather** | 96 | 391,146 | 195,573 | 18,626 | 9,313 | 71,658 | 35,829 | **8,274** | **4,137** |
| | 192 | 973,770 | 778,197 | 46,370 | 37,057 | 107,466 | 71,637 | **12,306** | **8,169** |
| | 336 | 2,573,466 | 2,377,893 | 122,546 | 113,233 | 161,178 | 125,349 | **18,354** | **14,217** |
| | 720 | 11,097,114 | 10,901,541 | 528,434 | 519,121 | 304,410 | 268,581 | **34,482** | **30,345** |
| **Exchange** | 96 | 149,008 | 74,504 | 18,626 | 9,313 | 29,200 | 14,600 | **3,152** | **1,576** |
| | 192 | 370,960 | 296,456 | 46,370 | 37,057 | 43,792 | 29,192 | **4,688** | **3,112** |
| | 336 | 980,368 | 905,864 | 122,546 | 113,233 | 65,680 | 51,080 | **6,992** | **5,416** |

Table A.16 reports the corresponding trainable parameter counts. For each dataset and method, the parameter difference between the input-output and output-only variants corresponds to the input adapter. Since the parameter count of the input adapter is determined by the look-back window rather than the prediction horizon, its relative cost becomes smaller as $H$ increases.

## A.6 Runtime and Parameter Efficiency

Table A.17: Runtime and parameter comparison on Weather with the longest forecasting horizon $H = 720$, a relatively demanding setting in our benchmark. Adaptation time, reported in milliseconds, includes calibration, loss computation, backward update, and period estimation, excluding the source forecaster forward pass. Overall runtime is reported in seconds. Results are averaged over 10 repeated runs.

| Source Forecaster | Method | Trainable Parameters | MSE | Adaptation Time (ms) | Overall Runtime (s) |
|---|---|---|---|---|---|
| DLinear | TAFAS | 11,097,114 | 0.3509 | 3.60 ± 0.25 | 3.49 ± 0.14 |
| | PETSA | 304,410 | 0.3557 | 6.88 ± 0.63 | 3.85 ± 0.23 |
| | FAC (Ours) | **34,482** | **0.3426** | 4.25 ± 0.41 | 3.55 ± 0.15 |
| FreTS | TAFAS | 11,097,114 | 0.3423 | 5.81 ± 0.18 ms | 7.57 ± 0.18 s |
| | PETSA | 304,410 | 0.3550 | 21.35 ± 0.56 ms | 8.38 ± 0.24 s |
| | FAC (Ours) | **34,482** | **0.3376** | 6.01 ± 0.39 ms | 7.27 ± 0.24 s |
| iTransformer | TAFAS | 11,097,114 | 0.3418 | 7.29 ± 1.15 ms | 5.56 ± 0.12 s |
| | PETSA | 304,410 | 0.3470 | 10.32 ± 0.58 ms | 6.67 ± 1.50 s |
| | FAC (Ours) | **34,482** | **0.3357** | 7.92 ± 1.35 ms | 5.56 ± 0.22 s |
| OLS | TAFAS | 11,097,114 | 0.3467 | 3.10 ± 0.31 ms | 3.67 ± 0.29 s |
| | PETSA | 304,410 | 0.3598 | 6.79 ± 0.46 ms | 4.07 ± 0.15 s |
| | FAC (Ours) | **34,482** | **0.3401** | 3.54 ± 0.28 ms | 3.79 ± 0.23 s |
| PatchTST | TAFAS (PatchTST) | 528,434 | 0.3356 | 2.97 ± 0.12 ms | 20.76 ± 0.29 s |
| | PETSA | 304,410 | 0.3529 | 95.70 ± 1.98 ms | 27.78 ± 0.40 s |
| | FAC (Ours) | **34,482** | **0.3341** | 18.16 ± 0.29 ms | 27.21 ± 0.20 s |

*Note.* TAFAS and PETSA are evaluated under the same protocol as FAC, where adaptation uses only matured ground truth. For PatchTST, TAFAS follows its original backbone-specific implementation, which uses single-channel output calibration and bypasses input calibration. For frequency-domain calibration methods, including FAC and our PETSA variant, we use multivariate calibration across variables.

Runtime efficiency is important in practice, but reducing the number of trainable parameters does not necessarily yield proportional wall-clock speedups. As shown in Table A.17, TAFAS generally has lower runtime overhead, whereas FAC achieves frequency-aware calibration with a substantially smaller number of trainable parameters. The additional Fourier-transform operations and tensor transformations used by FAC likely contribute to its higher runtime overhead in some settings. Compared with PETSA, FAC uses a simpler adaptation objective and achieves better forecasting accuracy with substantially fewer trainable parameters and lower runtime in most settings. The runtime difference relative to TAFAS is most pronounced for PatchTST, where TAFAS uses its lightweight backbone-specific implementation.

## A.7 Additional Frequency-Domain Correction Spectra

We provide additional frequency-domain correction spectra across datasets and forecasting horizons to complement the analysis in Figure 4. Recall that these spectra are not intended to measure forecasting error. Instead, they characterize the magnitude and structure of the prediction changes induced by adaptation in the frequency domain.

For each adaptation method $q \in \{\text{FAC}, \text{TAFAS}, \text{PETSA}\}$, we first compute the correction for each collected prediction window. Specifically, we denote the correction as

$$\Delta \hat{\mathbf{Y}}^{q,[n]} = \hat{\mathbf{Y}}^{q,[n]} - \hat{\mathbf{Y}}^{[n]}, \qquad n = 1, \dots, N,$$

where $\hat{\mathbf{Y}}^{[n]}$ denotes the prediction before adaptation and $\hat{\mathbf{Y}}^{q,[n]}$ denotes the prediction after applying method $q$ to the $n$-th prediction window. We then concatenate the corrections from all prediction windows along the prediction-window dimension, yielding

$$\Delta \hat{\mathbf{Y}}^q \in \mathbb{R}^{N \times H \times C},$$

where $N$ is the total number of prediction windows, $H$ is the forecasting horizon, and $C$ is the number of channels. Next, we apply rFFT along the forecasting-horizon dimension and take the magnitude as

$$\mathbf{A}^q = \left| \text{rFFT}_H \left( \Delta \hat{\mathbf{Y}}^q \right) \right| \in \mathbb{R}^{N \times F \times C},$$

where $F = \lfloor H/2 \rfloor + 1$ is the number of rFFT frequency components. The plotted curve is obtained by averaging over the prediction windows and channels as

$$\bar{\mathbf{a}}^q = \text{Avg}_{N,C} \left( \mathbf{A}^q_{f>0} \right) \in \mathbb{R}^{F-1}.$$

Here, $f > 0$ indicates that the DC component is omitted.

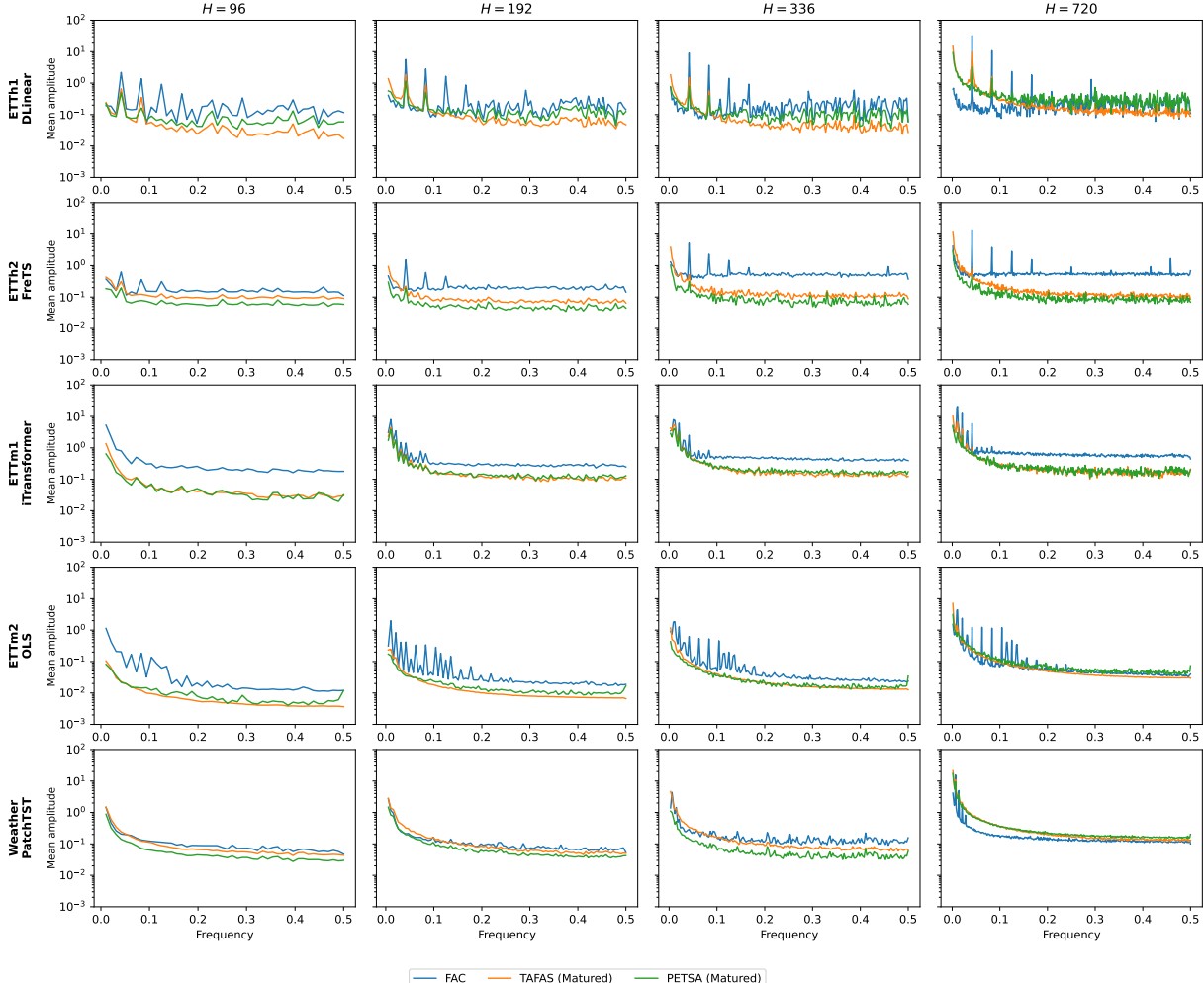

Figure A.2: Additional frequency-domain correction spectra across forecasting horizons. For each adaptation method, we compute the correction, apply rFFT to this correction along the forecasting horizon, and plot the spectral magnitude averaged over prediction windows and channels. The DC component is omitted. Columns correspond to $H \in \{96, 192, 336, 720\}$, and rows correspond to dataset–forecaster pairs ordered alphabetically by dataset name. The y-axis is shown on a logarithmic scale. We omit Exchange because no matured mini-batch is available under the $H = 720$ setting in our protocol.

Figure A.2 shows that the qualitative behavior observed in the main text is not limited to a single forecasting horizon. Across diverse settings, FAC produces more localized spectral corrections than TAFAS and PETSA, whose corrections often vary more smoothly across frequencies. At the same time, some cases, such as Weather with PatchTST, show smaller differences among the methods, indicating that the spectral behavior also depends on the source forecaster and dataset.

### A.8  Additive Frequency-Band Decomposition of Residual-Error Reduction

Using the frequency-band partition and residual spectral-energy definition introduced in the main text, we further decompose the total residual-error reduction into additive bandwise contributions. For adaptation method $q \in \{\text{FAC}, \text{TAFAS}, \text{PETSA}\}$, we define

$$\Delta_b^q = 100 \times \frac{E_b(\hat{\mathbf{Y}}) - E_b(\hat{\mathbf{Y}}^q)}{\sum_{b'} E_{b'}(\hat{\mathbf{Y}})}, \tag{20}$$

where all frequency bands share the same denominator. Consequently, their contributions are additive:

$$\Delta_{\text{total}}^q = \sum_{b \in \{\text{low}, \text{mid}, \text{high}\}} \Delta_b^q. \tag{21}$$

Table A.18: Additive decomposition of total residual-error reduction across frequency bands for representative settings at $H = 720$. Each band reports its percentage-point contribution to the total reduction in residual spectral energy. The three bandwise contributions sum to the Total value up to rounding. Within each setting, the largest value in each column is highlighted in bold.

| Setting | Method | Low | Middle | High | Total |
|---|---|---|---|---|---|
| ETTh1 + DLinear | TAFAS | 2.6809 | 0.0061 | 0.0050 | 2.6920 |
| | PETSA | 2.4999 | 0.0125 | 0.0114 | 2.5238 |
| | FAC | **3.0137** | **0.1470** | **0.0222** | **3.1829** |
| ETTh2 + FreTS | TAFAS | 7.3853 | 0.0041 | 0.0022 | 7.3916 |
| | PETSA | 3.6529 | 0.0029 | 0.0015 | 3.6572 |
| | FAC | **7.7298** | **0.1262** | **0.1591** | **8.0151** |
| ETTm1 + iTransformer | TAFAS | 7.3295 | 0.0106 | 0.0096 | 7.3496 |
| | PETSA | 6.6962 | 0.0063 | 0.0060 | 6.7085 |
| | FAC | **9.2175** | **0.0672** | **0.0750** | **9.3597** |
| ETTm2 + OLS | TAFAS | 2.2883 | **0.0010** | **0.0005** | 2.2897 |
| | PETSA | 2.0576 | 0.0006 | 0.0001 | 2.0582 |
| | FAC | **3.0302** | 0.0004 | 0.0002 | **3.0308** |

Table A.18 confirms that most of the absolute error reduction arises from the low-frequency band, where the residual spectral energy is concentrated. FAC nevertheless contributes more middle- and high-frequency reduction in the first three settings, whereas the improvement for ETTm2 with OLS is almost entirely low-frequency.

### A.9  Comparison with a Temporal-Domain Counterpart

To isolate the effect of frequency-domain parameterization, we construct a temporal-domain counterpart that preserves the element-wise affine and gated residual structure of FAC. On the input side, it computes

$$\Delta \mathbf{X}_k^{[j]} = \mathbf{X}_k^{[j]} \odot \mathbf{W}^{\text{in}} + \mathbf{B}^{\text{in}}, \qquad \mathbf{X}_{\text{temp}, k}^{[j]} = \mathbf{X}_k^{[j]} + \tanh(\boldsymbol{\alpha}^{\text{in}}) \odot \Delta \mathbf{X}_k^{[j]}. \tag{22}$$

Here, $\mathbf{W}^{\text{in}}, \mathbf{B}^{\text{in}} \in \mathbb{R}^{L \times C}$ and $\boldsymbol{\alpha}^{\text{in}} \in \mathbb{R}^C$. The output adapter follows the same form over the forecasting horizon, with affine parameters in $\mathbb{R}^{H \times C}$. Unlike FAC, which applies its affine transformation to Fourier coefficients, the temporal counterpart applies the corresponding transformation directly to individual time positions. The two designs have closely matched trainable parameter counts: the temporal adapter uses real-valued affine coefficients over all time positions, whereas FAC uses complex-valued affine coefficients over the approximately half-sized one-sided frequency representation. Both adapters are optimized using the same matured-only protocol and MSE objective, leaving the calibration domain as the primary design difference.

The temporal adapter improves upon the source model in 113 of the 115 settings, indicating that it is itself an effective adaptation baseline. Nevertheless, as shown in Table A.19, FAC achieves lower MSE in all 115

settings. This consistent advantage supports direct frequency-domain parameterization over a comparably sized temporal-domain alternative.

Table A.19: Full FAC versus a comparably sized temporal-domain affine counterpart across datasets and source forecasters. "Temporal" denotes the temporal-domain counterpart. Lower MSE is highlighted within each pair.

| Dataset | Horizon | DLinear | | FreTS | | iTransformer | | OLS | | PatchTST | |
|---|---|---|---|---|---|---|---|---|---|---|---|
| | | FAC | Temporal | FAC | Temporal | FAC | Temporal | FAC | Temporal | FAC | Temporal |
| ETTh1 | 96 | **0.4554** | 0.4691 | **0.4357** | 0.4455 | **0.4400** | 0.4463 | **0.4382** | 0.4501 | **0.4241** | 0.4319 |
| | 192 | **0.5089** | 0.5210 | **0.4934** | 0.5017 | **0.5010** | 0.5078 | **0.4921** | 0.5040 | **0.4797** | 0.4883 |
| | 336 | **0.5582** | 0.5658 | **0.5498** | 0.5543 | **0.5587** | 0.5659 | **0.5422** | 0.5509 | **0.5525** | 0.5605 |
| | 720 | **0.6891** | 0.7085 | **0.7067** | 0.7148 | **0.6687** | 0.6985 | **0.6780** | 0.6971 | **0.7096** | 0.7108 |
| ETTh2 | 96 | **0.2288** | 0.2318 | **0.2338** | 0.2377 | **0.2561** | 0.2629 | **0.2274** | 0.2300 | **0.2374** | 0.2384 |
| | 192 | **0.2819** | 0.2835 | **0.2814** | 0.2860 | **0.2970** | 0.3053 | **0.2796** | 0.2834 | **0.2747** | 0.2830 |
| | 336 | **0.3144** | 0.3234 | **0.3172** | 0.3302 | **0.3284** | 0.3376 | **0.3137** | 0.3253 | **0.3079** | 0.3220 |
| | 720 | **0.3825** | 0.4043 | **0.3788** | 0.4099 | **0.3982** | 0.4218 | **0.3854** | 0.4107 | **0.4002** | 0.4272 |
| ETTm1 | 96 | **0.3465** | 0.3681 | **0.3517** | 0.3638 | **0.3459** | 0.3743 | **0.3462** | 0.3713 | **0.3530** | 0.3897 |
| | 192 | **0.4139** | 0.4423 | **0.4183** | 0.4302 | **0.4087** | 0.4309 | **0.4140** | 0.4436 | **0.4213** | 0.4504 |
| | 336 | **0.4791** | 0.5168 | **0.4836** | 0.4982 | **0.4781** | 0.5014 | **0.4794** | 0.5166 | **0.4787** | 0.5039 |
| | 720 | **0.5494** | 0.5921 | **0.5533** | 0.5686 | **0.5460** | 0.5953 | **0.5497** | 0.5903 | **0.5399** | 0.5597 |
| ETTm2 | 96 | **0.1560** | 0.1590 | **0.1563** | 0.1575 | **0.1613** | 0.1642 | **0.1563** | 0.1599 | **0.1558** | 0.1577 |
| | 192 | **0.1894** | 0.1925 | **0.1901** | 0.1918 | **0.2067** | 0.2172 | **0.1897** | 0.1933 | **0.1976** | 0.2092 |
| | 336 | **0.2286** | 0.2320 | **0.2294** | 0.2319 | **0.2623** | 0.2759 | **0.2289** | 0.2328 | **0.2422** | 0.2480 |
| | 720 | **0.2984** | 0.3023 | **0.2955** | 0.3011 | **0.3290** | 0.3386 | **0.2973** | 0.3055 | **0.3192** | 0.3280 |
| Weather | 96 | **0.1797** | 0.1853 | **0.1678** | 0.1801 | **0.1605** | 0.1678 | **0.1728** | 0.1904 | **0.1635** | 0.1695 |
| | 192 | **0.2231** | 0.2331 | **0.2114** | 0.2269 | **0.2103** | 0.2188 | **0.2171** | 0.2336 | **0.2050** | 0.2150 |
| | 336 | **0.2717** | 0.2820 | **0.2621** | 0.2798 | **0.2618** | 0.2734 | **0.2665** | 0.2868 | **0.2570** | 0.2756 |
| | 720 | **0.3426** | 0.3607 | **0.3376** | 0.3594 | **0.3357** | 0.3547 | **0.3401** | 0.3639 | **0.3341** | 0.3459 |
| Exchange | 96 | **0.0875** | 0.0909 | **0.0810** | 0.0822 | **0.0867** | 0.0881 | **0.0793** | 0.0812 | **0.0813** | 0.0846 |
| | 192 | **0.1758** | 0.1821 | **0.1667** | 0.1726 | **0.1773** | 0.1834 | **0.1648** | 0.1720 | **0.1692** | 0.1759 |
| | 336 | **0.2985** | 0.3297 | **0.3166** | 0.3237 | **0.2983** | 0.3149 | **0.2924** | 0.3143 | **0.3011** | 0.3203 |

# B Protocol Compatibility under Matured-Only Supervision

## B.1 Variants of TAFAS and PETSA under the Matured-Only Protocol

The original implementations of TAFAS and PETSA already separate adaptation using the most recent matured mini-batch from adaptation using POGT from the current mini-batch. Once a historical mini-batch matures, it is replayed and used to update the original calibration modules against its complete target horizon. Our matured-only variants therefore retain this existing matured-ground-truth branch while disabling the current-mini-batch POGT update branch. Using the notation of Section 3.2.1, for $q \in \{\text{TAFAS}, \text{PETSA}\}$, the retained adaptation objective is

$$\mathcal{L}_k^{q,\text{mat}} = \mathcal{L}_{m(k)}^{q,\text{Matured}}. \tag{23}$$

Here, $\mathcal{L}_{m(k)}^{q,\text{Matured}}$ denotes the loss used by each method on a fully matured mini-batch: MSE for TAFAS and the composite adaptation objective for PETSA. After updating the calibration modules, the complete current mini-batch is re-forecast and evaluated as

$$\tilde{\mathbf{Y}}_k^{q,[j]} = \hat{\mathbf{Y}}_k^{q,\text{adp},[j]} = \mathcal{C}_k^{q,\text{out}}\left(\mathcal{F}\left(\mathcal{C}_k^{q,\text{in}}(\mathbf{X}_k^{[j]})\right)\right), \qquad j = 1, \dots, B_k. \tag{24}$$

This replaces the POGT-dependent partial stitching rule in Eq. 8. Thus, the conversion from the mixed protocol to the proposed matured-only protocol is direct: it disables POGT supervision and its associated partial stitching, but does not redefine the calibration architecture, adapter input or trainable parameters.

## B.2 Protocol Compatibility of COSA with Matured-Only Supervision

As discussed in Section 3.2.2, the streaming protocol implemented by COSA does not generally satisfy the matured-only criterion considered in this work. In particular, the target statistics entering its context are revealed but not necessarily matured relative to subsequent rolling predictions. When the forecasting horizon is large relative to the mini-batch size, statistics from a recently processed mini-batch may already enter the context of later predictions while its target span still overlaps with those prediction spans.

Furthermore, unlike TAFAS and PETSA, COSA uses target-derived information in two places: the complete targets provide adaptation supervision, while statistics computed from recently observed targets form part of the adapter input. Following the notation of COSA, let $\mathbf{Y}_t^{(0)}$ denote the frozen source-forecaster prediction at time $t$, and let $\mathbf{C}_t$ denote the context vector constructed from recently observed target statistics. COSA forms the adapter input

$$\mathbf{X}_t^{(a)} = [\mathbf{Y}_t^{(0)} \| \mathbf{C}_t], \tag{25}$$

and produces the corrected prediction as

$$\hat{\mathbf{Y}}_t = \mathbf{Y}_t^{(0)} + \tanh(g)\left(\mathbf{W}\mathbf{X}_t^{(a)} + \mathbf{b}\right). \tag{26}$$

In the official implementation, the predictions for the current mini-batch are first corrected using a context $\mathbf{C}_t^{\mathrm{pred}}$ constructed from the existing target-statistic memory. The target statistic computed from the complete target tensor of that mini-batch is then appended to the memory, producing an updated context $\mathbf{C}_t^{\mathrm{upd}}$, and the adapter is optimized on the same mini-batch. Hence, the adapter inputs used for prediction and adaptation are respectively

$$\mathbf{X}_t^{(a),\mathrm{pred}} = [\mathbf{Y}_t^{(0)} \| \mathbf{C}_t^{\mathrm{pred}}], \qquad \mathbf{X}_t^{(a),\mathrm{upd}} = [\mathbf{Y}_t^{(0)} \| \mathbf{C}_t^{\mathrm{upd}}], \tag{27}$$

where, in general, $\mathbf{C}_t^{\mathrm{pred}} \neq \mathbf{C}_t^{\mathrm{upd}}$. Consequently, the official implementation does not enforce the full-horizon maturation condition used in our protocol. A strict matured-only conversion would require delaying both the adapter update and the incorporation of target-derived statistics into the context, rather than simply disabling an existing update branch. We therefore do not include a matured-only COSA variant.

## C Target-Supervision Terminology and Notation

Table C.1: Summary of target-supervision terminology and notation.

| Term | Definition and observability | Notation | Methods |
|---|---|---|---|
| Revealed targets | Targets that become observable after prediction. | — | General TSF-TTA |
| POGT | The observed prefix of the current mini-batch targets before their complete horizons become observable. | $\mathcal{L}_k^{\mathrm{POGT}}$ | TAFAS, PETSA, DynaTTA |
| Matured ground truth | Targets from a past mini-batch whose complete horizons have been observed, satisfying $t_m + B_m + H - 1 \leq t_k$. | $\mathcal{L}_{m(k)}^{\mathrm{Matured}}$ | TAFAS, PETSA, DynaTTA, FAC |

*Note.* "Revealed values" and "revealed ground truth" are descriptive phrases for revealed targets and do not denote separate supervision categories.

