# OpenReview forum: "Towards Principled Test-Time Adaptation for Time Series Forecasting"
_TMLR — Under review for TMLR_

### Review · Reviewer_pW25 · 2026-06-30

**Summary Of Contributions:**

This paper studies test-time adaptation (TTA) for time series forecasting (TSF) under distribution shift. It makes three contributions:

- Protocol analysis. The authors critically re-examine how existing TSF-TTA methods (TAFAS, PETSA, COSA) use revealed targets for adaptation supervision. They identify that the mixed-supervision protocol (combining POGT and matured ground truth) and the streaming adaptation protocol both suffer from protocol-level ambiguity regarding the timing and role of supervision. They advocate for a cleaner adaptation protocol that restricts supervision to matured ground truth only, and provide empirical evidence (Figure 2, Table 4) that POGT-based supervision is not consistently more beneficial than simply using revealed values as input context for later direct predictions.

- Frequency-domain diagnosis. Under the proposed matured-only protocol, the authors analyze the spectral structure of prediction corrections induced by existing adapters (TAFAS, PETSA). They observe that these corrections tend to be smooth and weakly structured across frequencies, suggesting that existing parameterizations may not fully exploit frequency-selective adaptation.

- Frequency-Aware Calibration (FAC). Motivated by the above observation, FAC directly parameterizes prediction corrections in the frequency domain via element-wise complex affine masks (FreqGCMs). The design is lightweight, with parameter count scaling as O(C(L+H)), substantially fewer than TAFAS and PETSA. Across six benchmark datasets, five source forecasters, and multiple horizons, FAC achieves competitive or superior forecasting performance under the proposed protocol.

Key strengths:

- The protocol-level analysis is a timely and valuable contribution that clarifies the supervision assumptions underlying TSF-TTA.

- The frequency-domain diagnostic perspective provides useful insights.

- FAC is elegant in its simplicity and achieves impressive parameter efficiency.

Key weaknesses:

- The novelty of the protocol contribution is somewhat incremental—it is primarily a simplification rather than a fundamentally new framework.

- The frequency-domain analysis is largely descriptive without theoretical grounding for why frequency-domain parameterization should be preferred.

- The experimental evaluation, while extensive, is limited to standard benchmarks and lacks comparison with several recent baselines. Some empirical claims (e.g., FAC consistently outperforms) would benefit from statistical significance testing.

**Audience:**

Yes

**Audience Explanation:**

Test-time adaptation for time series forecasting is an active and growing research area, and this paper addresses a genuine need for clearer protocol definitions. The protocol-level analysis and the matured-GT-only formulation provide a useful reference point for future TSF-TTA research. The frequency-domain diagnostic perspective, while not deeply novel, offers a complementary lens for understanding adaptation behavior that is not commonly found in the TSF-TTA literature. The proposed FAC method, with its strong parameter efficiency, may be of practical interest for deployment scenarios where adapter size is constrained. Overall, the paper contributes both conceptual clarity and a practical method to this subfield.

**Broader Impact Concerns:**

Nope.

**Claims And Evidence:**

Yes

**Claims Explanation:**

The main empirical claims are generally well-supported. Table 1 presents comprehensive results across diverse settings, and the conclusion that FAC achieves competitive performance under the matured-only protocol is backed by the data. The parameter efficiency claim (Table 2) is clearly demonstrated. However, there are several concerns:

- No statistical significance analysis. All results report single-point MSE values without confidence intervals, standard deviations, or significance tests across multiple runs. Given that the improvements of FAC over TAFAS and PETSA are often marginal (e.g., ETTh2 H=96: FAC 0.2288 vs. PETSA 0.2312 on DLinear, a difference of ~1%), it is difficult to assess whether these differences are statistically meaningful or within noise.

- The frequency-domain correction analysis (Figure 4) is descriptive but not explanatory. The authors observe that FAC produces more "localized peaks" while TAFAS/PETSA produce "smooth corrections." While this is presented as evidence of frequency-selective adaptation, the paper does not explain why localized spectral corrections should lead to better forecasting accuracy, nor does it establish a causal link between the spectral structure of corrections and forecasting performance. The authors acknowledge this limitation ("a larger correction magnitude does not necessarily imply a more accurate forecast"), but the overall narrative implicitly suggests that frequency-selective corrections are desirable.

- The POGT analysis (Section 3.2.1, Figure 2) is insightful but somewhat one-sided. The comparison of overlapping-region MSE between adjusted predictions and direct predictions is informative, but it does not fully account for the potential benefit of POGT in enabling earlier adaptation triggers or providing richer gradient signals in the early stages of testing when few matured mini-batches are available.

- COSA is discussed but not compared empirically. COSA (Im & Kwon, 2026) is presented as a competing method with a different protocol, but it is not included in the experimental comparison. While the authors may argue that COSA operates under a different protocol, at least a discussion of why a direct comparison is infeasible—or a best-effort adaptation of COSA to the matured-only protocol—would strengthen the evaluation.

**Requested Changes:**

Critical changes:

- Statistical significance analysis. Please report confidence intervals or standard deviations across multiple runs (e.g., different random seeds or data splits) for the main results in Table 1. Given that many improvements are marginal, significance testing (e.g., paired t-test or Wilcoxon signed-rank test across datasets/horizons) is essential to substantiate the claim that FAC "achieves competitive and consistent performance."


- Include COSA as a baseline, or provide a principled justification for its exclusion. COSA is discussed extensively in Sections 2 and 3.2.2 as a competing method, yet it is absent from the experimental comparison. Either adapt COSA to the matured-only protocol and include it in Table 1, or provide a detailed explanation of why such an adaptation is not feasible and discuss COSA's expected behavior under the proposed protocol.


- Strengthen the connection between frequency-domain corrections and forecasting accuracy. The paper demonstrates that FAC produces more localized spectral corrections, but does not establish that this property is causally linked to improved forecasting. Consider adding an analysis that correlates the structure of frequency-domain corrections with per-frequency forecasting error reduction (e.g., decompose the MSE improvement by frequency band to show that FAC's corrections are indeed targeting the right frequency components).

Important changes (would strengthen the paper):

- Broader dataset coverage. The evaluation is limited to six standard benchmarks (ETT family, Weather, Exchange), which are all relatively well-behaved and stationary-ish after standardization. Including datasets with more severe or abrupt distribution shifts (e.g., electricity, traffic with known regime changes, or synthetic datasets with controlled shift types) would better demonstrate the value of the proposed protocol and method under challenging conditions.


- Comparison with more recent online forecasting methods. While the paper distinguishes TSF-TTA from online time series forecasting (OTSF), some recent OTSF methods are closely related. A brief empirical comparison or at least a discussion of how FAC's matured-only protocol relates to the "clean" evaluation protocols advocated by Act-Now and DSOF would position the contribution more clearly.


- Runtime discussion in the main text. The paper acknowledges in the limitations section and Appendix A.4 that FAC's parameter efficiency does not always translate to wall-clock speedups. Given that Table A.4 shows FAC sometimes has higher adaptation time than TAFAS (e.g., 18.16ms vs. 2.97ms for PatchTST on Weather), this should be discussed more prominently in the main text rather than deferred to the appendix. The practical implications of FFT/iFFT overhead should be addressed.


- Clarify the TAFAS implementation for PatchTST. The paper notes that TAFAS uses a PatchTST-specific implementation with single-channel calibration and no input calibration (Table 1 footnote, Table A.3). This means the comparison between FAC and TAFAS on PatchTST is not entirely apples-to-apples, as FAC uses multivariate calibration with both input and output modules. Please either provide a FAC variant with PatchTST-specific calibration for fair comparison, or discuss this asymmetry explicitly.


Theoretical or intuitive justification for frequency-domain parameterization. While the empirical results support FAC's design, the paper would benefit from a more principled argument for why frequency-domain corrections are advantageous for TSF-TTA. For example, under what types of distribution shift (e.g., periodicity changes, amplitude shifts, phase drift) does frequency-domain calibration provably or intuitively outperform temporal-domain calibration? A brief theoretical discussion or illustrative synthetic experiment would strengthen the motivation.

---

> ### Author Response · Authors · 2026-07-19
> **Response to Reviewer pW25 (Part 1)**
>
> We sincerely thank the reviewer for the careful and constructive review, as well as for recognizing the value of our protocol-level analysis, frequency-domain diagnostic perspective, and parameter efficiency of FAC.
>
> The principal changes include: (i) five-seed experiments and paired significance tests; (ii) a detailed analysis of COSA under matured-only supervision; (iii) a frequency-band error decomposition relating spectral corrections to forecasting-error reduction; (iv) additional experiments on Electricity and Traffic; and (v) expanded discussions of online forecasting protocols, runtime, PatchTST-specific implementation differences, and the intuition behind frequency-domain calibration.
>
> ## Critical changes
>
> ### 1. Statistical significance analysis
>
> > Please report confidence intervals or standard deviations across multiple runs and perform significance tests because many improvements are marginal.
>
> **Response.** We agree that the original single-run results were insufficient to establish whether the relatively small improvements in some settings were statistically stable. We therefore repeated the TTA experiments with **five adaptation seeds** for each matched setting, while using the same pretrained source-forecaster checkpoint across seeds. This setup isolates the variation introduced during adaptation while holding the pretrained source forecaster constant.
>
> We now report complete setting-level MSE mean ± standard deviation results in **Appendix A.1 (Tables A.1–A.3)**. We additionally perform paired one-sided Wilcoxon signed-rank tests across matched source-forecaster and prediction-horizon settings, using the five-seed mean for each setting and full-precision values before rounding.
>
> The results provide the following evidence:
>
> (i) At the dataset level, the paired tests indicate that FAC yields statistically significant improvements over the frozen source forecaster and PETSA on all six primary datasets. Compared with TAFAS, the improvement is statistically significant on ETTh2, ETTm1, ETTm2, and Weather, but not on ETTh1 or Exchange.
>
> (ii) Across all 115 matched settings, FAC records **115/0/0 wins/ties/losses against the frozen source forecaster** ($p=6.56\times10^{-21}$), **96/3/16 against TAFAS** ($p=3.50\times10^{-11}$), and **109/0/6 against PETSA** ($p=9.43\times10^{-18}$).
>
> (iii) Run-to-run variation across the five adaptation seeds is small in most settings.
>
> We have also revised the main-text discussion to summarize these findings.
>
> **Location in the revised manuscript:** Section 5.2.1; Appendix A.1, Tables A.1–A.6.
>
> ---
>
> ### 2. COSA baseline and compatibility with matured-only supervision
>
> > Either adapt COSA to the matured-only protocol and include it, or provide a detailed explanation of why such an adaptation is not feasible and discuss its expected behavior.
>
> **Response.** We agree that the original brief footnote did not adequately justify COSA's exclusion. We have therefore added a detailed protocol-level analysis in **Appendix B.2**.
>
> TAFAS and PETSA already include a separate update based on fully matured targets, so their POGT updates can be removed without changing the adapter input. COSA differs because target-derived information is used both to supervise the adapter and to construct its input context:
>
> 1. complete targets supervise the adapter update; and
> 2. statistics computed from recently observed targets form part of the adapter input.
>
> In the released implementation, the current mini-batch is first corrected using a context constructed from the existing target-statistic memory. The mean of the complete target tensor for that mini-batch is then appended to memory, after which the context is recomputed and the adapter is optimized on the same mini-batch. Thus, the contexts used for prediction and adaptation generally differ, which we denote by $C_t^{\mathrm{pred}}$ and $C_t^{\mathrm{upd}}$.
>
> Consequently, the official implementation does not enforce the full-horizon maturation condition used in our protocol. A strict conversion would require that a mini-batch contribute to both the adapter update and the target-derived context only after its full target span has matured. It would also require specifying which context should be paired with that mini-batch when it is later used for adaptation. Since this behavior is not defined by the original method, COSA does not admit a unique direct conversion analogous to removing the POGT update from TAFAS or PETSA.
>
> Such a conversion would also remove COSA’s immediate use of newly observed target statistics. We would therefore expect it to respond less immediately to recent shifts, while its actual behavior would depend on how the delayed context is constructed. For these reasons, we do not report a newly designed variant under the name “matured-only COSA.”
>
> **Location in the revised manuscript:** Section 5.1 footnote; Appendix B.2.

---

> ### Author Response · Authors · 2026-07-19
> **Response to Reviewer pW25 (Part 2)**
>
> Continuing our response to the critical changes:
>
> ### 3. Connecting frequency-domain corrections to forecasting accuracy
>
> > The correction spectra are descriptive; please connect their structure to per-frequency forecasting-error reduction.
>
> **Response.** We agree that the correction magnitude alone does not establish improved forecasting accuracy. We have revised the text to explicitly distinguish two quantities:
>
> (i) the spectrum of the **adaptation-induced prediction correction**, which characterizes how an adapter changes a forecast; and
>
> (ii) the spectrum of the **forecasting residual**, which measures prediction error.
>
> We have also added a new frequency-band error analysis. Specifically, we partition the one-sided rFFT range into low-, middle-, and high-frequency bands and measure the reduction in residual energy within each band before and after adaptation. The new **Table 6** shows that FAC achieves the largest total residual-error reduction in all four representative settings. In the first three settings, FAC also produces larger middle- and high-frequency reductions, consistent with the localized correction spectra in Figure 4. The ETTm2+OLS setting exhibits a different pattern: all methods achieve little middle- or high-frequency error reduction, and the improvement is concentrated mainly in the low-frequency band.
>
> These results indicate an association between FAC's frequency-selective corrections and error reduction in the corresponding bands, although they do not by themselves establish causality. We additionally provide an additive decomposition of the total residual-error reduction in **Appendix A.8**.
>
> **Location in the revised manuscript:** Section 5.3.2, Figure 4 and Table 6; Appendix A.8.
>
> ---
>
> ## Additional concern raised in the evidence discussion
>
> ### 4. Potential benefits of POGT in early adaptation
>
> > The POGT analysis may be one-sided because POGT can enable earlier updates or provide additional gradient signals when few matured mini-batches are available.
>
> **Response.** We agree that POGT can provide more immediate supervision, particularly before sufficient matured mini-batches have become available. Our claim is not that POGT is never useful or that delayed supervision is universally preferable. The full comparison between original mixed-supervision TAFAS and matured-only TAFAS shows that the original protocol often performs slightly better, but POGT does not necessarily provide a consistent empirical advantage across datasets, prediction horizons, and source forecasters. Moreover, its use complicates the protocol because the same revealed values can serve both as adaptation supervision and as input context for later rolling predictions.
>
> To make this scope clearer, we have revised the opening of Section 4 to explicitly acknowledge that the matured-only protocol removes the timely feedback provided by POGT within the current mini-batch and instead relies on delayed supervision from matured mini-batches. We therefore present matured-only supervision as a cleaner reference protocol, rather than claiming that POGT has no potential benefit or that delayed supervision is universally superior.
>
> **Location in the revised manuscript:** Section 4 opening paragraph; Section 5.3.1 and Table 5; Appendix A.4.

---

> ### Author Response · Authors · 2026-07-19
> **Response to Reviewer pW25 (Part 3)**
>
> ## Important changes
>
> ### 5. Broader dataset coverage
>
> > Include datasets with more severe or abrupt shifts, such as Electricity or Traffic.
>
> **Response.** Thank you for this suggestion. We have expanded the evaluation to include **Electricity** and **Traffic**, which contain 321 and 862 variables, respectively, and represent the energy and transportation domains. We evaluate DLinear, FreTS, and iTransformer over $H\in\{96,192,336,720\}$, yielding **24 additional model–horizon settings**. Each completed adaptation setting is repeated with five random seeds.
>
> Across the 21 settings for which all methods completed successfully, FAC outperforms the frozen source forecaster and TAFAS in **19/21** settings and PETSA in **21/21** settings. On the nine Traffic settings with $H\in\{96,192,336\}$, FAC outperforms all three alternatives in **9/9** settings. We report the three Traffic settings with $H=720$ separately because TAFAS exceeds the available GPU memory for FreTS and iTransformer. In these settings, FAC achieves the lowest available MSE for FreTS and iTransformer and ties PETSA for DLinear at the reported precision.
>
> These additional experiments broaden the evaluation to larger-scale multivariate datasets from two further application domains.
>
> **Location in the revised manuscript:** Sections 5.1 and 5.2.1, Table 2; Appendix A.2.
>
> ---
>
> ### 6. Relationship to recent online forecasting protocols
>
> > Compare with, or discuss the relationship to, the clean evaluation protocols advocated by Act-Now and DSOF.
>
> **Response.** Thank you for pointing out this connection. Section 2.1 discusses Act-Now and DSOF and positions these methods as related to TSF-TTA in spirit, while distinguishing their primary learning objectives. These studies and our work share a concern with the temporal validity of online supervision and evaluation, but the specific protocol issues are different. Act-Now and DSOF mainly address more direct forms of information leakage in OTSF, such as evaluating time steps that have already influenced model updates. Our analysis concerns related but distinct issues arising from overlapping long-horizon forecasts: targets used to update an adapter may still lie within subsequent prediction spans, while POGT used to adjust an earlier forecast later enters the look-back windows of subsequent forecasts. Our matured-only protocol addresses these issues by requiring the full target span to mature before it is used for adaptation.
>
> The learning and evaluation settings also differ substantially. Act-Now and DSOF continuously update online forecasting models, whereas our setting keeps the pretrained source forecaster frozen and updates only a lightweight adapter. Their experiments also use different forecasting horizons and online update procedures. For example, DSOF evaluates horizons in $\{1,24,48\}$, whereas our evaluation considers horizons up to 720. We therefore discuss their relationship to our protocol rather than include them as direct baselines.
>
> **Location in the revised manuscript:** Section 2.1.
>
> ---
>
> ### 7. Runtime and FFT/iFFT overhead
>
> > Runtime limitations should be discussed in the main text rather than only in the appendix.
>
> **Response.** We agree that parameter efficiency and wall-clock efficiency should be distinguished more prominently. We have therefore renamed Section 5.2.2 as **Parameter and Runtime Efficiency** and added a main-text discussion clarifying that FAC's substantially smaller number of trainable parameters does not always translate into lower wall-clock runtime under our current implementation. We also note that the additional rFFT/iFFT operations and tensor transformations required by the current implementation likely contribute to the observed runtime overhead in some settings.
>
> The detailed quantitative comparison remains in **Appendix A.6 (Table A.11)**, where adaptation time and total runtime are averaged over ten repeated runs. The difference is most pronounced for PatchTST: TAFAS uses its PatchTST-specific implementation with single-channel output calibration and bypasses input calibration, requiring 2.97 ms per adaptation step, whereas FAC retains both input and output FreqGCMs and requires 18.16 ms. We now state this architectural difference explicitly and clarify that fewer trainable parameters do not necessarily imply lower wall-clock runtime.
>
> We have also revised the limitations and future-work discussion to acknowledge this implementation-level overhead and the potential for reducing redundant tensor transformations and other implementation costs.
>
> **Location in the revised manuscript:** Section 5.2.2; Section 6; Appendix A.6, Table A.11.

---

> ### Author Response · Authors · 2026-07-19
> **Response to Reviewer pW25 (Part 4)**
>
> ### 8. PatchTST-specific TAFAS implementation
>
> > Clarify the asymmetry between TAFAS and FAC on PatchTST, or provide a corresponding FAC variant.
>
> **Response.** We agree that this implementation asymmetry should be stated explicitly. For PatchTST, we retain the official TAFAS-specific implementation, which uses single-channel output calibration and bypasses input calibration. FAC does not introduce a PatchTST-specific variant and uses the same input and output FreqGCM design as for the other source forecasters. The two adapter architectures are therefore not identical, although both are evaluated under the same matured-only supervision protocol. We have clarified this asymmetry in the note below the main results table.
>
> **Location in the revised manuscript:** Table 1 note.
>
> ---
>
> ### 9. Intuition for frequency-domain parameterization
>
> > Explain under what types of shift frequency-domain calibration should be advantageous.
>
> **Response.** We agree that the original manuscript did not sufficiently explain the intuition behind the frequency-domain parameterization. We have expanded Section 4 to clarify that FAC associates learnable complex coefficients with individual Fourier components, allowing the amplitude and phase of each component to be adjusted directly. Because each Fourier basis function spans the full temporal window, modifying a coefficient produces a coordinated correction across multiple temporal positions after the inverse transform. This provides a compact representation of structured full-window corrections without requiring a general dense transformation across temporal positions.
>
> Changes in the amplitude or phase of periodic components can be represented as changes in the magnitude or phase of their Fourier coefficients. FAC makes these adjustments explicit in its parameterization, whereas temporal-domain adapters realize them through temporal transformations.
>
> We also connect this intuition to the frequency-band error analysis. Table 6 reports within-band residual-error reduction rates and shows that FAC achieves larger middle- and high-frequency reduction rates in the first three representative settings, whereas the improvement for ETTm2 with OLS is concentrated mainly in the low-frequency band. The additive decomposition in Appendix A.8 further shows that most of the absolute error reduction arises from the low-frequency band in all four settings, while FAC contributes more middle- and high-frequency reduction in the first three. Together, these analyses show that FAC can reduce residual error across different frequency bands, while the extent of middle- and high-frequency reduction depends on the forecasting setting.
>
> **Location in the revised manuscript:** Section 4; Section 5.3.2 and Table 6; Appendix A.8.
>
> ---
>
> We again thank the reviewer for the detailed suggestions. They led us to strengthen both the empirical evidence and the scope of the discussion, and to state more clearly where the proposed protocol and FAC are useful as well as where their limitations remain.

---

> ### Author Response · Authors · 2026-07-22
> **Update to Appendix Table Numbering**
>
> We would like to note that, following additional revisions made in response to a subsequent review, the appendix table numbering has shifted. The runtime and parameter comparison previously cited in our response as Appendix A.6, Table A.11 is now reported in Appendix A.6, Table A.17. The underlying results and discussion remain unchanged. We apologize for any confusion this numbering change may cause.

---

### Review · Reviewer_zEPJ · 2026-07-03

**Summary Of Contributions:**

This paper studies test-time adaptation (TTA) for time series forecasting (TSF), where a frozen source forecaster is adapted at deployment using ground-truth targets that are revealed sequentially. The authors first examine how existing TSF-TTA methods use these revealed targets and observe that current methods are rather heterogeneous in how different kinds of data are merged. For instance: some methods combine partially-observed ground truth (POGT) from the current mini-batch with matured supervision from past mini-batches (past mini-batches whose entire forecasting horizon has since been fully observed), while streaming approaches update on targets revealed for the current mini-batch. Following this line of analysis, the authors observe that these approaches may draw supervision from signals that might be better treated as input context. The proposed approach leans towards adaptation using only matured data.

A second contribution of the paper is Frequency-Aware Calibration (FAC), a lightweight adapter that parameterizes prediction corrections directly in the frequency domain via complex affine masks with gated residual correction. The authors consider six datasets for experiments and show that the proposed approach outperforms the state of the art.

The main strength of the paper is the identification of the heterogeneous and sometimes unclear use of revealed data in the context of TTA. The authors provide numerical evidence (Table 4) that dropping partially-observed ground truth and adapting on only the most recent matured batch yields performance comparable to the mixed-supervision protocol — supporting their argument that POGT is not consistently necessary and that a matured-only protocol is a cleaner alternative rather than a more accurate one

The main weakness of the paper is the evaluation done in 6 datasets. The a limited amount of dataset for evaluations it is difficult to draw conclusion that extend to other scenario or data dynamics. There has been several efforts and studies suggesting that at a larger collection of datasets should be used to draw statistically meaningful conclusions.

**Audience:**

Yes

**Audience Explanation:**

The task of Test-Time Adaption is relevant as the development of pretrained time series forecasting models is getting a relevant amount of attention. Further, from the perspective of practitioners this is relevant as in some contexts it might be necessary to freeze forecasting models and still produce forecasts in a streaming fashion. Hence, the proposed approach provides an interesting analysis and insights on how to better leverage new data available without modifying the frozen forecasts.

**Broader Impact Concerns:**

No broader impact concerns from my side.

**Claims And Evidence:**

No

**Claims Explanation:**

The authors do provide an interesting an relevant analysis in terms of modeling, including the proposed novel approach based on the frequency domain. Yet, the main limitation in the current status of the paper is the empirical evidence based on the evaluation in a collection of six datasets. Whereas this collection of datasets has a widespread usage, there has been relevant efforts in the field of time series forecasting where larger and more extensive dataset collections are introduced. Some references are:
- GIFT-Eval: A Benchmark For General Time Series Forecasting Model Evaluation [https://arxiv.org/abs/2410.10393]
- fev-bench: A Realistic Benchmark for Time Series Forecasting [https://arxiv.org/html/2509.26468v1]
- TS-Arena -- A Live Forecast Pre-Registration Platform [https://arxiv.org/abs/2512.20761]

Critical studies on benchmarking in time series forecasting are:
- There are no Champions in Supervised Long-Term Time Series Forecasting [https://arxiv.org/pdf/2502.14045]
- Seeking SOTA: Time-Series Forecasting Must Adopt Taxonomy-Specific Evaluation to Dispel Illusory Gains [https://arxiv.org/pdf/2603.15506]

**Requested Changes:**

### Critical
The empirical evaluation needs to be extended to a larger collection of datasets. As mentioned in the previous section the authors can take as references benchmarking proposals as Gift-Eval, fev-bench, ts-arena, among others.

### Recommended
There are several parts in the paper where some gaps are left to the reader to be filled in. I enumerate some of them in what follows:
- State the meaning of "mini-batch" explicitly. Section 3.1 implies, but never states, that a mini-batch is a contiguous, temporally ordered block of windows from a single multivariate series. Because "mini-batch" usually connotes a shuffled training batch, one sentence clarifying this would be helpful for readers to have a concrete notion of this.
- Regarding Fig.1: It takes a fair amount of time for the reader to understand what is stated in this figure. This is particularly prominent given the notation used in it. For instance, $\tilde{y}^{[1]}_{t_k+1}$ is not defined before page 4. Up to page 4 what has been defined is in eq.1 and $y$ variables do not have tildes (~) but rather hats (^). Similarly $x$ variables do not have a super script
- Equation 6 and 7 are using MSE as loss functions. Does this mean that the proposed approach only works for point forecasts? What about probabilistic or quantile forecasting models?
- Equations 5,6,7 seem to be correct. I would suggest the authors to add a bit more of motivation of clarification on the meaning of these equations. In particular, after some thoughts, it seems that what the authors are considering for a loss function is that for the current batch $k$ most likely not all observations have landed, and hence the current batch is not mature, and hence some of the observations lack the observations to be properly evaluated. To provide a better estimation on the accuracy of the current batch, the evaluation is complemented by the most recently mature batch, which has all observations necessary to properly estimate the quality of the forecasts.
- It would be helpful if the authors provide more guidance on the dimensions and nuances of the notation used here. For instance, Equation 6 is applied on 2-Dimensional tensors (matrices)[shape ( B_k−1 , C )], whereas Equation 7 is applied on 3-D tensors [shape ( B_{m(k)} , H , C )]
- In Section 3.2.2  The authors say “This shows that the supervision used to update on batch k is not necessarily matured relative to subsequent rolling prediction spans.” This is understandable, but can the authors make more explicit that this is a problem? For instance: Is this leading to degenerate solutions? is this leading to biased forecasts? is this leading to highly correlated errors?
- On the proposed approach FAC: this is based on operations in the Frequency domain, which fits well with loss functions like MSE. Given this, is it possible to have an extension to other loss functions, like MAPE? What about the case of probabilistic or quantile forecasting models?
- On the proposed approach FAC:. Please state the shapes of $W$, $B$, and $\alpha$ (input and output), whether $\alpha$ is scalar or per-channel.

---

> ### Author Response · Authors · 2026-07-19
> **Response to Reviewer zEPJ (Part 1)**
>
> We sincerely thank the reviewer for the careful and constructive review, as well as for recognizing the relevance of the protocol-level analysis and the potential value of Frequency-Aware Calibration. We particularly appreciate the concern regarding dataset coverage and the detailed suggestions for improving the clarity of the formulation and notation. In response, we have expanded the empirical evaluation and revised the manuscript to improve its overall clarity and precision.
>
> ## Critical Change
>
> ### 1. Broader Dataset Coverage
>
> > The empirical evaluation needs to be extended to a larger collection of datasets.
>
> **Response.** We agree that the original evaluation on six datasets provided limited coverage of forecasting domains and variable scales. We have therefore expanded the evaluation to include **Electricity** and **Traffic**, increasing the total number of evaluated datasets from six to eight.
>
> Electricity and Traffic contain 321 and 862 variables, respectively, substantially exceeding the variable counts of the six primary datasets and providing additional coverage across the energy and transportation domains. Motivated by the broader benchmarking principles emphasized by GIFT-Eval, we evaluate DLinear, FreTS, and iTransformer as representative linear, MLP-based, and Transformer-based source forecasters, respectively. We evaluate each forecaster on both datasets over $H\in \lbrace 96,192,336,720 \rbrace$, resulting in **24 additional settings** across two datasets, three source forecasters, and four forecasting horizons. For each adaptation method that completes successfully, we repeat adaptation with five seeds while fixing the source forecaster checkpoint.
>
> Across the common comparison set of 21 settings, comprising all 12 Electricity settings and the nine Traffic settings with $H\in \lbrace 96,192,336 \rbrace$, the results show that:
>
> (i) FAC achieves lower MSE than the frozen source forecaster and TAFAS in **19/21** settings;
>
> (ii) FAC achieves lower MSE than PETSA in **21/21** settings; and
>
> (iii) on Traffic, FAC outperforms all three alternatives in all nine settings within this common comparison set.
>
> The three Traffic settings at $H=720$ are reported separately because TAFAS exceeds the available GPU memory for FreTS and iTransformer. At $H=720$, FAC achieves the lowest available MSE for these two source forecasters and ties PETSA for DLinear at the reported precision.
>
> These experiments broaden the evaluation in terms of both application domain and number of variables. At the same time, our evaluation remains within the standard multivariate long-term forecasting setting and does not cover the broader range of domains, frequencies, and forecasting tasks represented in benchmarks such as GIFT-Eval. We now state this remaining limitation explicitly.
>
> **Location in the revised manuscript:** Sections 5.1 and 5.2.1, Table 2; Appendix A.2; Section 6.
>
> ---
>
> ## Recommended Changes
>
> ### 2. Explicit Definition of a Mini-Batch
>
> > State the meaning of “mini-batch” explicitly.
>
> **Response.** Thank you for pointing out this potential ambiguity. We follow the terminology used in TAFAS, where consecutive rolling forecasting instances are aggregated into a “test mini-batch.” This differs from a conventional shuffled training mini-batch. We now state explicitly that, throughout this paper, a mini-batch refers to a contiguous, temporally ordered block of rolling windows from a single multivariate time series.
>
> **Location in the revised manuscript:** Section 3.1.
>
> ---
>
> ### 3. Figure 1 Notation and Readability
>
> > Figure 1 is difficult to understand, and some of its notation is used before being formally defined.
>
> **Response.** We agree that the original figure relied too heavily on notation introduced later in the manuscript. We have therefore revised both Figure 1 and its caption to clarify the protocol comparison and the notation used in the figure.
>
> First, we removed the input and prediction symbols used in the original figure and replaced them with explicit time labels marking the start and end of each input and forecast window. The revised caption also defines $L$, $H$, $B_k$, and $t_k$ directly and connects the shaded regions to the mixed-supervision objective and its two components in Eqs. 4--6, and to the streaming objective in Eq. 11.
>
> Second, the revised figure now explicitly depicts the most recent matured mini-batch, which was omitted from the original version, and distinguishes POGT supervision, supervision from matured ground truth, and streaming supervision from the current mini-batch through the figure legend.
>
> We hope these changes make Figure 1 easier to interpret before the formal notation is introduced in Section 3.1.
>
> **Location in the revised manuscript:** Figure 1 and its caption; Section 3.1.

---

> ### Author Response · Authors · 2026-07-19
> **Response to Reviewer zEPJ (Part 2)**
>
> ### 4. Point and Probabilistic Forecasting
>
> > Equation 6 and 7 are using MSE as loss functions. Does this mean that the proposed approach only works for point forecasts? What about probabilistic or quantile forecasting models?
>
> **Response.** Thank you for raising this distinction. Following the consolidation of the sample-level definitions in Section 3.1, the equations referred to by the reviewer as Eqs. 6 and 7 are now Eqs. 5 and 6, respectively, in the revised manuscript. The experiments in this work focus on deterministic point forecasting, consistent with the source forecasters and TSF-TTA baselines considered in our evaluation.
>
> The proposed supervision protocol is not inherently restricted to MSE. Its central role is to specify when target observations may be used for adaptation, while the adaptation loss can in principle be chosen according to the forecast representation. However, the current FAC implementation is designed and evaluated for deterministic point forecasts with an MSE adaptation loss. We now note in the future-work discussion that other point-forecasting losses, such as MAE or stabilized MAPE, could be explored, while extensions to probabilistic or quantile forecasting would require additional design for distributional or multi-quantile outputs.
>
> **Location in the revised manuscript:** Section 6.
>
> ---
>
> ### 5. Interpretation of Eqs. 5--7
>
> > Equations 5, 6, and 7 seem to be correct. I would suggest the authors to add a bit more motivation or clarification on the meaning of these equations.
>
> **Response.** Thank you for this suggestion. Following the consolidation of the sample-level definitions in Section 3.1, the equations referred to by the reviewer as Eqs. 5--7 are now Eqs. 4--6, respectively, in the revised manuscript. We have added a brief explanation after Eqs. 4--6 to clarify the two operational roles of the mixed supervision objective.
>
> Equation 5 uses the revealed prefix associated with the current mini-batch, enabling timely adaptation before its full target horizon becomes available. Equation 6 instead uses the complete targets of the most recent matured mini-batch and provides supervision over the full forecasting horizon. These two terms may be used in separate adaptation steps in implementation, but together they characterize the mixed supervision protocol.
>
> **Location in the revised manuscript:** Section 3.2.1, immediately following Eqs. 4--6.
>
> ---
>
> ### 6. Dimensions in Eqs. 6 and 7
>
> > It would be helpful if the authors provided more guidance on the dimensions and nuances of the notation used here. For instance, Equation 6 is applied on 2-dimensional tensors with shape $(B_k-1,C)$, whereas Equation 7 is applied on 3-dimensional tensors with shape $(B_{m(k)},H,C)$.
>
> **Response.** In the revised manuscript, the equations referred to by the reviewer as Eqs. 6 and 7 are now Eqs. 5 and 6, respectively. In Eq. 5, both arguments of the MSE contain the $B_k-1$ revealed time steps of the first sample's prediction in the current mini-batch and therefore have shape $(B_k-1)\times C$. In Eq. 6, the complete $H$-step predictions and targets are collected across the $B_{m(k)}$ samples of the most recent matured mini-batch, giving both arguments shape $B_{m(k)}\times H\times C$.
>
> We have added these dimensions immediately after Eqs. 5 and 6.
>
> **Location in the revised manuscript:** Section 3.2.1, immediately following Eqs. 5 and 6.
>
> ---
>
> ### 7. Temporal Overlap in Streaming Adaptation
>
> > In Section 3.2.2, the authors state that the supervision used to update on batch $k$ is not necessarily matured relative to subsequent rolling prediction spans. Can the authors make more explicit why this is a problem? For instance, does it lead to degenerate solutions, biased forecasts, or highly correlated errors?
>
> **Response.** Thank you for raising this question. We do not claim that this temporal overlap necessarily leads to degenerate solutions, biased forecasts, or highly correlated errors; establishing such effects would require separate empirical or theoretical analysis.
>
> Our concern is at the protocol level. As shown in Eq. 13, targets used to update the adapter on batch $k$ may also lie within the forecast spans of later mini-batches. Consequently, some targets appearing in later evaluated forecast spans may already have been used for adaptation. This lack of temporal separation makes the resulting adaptation protocol less clean and motivates our preference for supervision based only on matured ground truth.

---

> ### Author Response · Authors · 2026-07-19
> **Response to Reviewer zEPJ (Part 3)**
>
> ### 8. Extension to Other Losses and Forecasting Settings
>
> > On the proposed approach FAC: this is based on operations in the frequency domain, which fits well with loss functions like MSE. Given this, is it possible to have an extension to other loss functions, like MAPE? What about the case of probabilistic or quantile forecasting models?
>
> **Response.** Thank you for raising this important and interesting question. Although FAC is evaluated with MSE in the current work, its formulation may potentially support other objectives, since the frequency-domain correction is transformed back to the temporal domain before the adaptation loss is computed. For deterministic point forecasting, objectives such as MAE or appropriately stabilized variants of MAPE could therefore be explored without substantially changing the core FAC architecture.
>
> Extending FAC to probabilistic or quantile forecasting would be a broader direction and would require dedicated formulation and evaluation for distributional or multi-quantile outputs. We have not yet investigated these settings in the present study, but we consider this a promising extension for examining the generality of frequency-aware calibration across different forecasting objectives and output representations. We now include these directions in the future-work discussion.
>
> **Location in the revised manuscript:** Section 6.
>
> ---
>
> ### 9. Shapes of the FAC Parameters and Gates
>
> > On the proposed approach FAC: Please state the shapes of $W$, $B$, and $\alpha$ for the input and output modules, and clarify whether $\alpha$ is scalar or per-channel.
>
> **Response.** Thank you for pointing this out. Let $F_{\mathrm{in}}=\lfloor L/2\rfloor+1$ and $F_{\mathrm{out}}=\lfloor H/2\rfloor+1$ denote the numbers of frequency components. The input-side complex affine parameters satisfy $W_{\mathrm{in}}, B_{\mathrm{in}} \in \mathbb{C}^{F_{\mathrm{in}}\times C}$, while the output-side parameters satisfy $W_{\mathrm{out}}, B_{\mathrm{out}} \in \mathbb{C}^{F_{\mathrm{out}}\times C}$. The gates $\alpha_{\mathrm{in}}, \alpha_{\mathrm{out}} \in \mathbb{R}^{C}$ are therefore per-channel rather than scalar.
>
> We have added these dimensions following Eq. 16.
>
> **Location in the revised manuscript:** Section 4, immediately following Eq. 16.
>
> ---
>
> We again thank the reviewer for the detailed and thoughtful comments, which prompted us to broaden the empirical evaluation and make several valuable improvements throughout the manuscript.

---

### Review · Reviewer_FRPN · 2026-07-05

**Summary Of Contributions:**

The paper revisits test-time adaptation (TTA) protocols for time series forecasting, arguing that existing methods use revealed targets under heterogeneous adaptation schedules that lack a principled formulation. It proposes a cleaner adaptation protocol based solely on matured ground truth and introduces Frequency-Aware Calibration (FAC), a lightweight frequency-domain calibration module for adapting frozen forecasting models. Experiments across multiple datasets, forecasting horizons, and backbone models demonstrate competitive forecasting performance with substantially fewer trainable parameters.

The paper addresses an important and timely problem. The empirical evaluation is extensive and the parameter-efficiency results are compelling. However, I found the presentation substantially less clear than the underlying idea. The protocol motivation, mathematical formulation, and the proposed architecture are not connected tightly enough, making it difficult to understand exactly how FAC addresses the protocol-level problem raised in the earlier sections.

**Additional Comments:**

I appreciate the authors for tackling what I believe is an important conceptual issue in test-time adaptation. My primary concern is not the experimental evidence but the communication of the ideas. Throughout the paper I repeatedly felt that the work may be “onto something,” yet I struggled to fully understand and appreciate the contribution because the presentation emphasizes notation and implementation details over intuition. A clearer narrative connecting the protocol motivation, the proposed adaptation schedule, and the design of FAC would significantly strengthen the paper.

**Audience:**

Yes

**Audience Explanation:**

Test-time adaptation for time series forecasting is an active and important research direction, and revisiting adaptation protocols is a worthwhile contribution. The notion of separating protocol design from adapter design could have broad impact on future work in this area.

That said, I believe the paper would benefit substantially from a clearer presentation so readers can more easily appreciate its conceptual contributions.

**Broader Impact Concerns:**

I have no broader impact or ethical concerns.

**Claims And Evidence:**

Yes

**Claims Explanation:**

The experimental evaluation is comprehensive, covering multiple benchmark datasets, forecasting horizons, and backbone forecasting models. Comparisons against representative TTA baselines are thorough, and the parameter-efficiency analysis is informative. Overall, the empirical evidence supports the main performance claims.

However, my confidence in the conceptual contribution is reduced because the presentation is often difficult to follow. Several key arguments rely on intuition that is not communicated clearly enough.

For example:

* Figure 1 is intended to motivate the protocol differences but is not sufficiently self-contained.
* The connection between Figure 1 and Equations (6)-(7) is difficult to follow.
* Figure 3 introduces FAC as the proposed solution, yet it reads primarily as a new network architecture rather than a solution to the protocol issues introduced in Figures 1 and 2. Consequently, I struggled to understand how the proposed protocol and the proposed architecture reinforce one another.

**Requested Changes:**

Critical

1. Improve the overall presentation and motivation.
    Although the central idea appears relatively fundamental, the paper introduces a large amount of notation and mathematical formalism that makes the contribution harder—not easier—to understand. I believe the paper would benefit from a simpler and more intuitive presentation.
1. Strengthen the connection between the protocol discussion and FAC.
    The first part of the paper argues that existing adaptation schedules are problematic, while the second part introduces FAC. However, the transition between these two parts is weak. Figure 3 largely presents a new architecture without clearly explaining how it addresses the protocol-level issues established earlier. I would like to see a much more explicit narrative connecting:
    * the identified protocol problems,
    * the proposed matured-ground-truth protocol, and
    * why FAC is the natural solution under that protocol.
1. Improve Figure 1 and its relationship to the mathematical formulation.
    Figure 1 is arguably the most important figure in the paper but is not sufficiently self-contained. In particular, the correspondence between Figure 1 and Equations (6) and (7) is difficult to follow. Additional annotations or a clearer walkthrough would greatly improve readability.
1. Clarify the intuition behind FAC.
    Figure 3 is visually complex, yet I still did not come away with a clear understanding of the key insight (“the trick”) behind why frequency-domain calibration should work better. The paper would benefit from a more intuitive explanation before introducing implementation details.

Suggestions that would strengthen the paper

* Make the abstract more accessible to readers who are not already familiar with TSF-TTA terminology.
* Discuss more explicitly how frequency-domain calibration avoids information leakage and whether its effectiveness depends on strong periodicity or seasonality. It would also be useful to discuss expected behavior on datasets without clear seasonal patterns.
* Consider reducing unnecessary notation where possible to improve readability.

---

> ### Author Response · Authors · 2026-07-19
> **Response to Reviewer FRPN (Part 1)**
>
> We sincerely thank the reviewer for the careful and constructive assessment. We particularly appreciate the recognition that the empirical evidence supports the main claims, as well as the detailed comments on presentation, motivation, and the connection between the protocol analysis and FAC. These comments prompted us to revise the manuscript extensively so that the temporal role of supervision, the motivation for the matured-ground-truth protocol, and the design intuition behind FAC are introduced more clearly and connected more explicitly.
>
> ## Critical Changes
>
> ### 1. Overall Presentation and Motivation
>
> > Improve the overall presentation and motivation. Although the central idea appears relatively fundamental, the paper introduces a large amount of notation and mathematical formalism that makes the contribution harder—not easier—to understand. I believe the paper would benefit from a simpler and more intuitive presentation.
>
> **Response.** We agree that the original presentation introduced the formalism before providing sufficient intuition. We retain the notation needed for a precise description of the rolling forecasting setting, but have revised the presentation to make it easier to interpret and to introduce the main ideas earlier.
>
> Section 3.1 now explains the meaning of a test mini-batch before introducing the indexed formulation. We also revised Figure 1 and its caption to display the matured ground truth branch, use explicit time labels, and connect the supervision regions to the corresponding objectives.
>
> We added brief explanations after the mixed-supervision objectives to clarify their dimensions and temporal roles. We also reduced repetition by consolidating the sample-level definitions in Section 3.1 and removing a repeated definition of the matured ground truth loss in Section 3.3. Finally, Section 4 now introduces the connection between the protocol and adapter design, followed by the intuition behind FAC, before presenting the architecture and equations.
>
> **Location in the revised manuscript:** Figure 1 and its caption; Sections 3.1--3.3; opening of Section 4.
>
> ---
>
> ### 2. Connection Between the Protocol Discussion and FAC
>
> > Strengthen the connection between the protocol discussion and FAC. The first part of the paper argues that existing adaptation schedules are problematic, while the second part introduces FAC. However, the transition between these two parts is weak. Figure 3 largely presents a new architecture without clearly explaining how it addresses the protocol-level issues established earlier. I would like to see a much more explicit narrative connecting the identified protocol problems, the proposed matured-ground-truth protocol, and why FAC is the natural solution under that protocol.
>
> **Response.** We agree that the original manuscript did not explain the transition from the supervision protocol to the adapter design sufficiently clearly. We have therefore revised the opening of Section 4 to make this connection more explicit.
>
> We do not intend to claim that matured-ground-truth supervision uniquely implies a frequency-domain adapter. Rather, under the proposed protocol, adaptation no longer uses the timely feedback provided by POGT within the current mini-batch and instead relies on delayed but complete supervision from matured mini-batches. This places greater emphasis on the adapter's ability to derive useful prediction corrections from the supervision that remains available.
>
> We then examine how existing TSF-TTA adapters parameterize such corrections. Although temporal-domain transformations can realize global and frequency-selective corrections, these effects are represented indirectly through their temporal parameterization. Our empirical analysis shows that the resulting corrections are often relatively smooth and weakly selective across frequencies. This observation motivates FAC, which directly parameterizes prediction corrections through individual Fourier components.
>
> The revised narrative also clarifies the role of Figure 3: the proposed protocol determines which supervision is available for adaptation, while FAC determines how that supervision is used to calibrate predictions.
>
> **Location in the revised manuscript:** Section 4, opening paragraph.

---

> ### Author Response · Authors · 2026-07-19
> **Response to Reviewer FRPN (Part 2)**
>
> Continuing our response to the critical changes:
>
> ### 3. Figure 1 and Its Relationship to the Mathematical Formulation
>
> > Improve Figure 1 and its relationship to the mathematical formulation. Figure 1 is arguably the most important figure in the paper but is not sufficiently self-contained. In particular, the correspondence between Figure 1 and Equations (6) and (7) is difficult to follow. Additional annotations or a clearer walkthrough would greatly improve readability.
>
> **Response.** We agree that the relationship between Figure 1 and the mathematical formulation was not sufficiently clear. Following the consolidation of the sample-level definitions in Section 3.1, the equations referred to by the reviewer as Eqs. (6) and (7) are now Eqs. (5) and (6), respectively, in the revised manuscript. We have revised Figure 1 and expanded its caption and surrounding discussion to make this correspondence more explicit.
>
> The revised caption now states that, in Figure 1a, the blue POGT region corresponds to
>
> $$
> \mathcal{L}^{\mathrm{POGT}}_{k}
> $$
>
> in Eq. (5), while the green matured-ground-truth region corresponds to
>
> $$
> \mathcal{L}^{\mathrm{Matured}}_{m(k)}
> $$
>
> in Eq. (6). These two terms together form the mixed-supervision objective in Eq. (4). In Figure 1b, the red current-mini-batch supervision region corresponds to
>
> $$
> \mathcal{L}^{\mathrm{stream}}_{k}
> $$
>
> in Eq. (11).
>
> The revised discussion following Eqs. (5) and (6) further explains the dimensions and temporal roles of the two supervision terms and how they jointly define the mixed-supervision protocol.
>
> **Location in the revised manuscript:** Figure 1 and its caption; Section 3.2.1, discussion following Eq. (6).
>
> ---
>
> ### 4. Intuition Behind FAC
>
> > Clarify the intuition behind FAC. Figure 3 is visually complex, yet I still did not come away with a clear understanding of the key insight (“the trick”) behind why frequency-domain calibration should work better. The paper would benefit from a more intuitive explanation before introducing implementation details.
>
> **Response.** We agree that the original presentation moved too quickly from the motivation for FAC to its implementation details. We have revised the opening of Section 4 to state the key intuition explicitly before introducing the FreqGCM architecture.
>
> The central idea is that each Fourier basis function extends across the full temporal window. Consequently, modifying its coefficient induces coordinated changes across multiple temporal positions after the inverse Fourier transform. FAC makes the amplitude and phase of each Fourier component directly adjustable and can therefore represent structured corrections through frequency-wise affine transformations, without requiring a general dense transformation across temporal positions.
>
> This explanation now appears before the detailed description of Figure 3 and the input and output FreqGCMs, so that the architectural components are introduced only after the underlying intuition has been established.
>
> **Location in the revised manuscript:** Section 4, paragraph immediately preceding the detailed FAC architecture.

---

> ### Author Response · Authors · 2026-07-19
> **Response to Reviewer FRPN (Part 3)**
>
> ## Additional Suggestions
>
> ### 5. Abstract Accessibility
>
> > Make the abstract more accessible to readers who are not already familiar with TSF-TTA terminology.
>
> **Response.** We agree that the original abstract assumed too much familiarity with the TSF-TTA setting. We have revised it to introduce the underlying setup before using the associated terminology.
>
> The revised abstract now explains that forecasting targets become observable after predictions are made and may then be reused for adaptation. It summarizes the differences among existing adaptation protocols in plain language and defines matured ground truth as targets from fully observed forecasting horizons. We further avoid introducing more specialized terms such as POGT and mini-batches in the abstract, while retaining the main protocol and FAC contributions.
>
> **Location in the revised manuscript:** Abstract.
>
> ---
>
> ### 6. Information Leakage and Dependence on Periodicity
>
> > Discuss more explicitly how frequency-domain calibration avoids information leakage and whether its effectiveness depends on strong periodicity or seasonality. It would also be useful to discuss expected behavior on datasets without clear seasonal patterns.
>
> **Response.** We thank the reviewer for raising these important questions.
>
> First, information leakage is determined by the supervision protocol rather than by frequency-domain calibration itself. Under our proposed protocol, FAC is updated only using matured mini-batches whose complete target horizons have already been observed before the current prediction step. FAC therefore inherits this temporal separation from the proposed protocol rather than from the use of Fourier transforms.
>
> Second, the dependence of FAC on periodicity and seasonality is a valuable question that warrants further investigation. FAC is not designed around an explicit assumption of strong periodicity or seasonality, as its frequency-domain parameterization allows corrections to be distributed across the full spectrum. However, our current experiments do not isolate the effect of seasonality, so we cannot yet determine how FAC's effectiveness varies when periodic structure is weak or absent. One plausible expectation is that the learned corrections may become less concentrated in a small number of frequency components, but this hypothesis requires systematic validation. We therefore regard evaluating FAC across datasets with different degrees and types of seasonality, including datasets without clear seasonal patterns, as an important direction for future work.
>
> We have added a brief discussion in the Future Work section on systematically characterizing how FAC's effectiveness and learned corrections vary with the temporal and spectral structure of the data.
>
> **Location in the revised manuscript:** Section 3.3; Figure 3; Section 6, Future Work.
>
> ---
>
> ### 7. Reduction and Clarification of Notation
>
> > Consider reducing unnecessary notation where possible to improve readability.
>
> **Response.** We agree that the density of notation made the original presentation harder to follow. We retain the core indices needed to describe the rolling forecasting setting precisely, but have made several simplifications and added explanations for the notation that remains.
>
> In Section 3.1, we consolidated the sample-level input, forecast, and target definitions and now explain the meaning of a test mini-batch before introducing the indexed formulation. In Section 3.3, we removed a repeated definition of the matured ground truth loss and instead refer back to its earlier definition. Figure 1 now uses explicit time labels rather than sample-level input and prediction symbols, and its caption directly connects the supervision regions to the corresponding objectives.
>
> We also added brief explanations after Eqs. (5) and (6) to state the dimensions and temporal roles of the two supervision terms. We hope these revisions reduce repetition and make the remaining notation easier to interpret without removing the distinctions required by the protocol analysis.
>
> **Location in the revised manuscript:** Figure 1 and its caption; Sections 3.1--3.3.

---

### Review · Reviewer_oGif · 2026-07-11

**Summary Of Contributions:**

1.	The paper revisits TSF test-time adaptation from an evaluation-protocol perspective and distinguishes partially observed, revealed, and fully matured supervision.
2.	It argues that POGT supervision is not consistently more beneficial than using the same revealed values as input context in later rolling windows, and identifies potential look-ahead issues in some streaming protocols.
3.	It proposes a matured-ground-truth-only adaptation protocol intended to provide a more conservative and unified evaluation setting.
4.	It introduces Frequency-Aware Calibration (FAC), which performs gated complex affine calibration on input and output Fourier coefficients.
5.	Experiments across six datasets, five forecasting backbones, and multiple horizons show that FAC is generally competitive with TAFAS and PETSA under the same protocol while using substantially fewer trainable parameters.
6.	Additional analyses examine protocol variants, early versus late predictions, input- and output-side calibration, runtime and parameter efficiency, and the spectral characteristics of adaptation-induced corrections.

**Audience:**

Yes

**Audience Explanation:**

Researchers working on time-series forecasting, test-time adaptation, online learning, and evaluation methodology would likely be interested in the paper’s distinction between partially revealed and fully matured supervision. The frequency-domain adapter and parameter-efficiency analysis may also appeal to researchers studying lightweight adaptation. The audience is somewhat specialized, but the protocol clarification and adapter design are relevant to a meaningful subset of TMLR readers.

**Claims And Evidence:**

Yes

**Claims Explanation:**

1.	The parameter-efficiency claim is clearly and convincingly supported by the reported parameter counts and scaling analysis.
2.	FAC’s competitive forecasting performance is broadly supported by the extensive benchmark, but many gains are very small and no variance, confidence intervals, or significance tests are reported.
3.	The matured-only protocol is clearly formalized and causally conservative, but the evidence supports only the narrower claim that POGT supervision is not consistently necessary, not that matured-only adaptation is universally more principled or better.
4.	The frequency-domain analysis shows that FAC produces different, often more localized spectral corrections, but it does not establish that this behavior causes the accuracy gains. Parameter-matched temporal baselines and quantitative selectivity measures are missing.
5.	The practical efficiency claim should be qualified: FAC uses far fewer trainable parameters, but it is not consistently faster and may incur substantial FFT-related runtime overhead.

**Requested Changes:**

1.	The justification for matured-ground-truth-only adaptation requires clearer framing and broader validation. The current results do not show that matured-only adaptation consistently outperforms revealed-target or POGT-based supervision; Some results favor the original TAFAS protocol. The paper should therefore clarify that matured-only adaptation is competitive and methodologically more conservative, rather than universally superior, and explain the trade-off between protocol cleanliness and adaptation responsiveness. More systematic comparisons across datasets, forecasting horizons, batch sizes, source forecasters, and shift severities are needed to determine when matured-only supervision helps or hurts and to support the claim that POGT supervision is not consistently necessary.
2.	The motivation for frequency-domain calibration should be strengthened. The current justification is largely empirical: existing adapters produce smooth spectral corrections, while FAC enables more frequency-selective updates. However, the paper should better explain why this inductive bias is expected to improve forecasting under distribution shift, for example by relating it to changes in seasonality, dominant periodicity, phase, trend, or high-frequency noise. A parameter-matched temporal-domain adapter should also be included to separate the benefit of frequency-domain parameterization from that of using a smaller, more constrained model. Quantitative measures of spectral selectivity would further support the claim that FAC’s gains arise from targeted frequency correction.
3.	Strengthen the evidence for the frequency-domain design. Introduce a quantitative measure of spectral concentration or selectivity and test whether it correlates with forecasting improvement. Ablations should separately examine the multiplicative mask, additive term, complex phase modification, residual gate, input module, and output module. A parameter-matched temporal-domain counterpart is particularly important.
4.	The evaluation should include both protocol-controlled and method-native comparisons. Evaluating all methods under matured-only supervision provides an internally consistent comparison of adapter architectures, but it may not reflect the strongest versions of TAFAS and PETSA, which were designed to use revealed targets. The paper should therefore report both: (1) a controlled comparison under the same matured-only protocol, and (2) a method-native comparison using each baseline’s original supervision protocol. The latter should cover PETSA as well as TAFAS across all datasets, showing whether FAC remains competitive with the full-strength baseline methods and quantifying the practical cost of adopting the matured-only protocol.
5.	The term “cleaner protocol” should be defined more precisely or replaced. The paper appears to use it for a protocol that adapts only with fully matured targets and avoids overlap between supervision and subsequent prediction contexts. This criterion should be stated explicitly, preferably in terms of causality, target maturity, supervision delay, and target overlap. Because “cleaner” implies methodological superiority and may unfairly characterize valid POGT-based protocols as contaminated, a more neutral term such as “matured-ground-truth-only” or “non-overlapping supervision” would be preferable.
6.	The terminology should be consolidated and clarified. The paper uses several closely related terms, such as “revealed targets,” “revealed values,” “revealed ground truth,” POGT, and “matured ground truth”, without defining their relationships in one place. A terminology table or definitions box should provide each term’s formal meaning, notation, observability condition, and associated methods. A timeline-based illustration should also clarify that POGT is the currently observed subset of a target horizon, whereas matured ground truth refers to a previously predicted horizon that has become fully observable.
7.	The main results should report uncertainty and aggregate performance. Because many differences between FAC and the baselines in Table 1 are small, the experiments should be repeated across multiple random seeds or independent runs, with mean and standard deviation reported. Aggregate statistics such as win rates, average ranks, and relative improvements across datasets and forecasting horizons would further support the claim that FAC is consistently competitive.

---

> ### Author Response · Authors · 2026-07-22
> **Response to Reviewer oGif (Part 1)**
>
> We sincerely thank the reviewer for the careful and constructive review, as well as for the positive assessment of the paper’s overall evidence and relevance. The comments helped us better distinguish the methodological motivation for matured-only supervision from claims about its empirical performance. In response, we have substantially expanded the evaluation and analysis, while revising the terminology and narrowing the claims to better reflect the evidence.
>
> ## Critical Changes
>
> ### 1. Framing and Validation of Matured-Ground-Truth-Only Supervision
>
> > The justification for matured-ground-truth-only adaptation requires clearer framing and broader validation. The current results do not show that matured-only adaptation consistently outperforms revealed-target or POGT-based supervision; some results favor the original TAFAS protocol. The paper should therefore clarify that matured-only adaptation is competitive and methodologically more conservative, rather than universally superior, and explain the trade-off between protocol cleanliness and adaptation responsiveness. More systematic comparisons across datasets, forecasting horizons, batch sizes, source forecasters, and shift severities are needed to determine when matured-only supervision helps or hurts and to support the claim that POGT supervision is not consistently necessary.
>
> **Response.** We agree and clarify that our intention was not to present matured-only supervision as universally superior to mixed supervision. However, we recognize that the previous framing did not
> make this distinction sufficiently clear. We have therefore revised Section 5.3.1 to characterize matured-only supervision as a temporally conservative reference condition for comparing adapter architectures without the additional POGT signal. We also clarify that the original mixed-supervision methods combine matured ground truth with timely POGT, which can improve adaptation responsiveness.
>
> We expanded the method-native evaluation to include both TAFAS and PETSA across all 115 primary settings. Original TAFAS achieves lower MSE in 77 settings, while its matured-only variant does so in 38. Original
> PETSA achieves lower MSE in 74 settings, while its matured-only variant does so in 40, with one tie. These results show that adding timely POGT supervision is beneficial more often, but not universally, while the
> matured-only variants remain competitive in many settings. Table 5 now presents a representative subset, with the complete comparisons reported in Appendix A.4.
>
> We agree that a systematic characterization across shift types or shift severities would be valuable. However, defining and comparing shift severity across heterogeneous real-world datasets requires careful methodological choices, including how distributional change is quantified and normalized. We therefore avoid drawing a general boundary for when matured-only supervision is preferable and now state this limitation explicitly.
>
> **Location in the revised manuscript:** Section 5.3.1 and Table 5; Appendix A.4 and Tables A.8--A.9; Section 6.

---

> ### Author Response · Authors · 2026-07-22
> **Response to Reviewer oGif (Part 2)**
>
> ### 2. Motivation for Frequency-Domain Calibration
>
> > The motivation for frequency-domain calibration should be strengthened. The current justification is largely empirical: existing adapters produce smooth spectral corrections, while FAC enables more frequency-selective updates. However, the paper should better explain why this inductive bias is expected to improve forecasting under distribution shift, for example by relating it to changes in seasonality, dominant periodicity, phase, trend, or high-frequency noise. A parameter-matched temporal-domain adapter should also be included to separate the benefit of frequency-domain parameterization from that of using a smaller, more constrained model. Quantitative measures of spectral selectivity would further support the claim that FAC’s gains arise from targeted frequency correction.
>
> **Response.** We agree that the empirical correction spectra alone do not fully motivate the design. Section 4 provides a structural motivation: coefficient-wise calibration in the Fourier domain enables coordinated temporal corrections while directly adjusting the amplitudes and phases of individual components. At the same time, the relationship between
> specific distribution shifts and fixed frequency bands depends on the temporal resolution, window, and spectral structure of the data. We therefore avoid assuming a direct correspondence between them.
>
> More importantly, we added a closely parameter-matched temporal-domain counterpart to isolate the effect of the calibration domain. This counterpart preserves FAC’s element-wise affine transformation and gated residual structure, but applies them directly to temporal positions rather than Fourier coefficients. Both adapters use the same matured-only supervision protocol and MSE objective, leaving the calibration domain as the primary design difference. The temporal counterpart improves over the source forecaster in 113 of the 115 primary settings, confirming that it is itself an effective adapter. Nevertheless, FAC achieves lower MSE in all 115 matched comparisons. We now summarize this result in Section 5.3.2 and report the complete comparison in Appendix A.9.
>
> We further agree that correction spectra alone do not establish that the observed spectral structure is related to forecasting improvement. As a complementary quantitative analysis, we added an evaluation of residual-error reduction within low-, middle-, and high-frequency bands. FAC achieves the largest total residual-error reduction in all four representative settings and produces larger middle- and high-frequency reductions in three of them. The corresponding additive decomposition is also reported in Appendix A.8. Across these cases, FAC's localized correction patterns coincide with residual-error reduction in the corresponding frequency bands. We interpret this result descriptively and do not claim that spectral localization causes the overall forecasting gains.
>
> **Location in the revised manuscript:** Section 4; Section 5.3.2 and Table 6; Appendix A.8--A.9, Tables A.18--A.19.

---

> ### Author Response · Authors · 2026-07-22
> **Response to Reviewer oGif (Part 3)**
>
> ### 3. Evidence for the Frequency-Domain Design
>
> > Strengthen the evidence for the frequency-domain design. Introduce a quantitative measure of spectral concentration or selectivity and test whether it correlates with forecasting improvement. Ablations should separately examine the multiplicative mask, additive term, complex phase modification, residual gate, input module, and output module. A parameter-matched temporal-domain counterpart is particularly important.
>
> **Response.** We agree that stronger quantitative and component-level evidence is important. We did not introduce a single scalar measure of spectral concentration or selectivity in this revision, because such a measure is sensitive to its definition and may not be directly comparable across heterogeneous forecasting settings. Instead, as a complementary quantitative analysis, we measure residual-error reduction within low-, middle-, and high-frequency bands to examine whether localized correction patterns coincide with error reduction in the corresponding bands. We interpret this analysis descriptively and do not treat it as a direct measure of spectral selectivity or as evidence of a causal relationship between spectral localization and forecasting improvement.
>
> We added separate component ablations across all 115 primary settings. Full FAC achieves lower MSE than the variant without the multiplicative term in all 115 settings. It also achieves lower MSE than the variants without the additive term in 92 settings, without complex phase modification in 84 settings, and without the residual gate in 60 settings. Full FAC outperforms the input-only variant in 111 settings, while the output-only comparison indicates that input- and output-side calibration play complementary roles. The complete results are reported in Appendix A.5, and we now provide an explicit reference to these ablations in Section 5.3.
>
> We also added a closely parameter-matched temporal-domain counterpart. FAC achieves lower MSE in all 115 matched comparisons, with the complete results reported in Appendix A.9 and summarized in Section 5.3.2.
>
> **Location in the revised manuscript:** Section 5.3; Section 5.3.2 and Table 6; Appendix A.5, Tables A.10--A.16; Appendix A.8--A.9, Tables A.18--A.19.
>
> ---
>
> ### 4. Protocol-Controlled and Method-Native Comparisons
>
> > The evaluation should include both protocol-controlled and method-native comparisons. Evaluating all methods under matured-only supervision provides an internally consistent comparison of adapter architectures, but it may not reflect the strongest versions of TAFAS and PETSA, which were designed to use revealed targets. The paper should therefore report both: (1) a controlled comparison under the same matured-only protocol, and (2) a method-native comparison using each baseline’s original supervision protocol. The latter should cover PETSA as well as TAFAS across all datasets, showing whether FAC remains competitive with the full-strength baseline methods and quantifying the practical cost of adopting the matured-only protocol.
>
> **Response.** We agree that both evaluation settings are needed. Table 1 reports the protocol-controlled comparison under the common matured-only supervision condition. In response to this comment, we expanded the method-native evaluation to include both original TAFAS and original PETSA across all 115 primary settings.
>
> As detailed in our response to Comment 1, the comparisons between the original and matured-only variants quantify the practical effect of removing timely POGT supervision. We further compare FAC directly with the method-native baselines. FAC records 79 wins, two ties, and 34 losses against original TAFAS, and 102 wins, one tie, and 12 losses against original PETSA. These results show that FAC remains competitive when the baselines are evaluated under their original supervision protocols. Complete setting-level results are reported in Appendix A.4.
>
> **Location in the revised manuscript:** Section 5.2.1 and Table 1; Section 5.3.1 and Table 5; Appendix A.4, Tables A.8--A.9.

---

> ### Author Response · Authors · 2026-07-22
> **Response to Reviewer oGif (Part 4)**
>
> ### 5. Terminology for the Proposed Protocol
>
> > The term “cleaner protocol” should be defined more precisely or replaced. The paper appears to use it for a protocol that adapts only with fully matured targets and avoids overlap between supervision and subsequent prediction contexts. This criterion should be stated explicitly, preferably in terms of causality, target maturity, supervision delay, and target overlap. Because “cleaner” implies methodological superiority and may unfairly characterize valid POGT-based protocols as contaminated, a more neutral term such as “matured-ground-truth-only” or “non-overlapping supervision” would be preferable.
>
> **Response.** We agree that the term “cleaner” may imply methodological superiority over other valid supervision protocols. We have therefore removed this terminology when referring to our proposed protocol and replaced it with neutral descriptions such as “a protocol based solely on matured ground truth” and “matured-only supervision.” We also revised the relevant section and subsection titles accordingly.
>
> The proposed protocol remains operationally defined by target maturity: at mini-batch $k$, adaptation may use only mini-batches whose complete target horizons have already been observed, as formalized in Section 3.3. We use this criterion throughout the revised manuscript rather than characterizing the protocol as cleaner than POGT-based alternatives.
>
> **Location in the revised manuscript:** Abstract; Section 1; Section 3 and Section 3.3; Section 4; Section 6; Appendix B.
>
> ---
>
> ### 6. Consolidation of Target-Supervision Terminology
>
> > The terminology should be consolidated and clarified. The paper uses several closely related terms, such as “revealed targets,” “revealed values,” “revealed ground truth,” POGT, and “matured ground truth”, without defining their relationships in one place. A terminology table or definitions box should provide each term’s formal meaning, notation, observability condition, and associated methods. A timeline-based illustration should also clarify that POGT is the currently observed subset of a target horizon, whereas matured ground truth refers to a previously predicted horizon that has become fully observable.
>
> **Response.** We agree that these related terms should be consolidated. We added a terminology table in Appendix C that defines revealed targets, POGT, and matured ground truth, and clarifies that “revealed values” and “revealed ground truth” are descriptive phrases rather than separate supervision categories.
>
> We also revised Figure 1 to make the temporal distinction clearer. POGT is shown as the observed prefix of the current mini-batch target span, whereas matured ground truth comes from an earlier mini-batch whose full target horizons have become observable.
>
> **Location in the revised manuscript:** Figure 1; Appendix C.
>
> ---
>
> ### 7. Uncertainty and Aggregate Performance
>
> > The main results should report uncertainty and aggregate performance. Because many differences between FAC and the baselines in Table 1 are small, the experiments should be repeated across multiple random seeds or independent runs, with mean and standard deviation reported. Aggregate statistics such as win rates, average ranks, and relative improvements across datasets and forecasting horizons would further support the claim that FAC is consistently competitive.
>
> **Response.** We agree that repeated-run uncertainty and aggregate comparisons are important when the setting-level differences are small. We repeated each matched adaptation experiment using five adaptation seeds while fixing the source-forecaster checkpoint. Complete setting-level MSE means and standard deviations are reported in Appendix A.1, and Section 5.2.1 now summarizes the repeated-run results.
>
> We also added dataset-wise and overall aggregate analyses. FAC achieves the lowest average MSE on five of the six primary datasets. Across all 115 matched settings, it outperforms the frozen source forecaster in all 115 settings, TAFAS in 96 settings with three ties, and PETSA in 109 settings. Paired Wilcoxon tests are statistically significant for all three overall comparisons. At the dataset level, FAC significantly improves over the frozen source forecaster and PETSA on all six datasets, and over TAFAS on four of the six datasets.
>
> **Location in the revised manuscript:** Section 5.2.1; Appendix A.1, Tables A.1--A.6.
>
> ---

---

> ### Author Response · Authors · 2026-07-22
> **Response to Reviewer oGif (Part 5)**
>
> ## Additional Concern
>
> ### 8. Parameter Efficiency and Wall-Clock Runtime
>
> > The practical efficiency claim should be qualified: FAC uses far fewer trainable parameters, but it is not consistently faster and may incur substantial FFT-related runtime overhead.
>
> **Response.** We agree that trainable-parameter efficiency should be distinguished from wall-clock efficiency. The original submission already included a runtime comparison across all five source forecasters on Weather with $H=720$, together with a discussion of FFT-related overhead in the appendix and limitations. These results show that FAC is not uniformly faster than TAFAS, particularly for PatchTST, where TAFAS uses a lightweight output-only, single-channel implementation.
>
> In the revision, we renamed Section 5.2.2 as “Parameter and Runtime Efficiency” and added a main-text discussion of these results. We now explicitly limit our efficiency claim to FAC’s substantially smaller number of trainable parameters and favorable parameter scaling, without implying uniformly lower wall-clock runtime.
>
> **Location in the revised manuscript:** Section 5.2.2, Tables 3--4; Appendix A.6 and Table A.17; Section 6.
>
> ---
>
> We again thank the reviewer for the detailed and thoughtful comments, which prompted us to strengthen the empirical evaluation, narrow the protocol claims, and improve the clarity and completeness of the manuscript.

---

### Author Response · Authors · 2026-07-06
**Thank You for the Reviews**

Dear Reviewers and Action Editor,

We sincerely thank you for your thoughtful and constructive feedback. We appreciate your recognition of the relevance of test-time adaptation for time series forecasting, and your comments provide valuable guidance for improving our manuscript.

After carefully reading all three reviews, we are preparing revisions to address the main concerns raised in your comments. Our planned revisions focus on the following aspects:

1. Presentation, motivation, and readability:
We will revise the presentation to make the main ideas easier to follow, especially for readers who may not be deeply familiar with TSF-TTA protocols. In particular, the revision will make the mathematical formulation more succinct, introduce intuitive explanations before formal notation, and improve the accessibility of the abstract and motivation. Figures 1 and 3 will also be revised to make the protocol and architecture more self-contained. We will further clarify notation, mini-batch construction, tensor shapes in the loss definitions, and the current focus on point forecasting with MSE.

2. Connection between the matured-only protocol and FAC:
We will revise the transition from the protocol analysis to FAC so that the motivation for introducing FAC follows more naturally from the matured-only protocol. Rather than presenting FAC as a standalone architecture, the revision will better explain how FAC is motivated by the adaptation setting induced by the matured-only protocol. We will also add a more intuitive explanation of why frequency-domain calibration is a suitable lightweight adapter design in this setting.

3. Frequency-domain analysis:
We will strengthen the discussion and analysis of how frequency-domain corrections relate to forecasting accuracy. Beyond characterizing the correction spectra, we plan to add complementary evidence, such as error-spectrum or frequency-band analyses, to better explain when and why frequency-selective corrections are beneficial.

4. Empirical coverage:
We plan to extend the empirical evaluation to additional datasets with different temporal dynamics beyond the original six benchmarks. This additional evaluation is intended to provide a broader empirical view beyond the standard benchmark suite used in the initial submission.

5. Statistical significance and robustness:
We plan to add multi-seed results with standard deviations or confidence intervals for representative settings, and conduct appropriate significance tests where applicable. This analysis is intended to provide a more careful assessment of the observed differences, especially when the numerical margins between methods are small.

6. COSA comparison:
We will clarify why COSA is not directly included in the main comparison under the matured-only protocol. In particular, we will explain the nontrivial design choices required to adapt COSA, whose original streaming formulation relies on current-batch revealed targets and context constructed from revealed target statistics.

We will follow up with more specific responses to each reviewer and update the discussion with additional results as they become available during the response period.

Thank you again for the constructive feedback.

Best regards,

The Authors